# Climate, cryosphere and carbon cycle controls on Southeast Atlantic orbital-scale carbonate deposition since the Oligocene (30-0 Ma)

Anna Joy Drury[1,2*], Diederik Liebrand[1], Thomas Westerhold[1], Helen M. Beddow[3], David A. Hodell[4], Nina Rohlfs[1], Roy H. Wilkens[5], Mitch Lyle[6], David B. Bell[7], Dick Kroon[7], Heiko Pälike[1], Lucas J. Lourens[3]

[1]MARUM-Center for Marine Environmental Sciences, University of Bremen, Leobener Strasse 8, 28359 Bremen, Germany
[2]Department of Earth Sciences, University College London, Gower Street, London, WC1E 6BT, UK.
[3]Department of Earth Sciences, Faculty of Geosciences, Utrecht University, Utrecht, The Netherlands
[4]Department of Earth Science, University of Cambridge, Cambridge, UK
[5]University of Hawaii, School of Ocean and Earth Science and Technology, Honolulu, Hawaii 96822, USA
[6]College of Earth, Ocean, and Atmospheric Science, Oregon State University, Corvallis, Oregon 97331, USA
[7]School of GeoSciences, University of Edinburgh, Edinburgh, UK

*Correspondence to: ajdrury@marum.de; a.j.drury@ucl.ac.uk

**Abstract** The evolution of the Cenozoic cryosphere from unipolar to bipolar over the past 30 million years (Myr) is broadly known. Highly resolved records of carbonate ($CaCO_3$) content provide insight into the evolution of regional and global climate, cryosphere and carbon cycle dynamics. Here, we generate the first Southeast Atlantic $CaCO_3$ content record spanning the last 30 Myr, derived from X-ray fluorescence (XRF) ln(Ca/Fe) data collected at Ocean Drilling Program Site 1264 (Walvis Ridge, SE Atlantic Ocean). We present a comprehensive and continuous depth and age model for the entirety of Site 1264 (~316 m; 30 Myr). This constitutes a key reference framework for future palaeoclimatic and palaeoceanographic studies at this location. We identify three phases with distinctly different orbital controls on Southeast Atlantic $CaCO_3$ deposition, corresponding to major developments in climate, the cryosphere and the carbon cycle: 1) strong ~110 kyr eccentricity pacing prevails during Oligo-Miocene global warmth (~30-13 Ma); 2) increased eccentricity-modulated precession pacing appears after the mid Miocene Climate Transition (mMCT) (~14-8 Ma); 3) pervasive obliquity pacing appears in the late Miocene (~7.7-3.3 Ma) following greater importance of high-latitude processes, such as increased glacial activity and high-latitude cooling. The lowest $CaCO_3$ content (92-94%) occurs between 18.5-14.5 Ma, potentially reflecting dissolution caused by widespread early Miocene warmth and preceding Antarctic deglaciation across the Miocene Climate Optimum (~17-14.5 Ma) by 1.5 Myr. The emergence of precession-pacing of $CaCO_3$ deposition at Site 1264 after ~14 Ma could signal a reorganisation of surface and/or deep-water circulation in this region following Antarctic reglaciation at the mMCT. The increased sensitivity to precession at Site 1264 between 14-13 Ma is associated with an increase in mass accumulation rates (MARs) and reflects increased regional $CaCO_3$ productivity and/or recurrent influxes of cooler, less corrosive deep waters. The highest %$CaCO_3$ and MARs indicate the late Miocene-early Pliocene Biogenic Bloom (LMBB) occurs between ~7.8-3.3 Ma at Site 1264; broadly contemporaneous with the LMBB in the equatorial Pacific Ocean. At Site 1264, the onset of the LMBB roughly coincides with appearance of strong obliquity pacing of %$CaCO_3$ reflecting increased high latitude forcing. The global expression of the LMBB may reflect increased nutrient input into the global ocean resulting from enhanced aeolian dust and/or glacial/chemical weathering fluxes, due to enhanced glacial activity and increased meridional temperature gradients. Regional variability in the timing and amplitude of the LMBB may be driven by regional differences in cooling, continental aridification and/or changes in ocean circulation in the late Miocene.

# 1 Introduction

Over the last 30 Million years (Myr), Earth's climate system evolved considerably from the early unipolar Antarctic Coolhouse to our modern-day Icehouse world (Zachos et al., 2001; De Vleeschouwer et al., 2017, 2020; Littler et al., 2019; Westerhold et al., 2020). Inferred from benthic foraminiferal oxygen isotope data ($\delta^{18}O$), the Oligocene-early Miocene (30-17 million years ago [Ma]) was characterized by variable Antarctic ice volumes (Liebrand et al., 2017). This unipolar Coolhouse was marked by high amplitude glacial-interglacial cycles that were predominantly eccentricity paced (Wade, 2004; Pälike et al., 2006; Liebrand et al., 2016, 2017; Beddow et al., 2018). During the warm Miocene Climatic Optimum (MCO; 17-14.7 Ma), the Antarctic ice sheet shrank relative to its early Miocene size (Shevenell et al., 2004, 2008; Holbourn et al., 2015; Gasson et al., 2016; Levy et al., 2016), before prevalent unipolar conditions were re-established when Antarctica reglaciated across the mid Miocene Climate Transition (mMCT) around ~13.9 Ma (Shevenell et al., 2004, 2008; Holbourn et al., 2005, 2014; Levy et al., 2016). Following the onset of strong obliquity pacing at ~7.7 Ma (Drury et al., 2017, 2018b) and further global cooling during the late Miocene-early Pliocene (7-5 Ma, Herbert et al., 2016), a fully bipolar Icehouse world was established at ~2.7 Ma (Bailey et al., 2013).

High-resolution carbonate records provide insight into past dynamics and long-term evolution of Earth's climate and carbon cycle. Carbonate deposition is largely controlled by a combination of the amount of biogenic carbonate productivity in the surface waters and the degree of dissolution in the water column and/or seafloor, as well as sedimentary processes such as winnowing and dilution (Berger, 1970; Van Andel et al., 1975; Lyle et al., 1995, 2019; Lyle, 2003; Pälike et al., 2012). Primary and export productivity are sensitive recorders of past climate variability, responding to changes in solar insolation and nutrient availability (Coxall and Wilson, 2011; Pälike et al., 2012; Lyle and Baldauf, 2015; Carter et al., 2016; Liebrand et al., 2018). Dissolution at the seafloor is primarily driven by regional changes in the lysocline and carbonate compensation depth, with less carbonate preserving at greater depths and /or in areas with corrosive bottom waters (Berger, 1970; Van Andel et al., 1975; Lyle et al., 2019). Changes in deep-sea currents can alter the composition of the sediment through processes like winnowing or dilution, which respectively remove fine-grained material or increase certain sedimentary components relative to others (e.g., increased dilution with terrigenous material). Understanding past changes in carbonate deposition can inform about past climate development by helping to disentangle how global processes affected regional production and deposition of biogenic carbonates. Deep marine carbonate variability in the Equatorial Pacific Ocean is well-documented for the Cenozoic (Van Andel et al., 1975; Lyle, 2003; Pälike et al., 2012; Lyle and Baldauf, 2015; Kochhann et al., 2016; Beddow et al., 2018; Lyle et al., 2019). However, relatively few Atlantic records of comparable quality, resolution and extent exist (e.g., Liebrand et al., 2016), limiting our understanding of the palaeoceanographic evolution of this basin. Improving our understanding of the Southeastern Atlantic Ocean, including the Angola Basin, is of particular interest, as the water column structure, and surface and deep water ocean circulation in this region was affected by palaeoceanographic conditions in both the North Atlantic and Southern Oceans (Seidov and Maslin, 2001; Bell et al., 2015).

Here, we present the first astronomically tuned record of Southeastern Atlantic carbonate deposition spanning the last 30 Myr at orbital-scale resolution. We use expanded deep-sea sedimentary sequences from Ocean Drilling Program (ODP) Sites 1264 (Leg 208; Shipboard Scientific Party Leg 208, 2004a), located on the Angola Basin side of Walvis Ridge. New high-resolution X-ray fluorescence (XRF) core scanning data is collected for the mid Miocene-Present sediments at ODP Site 1264,
which is integrated with published Oligocene-early Miocene XRF data from ODP Sites 1264 and 1265 (Liebrand et al., 2016). The XRF ln(Ca/Fe) data is used to verify, update and revise the composite depth scale and splice at Site 1264 to form a continuous 315.96 m record. Carbonate content (%CaCO$_3$) is estimated using the XRF ln(Ca/Fe) data. We generate an astrochronology between 3 and 17 Ma using the new %CaCO$_3$ data, which is integrated with published Oligocene-early Miocene and Plio-Pleistocene age models (Bell et al., 2014; Liebrand et al., 2016). The resulting high-resolution records will
help determine shifts in the orbital pacing of Southeast Atlantic CaCO$_3$ deposition in relation to the broader climatic trends of the last 30 Myr. We investigate how widespread Miocene warmth followed by Antarctic glaciation influenced the pacing and preservation of Southeast Atlantic carbonate deposition. Finally, we establish the relative timing of the late Miocene-early Pliocene Biogenic Bloom (LMBB; acronym from Lyle et al., 2019) in the Southeast Atlantic versus Pacific Oceans and explore what this reveals about the global and regional driving forces of this multi-million-year productivity event.

## 2 Materials and Methods

### 2.1 ODP Sites 1264 and 1265

This study utilises material recovered at ODP Site 1264 on the Angola Basin side of Walvis Ridge in the Southeastern Atlantic (Fig. 1; 28∘31.955′S, 2∘50.730′E, 2505 m water depth; Shipboard Scientific Party Leg 208, 2004a), which was drilled during ODP Expedition 208 to provide a Cenozoic deep-sea record of the South Atlantic (Shipboard Scientific Party Leg 208,
2004a, 2004b). At Site 1264, a continuous ~316 m shipboard stratigraphic splice was developed back to the early Oligocene using magnetic susceptibility and 600/450nm colour reflectance data (Shipboard Scientific Party Leg 208, 2004a). Oligocene-early Miocene XRF ln(Ca/Fe) data were used to verify the shipboard splice and produce a revised composite depth (rmcd) scale (Liebrand et al., 2016, 2018). Liebrand et al. (2016) filled four short Oligo-early Miocene core gaps at Site 1264 with data from Site 1265 (Fig 1; 28∘ 50.101′ S, 2∘ 38.354′ E, 3059 m water depth; Shipboard Scientific Party Leg 208, 2004b) to
provide a continuous sedimentary sequence. Site 1264 and the relevant intervals of Site 1265 are characterised by high biogenic carbonate, with shipboard analysis indicating an average CaCO$_3$ content of 92-96 weight% (Shipboard Scientific Party Leg 208, 2004a, 2004b). At 1264, shipboard linear sedimentation rates (LSR) derived from bio-magnetostratigraphy were exceptionally low for the early-mid Miocene (19-12 Ma; LSR = 3.9-5.4 m/Myr, average of 4.7 m/Myr). Higher shipboard LSR occurred during the Oligo-early Miocene (30-19 Ma; LSR = 5.3-9.3 m/Myr, average of 7.1m/Myr) and early Plio-Pleistocene
(3-0 Ma; LSR = 4.5-7.4 m/Myr, average of 6.0 m/Myr). The highest shipboard LSR occurred in the late Miocene-early Pliocene (12-3 Ma), where LSR average 15.9 m/Myr (7.7-30.5 m/Myr. The shipboard LSR for the Oligocene-early Miocene (30-17 Ma) and Plio-Pleistocene (5.3-0.0 Ma) were confirmed by previous studies on Sites 1264 and 1265, which also support the

shipboard notion that these sites are excellent recorders of orbital-scale climate dynamics (Liebrand et al., 2011, 2016, 2017, 2018; Bell et al., 2014, 2015).

## 2.2 Core images

To assist with splice verification, astronomical tuning and data interpretation, we compiled composite core images for ODP Site 1264 from both cropped line-scan images and core table top photos using Code for Ocean Drilling Data (CODD 2.1 – www.codd-home.net; Wilkens et al., 2017). After cropping the original line scan images (JANUS – http://www-odp.tamu.edu/database/), core section images were compiled into a single image for each core and scaled to depth using the "Includes_Core_Image_Assembly" functions. The line scan images obtained during ODP Leg 208 "Walvis Ridge" are redder in colour than they appear in the core table-top photos. This is likely an artefact of the line scanning calibration, which had only recently been introduced at the time (Fig. 1). The individual core images were also compiled from lighting corrected table-top core-box photos using the "Includes_Core_Table_Photos" functions. These table-top core-box images more realistically represent the original colour of the cores. However, because the core-box-derived images are exceptionally white, we preferentially use the line scan-derived images, as the sedimentary cyclicity at Site 1264 is better visible in these images, and thus are more beneficial to evaluating and revising the stratigraphic splice together with XRF and physical property data (see Fig 2; Section 2.3 and 3.1). The individual core-box/line-scan core images were then combined into a single composite image along the revised Site 1264 splice using the "SpliceImages" function. The individual core-box/line-scan core images from Site 1265 (Westerhold et al., 2017) were spliced into the four Oligo-early Miocene core gaps in the 1264 splice image using the Site 1264 to Site 1265 ties from (Liebrand et al., 2016) updated to accommodate any splice revisions (see also Section 3.1; Supplementary Figure 5 and Table 5). This resulted in a continuous spliced image of the sedimentary succession at Site 1264-1265 spanning the early Oligocene to present day.

## 2.3 X-ray fluorescence core scanning

XRF core scanner data were collected at ODP Site 1264 between 0-205 rmcd (revised metres composite depth) to connect with previously published XRF core scanning data spanning  205-315.96 rmcd (Liebrand et al., 2016). The new XRF data were generated in four measurement campaigns in 2011 (195-205 rmcd), 2013 (29.21-153.28 rmcd), 2017 (141.49-195.12 rmcd) and 2018 (0-33.35 rmcd). Ca, Fe, K, Mn, Si and Ti were measured directly at the core surface of Site 1264 archive halves using a 10 kV run at 1-2 cm resolution over a 1.2 cm$^2$ area with a slit size of 10 mm down-core and 12 mm cross-core with XRF Core Scanner II/III (AVAATECH Serial No. 2/12) at the MARUM - University of Bremen. In 2018, Ba and Sr were additionally measured with the same slit conditions using a 50 kV run across intervals that proved more problematic to accurately correlate between holes. The following settings were used: 2011) MARUM XRF III, 10 kV/0.15 mA/10 s count time/Cl-Rh filter (see also Liebrand et al., 2016); 2013) MARUM XRF III, 10 kV/0.2 mA/15 s count time/Cl-Rh filter; 2017) MARUM XRF II, 10 kV/0.15 mA/15 s count time/no filter; 2018) MARUM XRF III, 50 kV/0.5 mA/7 s count time/Cu filter, 10 kV/0.035 mA/7 s count time/no filter. The split core surface was covered with a 4 μm thick SPEXCerti Prep Ultralene1 foil

to avoid contamination of the XRF measurement unit and desiccation of the sediment. Selected intervals were rerun during successive campaigns to account for differences in measurement intensity between datasets, including with the published Liebrand et al. (2016) data. All data were inspected directly following collection and outliers were removed if they were clearly associated with cracks and/or uneven sediment surface. A further four measurement points, often at section ends, were removed because intensities were unrealistically low. All earlier scanned XRF data were calibrated to the 2018 dataset using an individual linear regression for each element between 2018 and 2011 (including Liebrand et al., 2016), 2013 and 2017 respectively (see Supplementary Information and Supplementary Figure 1 and 2). All XRF core scanning intensity data and derived information are reported in Supplementary Table 1.

## 2.4 XRF-derived CaCO$_3$ estimates and CaCO$_3$ MARs

Liebrand et al. (2016) showed that the XRF ln(Ca/Fe) ratio shows a strong positive correlation with shipboard coulometric CaCO$_3$ data from Site 1264 (Shipboard Scientific Party Leg 208, 2004a). We use a similar approach to generate a continuous record of CaCO$_3$ content using the combined Site 1264 XRF ln(Ca/Fe) dataset calibrated to the shipboard CaCO$_3$ data (%CaCO$_3$ = 80.238±1.069 + (2.526 ± 0.188* ln(Ca/Fe)); r$^2$ = 0.622; Supplementary Figure 6). The Oligocene-early Miocene calibration used by Liebrand et al. (2016) is within the 2σ uncertainty of the new %CaCO$_3$ calibration, which equates to ±2.2% in the calibrated %CaCO$_3$ dataset. The uncertainty in the calibration likely originates from the scatter of the shipboard coulometry-derived %CaCO$_3$ data that were used in the calibration. This uncertainty only pertains to the absolute %CaCO$_3$ values. The trends and cyclicity observed in the calibrated CaCO$_3$ data are independent of this uncertainty, as these patterns are present in the raw ln(Ca/Fe) timeseries.

The new and recalibrated %CaCO$_3$ data from Site 1264 were combined with the %CaCO$_3$ data from Site 1265 (Liebrand et al., 2016) to form a ~315 m/~30 Myr continuous record of CaCO$_3$ content at Walvis Ridge. Bulk and CaCO$_3$ mass accumulation rates (MARs; g/cm$^2$/kyr) were calculated using the following formulas:

$$MAR_{Bulk} = \rho_{dry} \times LSR \tag{1}$$

$$MAR_{CaCO_3} = \rho_{dry} \times LSR \times \left( \frac{\%CaCO_3}{100} \right) \tag{2}$$

$$MAR_{detrital} = MAR_{Bulk} - MAR_{CaCO_3} \tag{3}$$

The LSR (cm/kyr) was calculated using the new astrochronology (Section 4). Dry bulk density $\rho_{dry}$ (g/cm$^3$) was estimated using the shipboard gamma ray attenuation (GRA) bulk density data calibrated to the shipboard discrete dry density data (Shipboard Scientific Party Leg 208, 2004a) (Supplementary Figure 7). The uncertainty in the MARs is difficult to quantify. The largest uncertainties affecting bulk, CaCO$_3$ and detrital MARs arise from uncertainties in the $\rho_{dry}$, which was calculated using shipboard GRA and discrete dry density data, and the LSR, both of which are difficult to estimate. CaCO$_3$ MARs additionally have ±2.2% 2σ calibration uncertainty. However, as %CaCO$_3$ is so high at Site 1264, the %CaCO$_3$ calibration

uncertainty will have a smaller affect compared with the changes in LSR. Because detrital MARs are low and calculated using the difference between bulk and $CaCO_3$ MARs, changes in detrital MARs should be treated cautiously.

## 3 Results

### 3.1 Site 1264 splice revision, off-splice mapping, and the 1264 to 1265 correlation

The line-scan core photos and XRF data, especially the ln(Ca/Fe) and Ba data, show that there are several misalignments in sedimentary features when using the shipboard composite depth scale, leading to duplicated and/or missing intervals in the shipboard splice (Fig. 2). These misalignments are especially pronounced in intervals where the shipboard physical property data was low amplitude due to very high $CaCO_3$ content, e.g., during the late Miocene-early Pliocene interval. Predominantly using the XRF ln(Ca/Fe) ratio, the shipboard splice was verified between 0 and 205 rmcd and revised where needed (Fig. 2,

Supplementary Figure 3; Supplementary Tables 2 & 3). Where the inter-hole correlation based on the ln(Ca/Fe) ratio was ambiguous, the Ba data were used (see Supplementary Figure 3). Revisions were made between 27 and 149 rmcd, and generally resulted in changes of less than 0.6 m relative to the shipboard composite depth (Supplementary Tables 2 & 3), with the exception where Core 1264A-11H was shifted by -1.26 m relative to Core 1264B-11H to improve the correlation (Fig. 2). The core images, the new ln(Ca/Fe) data and shipboard 600/450nm colour reflectance and MS data (Shipboard Scientific Party

Leg 208, 2004a) were also used to map off-splice intervals of Holes 1264A and 1264B onto the splice between 0 and 196.13 rmcd (Supplementary Table 4).

    Liebrand et al. (2016, 2018) previously revised the Oligo-early Miocene interval of the composite depth scale, stratigraphic splice and mapping pairs for Site 1264, as well as the Site-to-Site correlation between Sites 1264 and 1265. The revisions in the upper sedimentary succession (0-205 rmcd) result in a cumulative shift of -1.96 m to the Liebrand et al. (2016, 2018) Site

1264 composite depth scale, stratigraphic splice, mapping pairs and Site 1264 depths in the Site 1264 to Site 1265 correlation between 196.13 rmcd and 315.96 rmcd (Supplementary Tables 2-5). Furthermore, the line-scan composite core photos showed that the mapping of Core 1264B-29H to the splice, erroneously corrected in Liebrand et al. (2018) from original mapping in Liebrand et al. (2016), should be adjusted (see Supplementary Figure 4 and Table 4). The composite depth scale and splice revisions resulted in two small gaps in the Plio-Pleistocene Site 1264 isotope record (Bell et al., 2014): ~15 cm (~17 kyr)

between 27.25-27.40 rmcd and ~25 cm (~11 kyr) between 54.18-54.43 rmcd, which were filled with new isotope data (Westerhold et al., 2020).

### 3.2 Site 1264 XRF intensities, $CaCO_3$ estimates and MARs

    The XRF-derived $CaCO_3$ content at Site 1264 is generally high throughout, ranging between minimum values of 92% and maximum values of 97% (Fig 3). The range of observed %$CaCO_3$ variability is close to the 2.2% uncertainty associated with

the calibration. However, we are confident that both the long-term trends and short-term variability discussed below represent true changes in carbonate content, as these patterns originate in the original ln(Ca/Fe) ratio. The calibration uncertainty is most

relevant to the absolute carbonate content. The recalibrated CaCO$_3$ content span 93-96% between ~205 and ~316 rmcd (early Oligocene-early Miocene), which agrees within error with the original calibrated CaCO$_3$ content reported and discussed in detail by Liebrand et al. (2016). The lowest CaCO$_3$ content (92-93%) occurs between 205 and 190 rmcd (mid Miocene). The CaCO$_3$ content shows especially clear 0.5 m cycles in this interval. CaCO$_3$ content increases slightly to 94% between 190 and

180 rmcd (mid Miocene) and then remains around 94-95% until ~118 rmcd (early late Miocene). The CaCO$_3$ content initially displays short 0.6 m cycles, but after ~185 rmcd (mid Miocene), 0.2-0.3 m cycles are superimposed upon ~1-2 m cycles. CaCO$_3$ content then undergoes a two-step rapid increase to 96% between ~118 and 110 rmcd and again to 97% between 105 and 100 rmcd (both latest Miocene). CaCO$_3$ content remains around 97% until 90 rmcd, after which the CaCO$_3$ content decreases slightly to around 96% until 40 rmcd (early Pliocene). Between ~118 and ~40 rmcd (latest Miocene-early Pliocene),

~1.0 m cycles and occasionally ~0.5 m cycles are prevalent, although the amplitude of the short-term cycles is reduced compared to the deeper interval. CaCO$_3$ content slowly drops to 95% by 15 rmcd (Pleistocene) and decrease further to 93-94% in the upper 15 m of the record. Short-term cycles are less well expressed in this upper interval. The Si and K intensities are comparable throughout the record, although Si is generally slightly higher than K (Fig 3). Both elements, together with Fe and Ti intensities, display the same short-term variability and long-term trends (Fig 3 and Supplementary Figure 2), indicating that

these elements reflect changes in aluminosilicates. As the trends of Si and K are inverse to those seen in the CaCO$_3$ content, this supports that Site 1264 is predominantly composed of carbonate and clay, with minimal influence of biogenic silica. The amplitude of changes in Si and K becomes much smaller relative to CaCO$_3$ content changes between ~115-0 rmcd compared to ~315-115 rmcd.

Because CaCO$_3$ content accounts for >90% of the sediment mass, the bulk (0.3-4.7 g/cm$^2$/kyr) and CaCO$_3$ (0.3-4.5

g/cm$^2$/kyr) MARs are remarkably similar, with trends controlled almost completely by variability in LSR (Fig 3). LSR also strongly affect detrital MARs; however, these remain low throughout at Site 1264 (0.01-0.2 g/cm$^2$/kyr). The Oligocene-early Miocene CaCO$_3$ MARs generally oscillate between 1-2 g/cm$^2$/kyr from ~315 to 205 rmcd. MARs are very low (~0.3-0.7 g/cm$^2$/kyr) between 205 and 190 rmcd, before slowly increasing to 1.0-2.5 g/cm$^2$/kyr between 190 and ~118 rmcd. The highest CaCO$_3$ MARs (2.5-4.5 g/cm$^2$/kyr) occur between ~118 and 35 rmcd, with values decreasing back to 1-2 g/cm$^2$/kyr after ~35

rmcd. The highest frequency variability in the bulk and CaCO$_3$ MARs results from changes in dry bulk density (Shipboard Scientific Party Leg 208, 2004a); however, this variability is smaller than that variability reflecting the changing resolution of the astrochronology (see section 4.2).

## 4 Depth and age models for Site 1264

### 4.1. Cyclostratigraphy and initial bio-/magnetostratigraphic age model

Here, we describe the imprint of cyclic patterns on the Site 1264 CaCO$_3$ content (Fig 3). New cyclostratigraphy covers the upper ~205 m of the sedimentary succession at Site 1264, which corresponds to strata of middle Miocene to late Pleistocene age. Cyclostratigraphy of the lowermost ~111 m from Site 1264 (between ~205 and ~316 rmcd), which corresponds to the

early Oligocene to early Miocene time interval, was previously described in great detail (Liebrand et al., 2016). A cycle interpretation and age model were also previously presented for the upper ~57 m from Site 1264 (Bell et al., 2014); however, due to several splice revisions between 27 and 55 rmcd (see Section 3.1) we briefly re-evaluate the cycle imprint on this part of the record (see Sections 4.1.3 and 4.1.4).

The upper ~205 m is split into four intervals that are characterized by distinct cyclic patterns and/or average sedimentation rates (Sections 4.1.1, 4.1.2, 4.1.3 and 4.1.4). We apply an $11^{th}$ order polynomial age model computed on selected (i.e. high-quality) bio- and magnetostratigraphic depth-age points to obtain a first-order approximation of the durations of the cycles that we identify in the depth domain (see Supplementary Table 6, Supplementary Figures 8 and 9). After applying the polynomial age model, the record was tuned to generate an astrochronology (Section 4.2; Fig 4 and Supplementary Figure 10)

### 4.1.1 Depth interval between 205 and 190 rmcd

The depth-domain wavelet analysis of the $CaCO_3$ content between 205 and 190 rmcd highlights the lithological cycles in %$CaCO_3$, which broadly varies around 2 and 0.5 m in length (Fig 3) The bio-/magnetostratigraphic age model indicates that the LSR vary around 5 m/Myr. Applying this LSR to the 2 and 0.5 m cycles yield durations of approximately 405 and ~110

kyr, respectively. These durations are in very close agreement with the strong eccentricity pacing of $CaCO_3$ content variability found for the underlying early Oligocene-early Miocene sediment package (Liebrand et al., 2016). We infer that eccentricity pacing of the carbonate record remained dominant from the base of Site 1264 to ~190 rmcd regardless of the changes in LSR, which were lower between 205 and 190 rmcd compared to the deeper interval (Fig 3). This interpretation is in agreement with visual inspection of the data, which shows bundling of ~110 kyr cycles (e.g. the ~95 and ~125 kyr cycles) within longer 405

kyr cycles (Fig 3 and 4, panel I.a).

### 4.1.2 Depth interval between 190 and 115 rmcd

The depth interval between 190 and 115 rmcd is marked by cycles that gradually respectively shift from 0.2 to 0.3 m, from ~1 to ~2 m, and from ~4 to ~6 m in the depth-domain wavelet analysis of the $CaCO_3$ data (Fig 3). The bio-magnetostratigraphic

age model indicates that these gradually shifting, quasi-stable cyclicities in the depth domain reflect low, but gradually increasing LSR from ~10 m/Myr between 190 and 160 rmcd to ~15 m/Myr between 160 and 115 rmcd. Based on this initial age model, we tentatively link the 0.2 to 0.3 m cycles to precession, the ~1 to ~2 m cycles to ~110 kyr eccentricity and the ~4 to ~6 m cycles to 405 kyr eccentricity (Supplementary Figure 9). These inferred durations of these cycles correspond to known ratios between precession, short and long eccentricity of five precession cycles per ~110 kyr cycle, and about four ~110 kyr

cycles per 405 kyr cycle. Overall, the bio-/magnetostratigraphic age model suggests that the ~110-kyr eccentricity cycle remains the most strongly expressed cycle between 17 and 13 Ma, in line with the strong ~110-kyr eccentricity cycles observed between 30 and 17 Ma (see Section 4.1.1. and Liebrand et al., 2016). Strong ~110-kyr eccentricity cycles were also noted in this interval in the equatorial Pacific (Kochhann et al., 2016). The presence of a weak 405 kyr signal in the Site 1264 $CaCO_3$

content contrasts with the Oligocene interval, for which no uniform imprint of the 405 kyr cycle on $CaCO_3$ content could be discerned (Liebrand et al., 2016).

### 4.1.3 Depth interval between 115 and 35 rmcd

Because of several splice revisions in the upper 55 rmcd of Site 1264 (see Section 3.1.), we deem a modest re-evaluation of the cyclostratigraphy for this interval beneficial for subsequently obtaining a final tuned age model (see also Section 4.1.4.), even though detailed investigations were previously made (Bell et al., 2014). Visible inspection of the $CaCO_3$ content data and the associated depth-domain wavelet analysis both show that there is short-term cyclicity present in the data between 115 and 35 rmcd (Fig 3). However, the amplitude of these cycles is much reduced in comparison to the previous depth intervals, which means that the cycles are not statistically significant above the 95% level in the depth-domain wavelet analyses. Nevertheless, we document depth periodicities of ~0.5 m, ~1 m, ~3-4 m, and ~10-12 m. We compute average LSR of 20 to 30 m/Myr based on the bio-/magnetostratigraphic ages, and tentatively infer that these depth cycles are respectively linked to the 20 kyr precession (~0.5 m), the 40 kyr obliquity (~1 m), and the ~110 and 405 kyr eccentricity cycles (~3-4 and ~10-12 m) (Fig 3; Supplementary Figures 8 and 9). Between 55 and 35 rmcd, we visually derive an antiphase relationship between $CaCO_3$ content data and benthic foraminiferal $\delta^{18}O$ data (Supplementary Figure 11), which aids our tuning approach for this interval (see Section 4.2.2).

### 4.1.4 Depth interval between 35 and 0 rmcd

At Site 1264, clear cyclicity is generally hard to observe in the upper interval of the depth-domain $CaCO_3$ content wavelet analysis, except for the presence of occasional stronger ~1.0-1.5 m cycles. Visually, we can identify higher frequency cycles in the $CaCO_3$ content data, however the amplitude of these cycles is muted compared to the cycles observed between 115 and 35 rmcd. Benthic foraminiferal $\delta^{18}O$ maxima (Bell et al., 2014) appear to coincide with $CaCO_3$ content minima in the upper 35 m, however this phase relationship is not well-defined throughout this interval and becomes less clear at the top of the record. We derive averaged LSR of <10 m/Myr for 0-35 rmcd based on the initial bio-/magnetostratigraphic age model. The observed 1.0 to 1.5 m cycles are probably linked to the ~110-kyr eccentricity paced cycles or the main ice age cycles of the middle and late Pleistocene. This would indicate a change in response of both benthic foraminiferal $\delta^{18}O$ and $CaCO_3$ content during this time interval, in line with the evolution of the global cryosphere and climate systems during this time (Bailey et al., 2013). Based on the initial age model we note absence of clear precession and obliquity paced cyclicity in both benthic foraminiferal $\delta^{18}O$ and $CaCO_3$ content records during the last 2.5 Ma (Supplementary Figure 9).

### 4.2 Astronomically tuned age model

Two published astrochronologies exist for Sites 1264/1265: 1) an Oligo-early Miocene one (30 to 17 Ma) based on tuning $CaCO_3$ content to eccentricity (Liebrand et al., 2016); and 2) a Plio-Pleistocene one (5.3 to 0 Ma) based on a correlation between the Site 1264 benthic foraminiferal $\delta^{18}O$ record to the LR04 Plio-Pleistocene benthic $\delta^{18}O$ stack (Lisiecki and Raymo,

2005; Bell et al., 2014). Because of the splice revisions between 27 and 149 rmcd at Site 1264, we re-evaluated the Bell et al. (2014) chronology in the early Pliocene prior to 3.5 Ma/~27 rmcd. The Oligocene to early Miocene astrochronology remains unchanged, but we updated the depth-age tie points to accommodate the cumulative -1.96 m shift in the revised composite depth scale of the overlying sedimentary sequence. The cyclostratigraphic analyses in the depth-domain indicate that the combined Site 1264/1265 CaCO₃ content record is suitable for developing an astrochronology for the interval 17-3.5 Ma (see Section 4.1) using the flexible best-practice guidelines outlined in Sinnesael et al. (2019). Because of the variable imprint of eccentricity (E), obliquity (T) and precession (P) recorded at Site 1264, it was not possible to implement a uniform tuning strategy for the entire record. In all, we employed three distinct strategies to achieve a 30 Myr astrochronology for Site 1264 (Supplementary Table 7 and Figure 10):

I) 30-8.0 Ma: CaCO₃ content tuned to eccentricity; obliquity is also used in 2.4 Myr minima (when stable in the solution):

    I.a) 30-9.7 Ma: CaCO₃ content tuned to E (visually aided by δ¹⁸O, where available)

    I.b) 9.7-8.0 Ma: CaCO₃ tuned content to E(T)

II) 8.0-3.3 Ma: CaCO₃ content tuned to ET-P (visually aided by δ¹⁸O, where available)

III) 3.3-0.0 Ma: benthic δ¹⁸O tuned to LR04

## 4.2.1 early Oligocene-late Miocene (30.0-8.0 Ma)

Between 30-17 Ma, Liebrand et al. (2016) showed that CaCO₃ content maxima coincide with benthic δ¹⁸O maxima, which are both antiphase with the ~110 kyr eccentricity components. They generated an astrochronology by tuning CaCO₃ content minima to eccentricity maxima (see Liebrand et al., 2016 for details). As the variability and dominant cyclicity in CaCO₃ content for the 17-8 Ma interval are comparable to the 30-17 Ma interval (see Section 4.3.1 and 4.3.2), we consider the inverse phase relationship between CaCO₃ content and ~110 kyr eccentricity to be valid between 30-8 Ma. We therefore also employ the Liebrand et al. (2016) tuning strategy of CaCO₃ content minima to eccentricity maxima between 17-8 Ma (Fig 4). When benthic foraminiferal stable isotope records become available for the interval between 17-8 Ma, the stability of the Oligo-late Miocene phase relationship between CaCO₃ content and ~110 kyr eccentricity can be tested.

The CaCO₃ content to eccentricity tuning strategy is very robust where the amplitude modulation of ~110 kyr eccentricity is high; however, this amplitude is muted during 2.4 Myr eccentricity minima (~17.0-16.6 Ma, ~14.6-14.2 Ma, ~12.6-12.2 Ma, ~9.7-9.3 Ma). The imprint of obliquity is apparent in these 2.4 Myr eccentricity minima and can act as an alternative tuning target when ~110 kyr eccentricity amplitude is reduced. However, because of uncertainties in past changes to tidal dissipation and dynamical ellipticity, the exact phase of obliquity is not known before 10 Ma (Lourens et al., 2004; Zeeden et al., 2013, 2014). We therefor apply two slightly adapted approaches of the Liebrand et al. (2016) tuning strategy to the 17-8 Ma interval:

I.a) From 17-9.7 Ma, we tune CaCO₃ content minima to E maxima (La2004; Laskar et al., 2004), with an uncertainty better than ±50 kyr. This uncertainty increases to up to ±100 kyr in the 2.4 Myr minima at ~17.0-16.6 Ma, ~14.6-14.2 Ma, ~12.6-12.2 Ma (Fig 4, Panel Ia).

I.b) From 9.7-8.0 Ma, we tune $CaCO_3$ content minima to E(T) maxima (La2004; Laskar et al., 2004). Generally, the $CaCO_3$ content minima are tuned to E maxima, with an uncertainty better than ±50 kyr. During the 2.4 Myr eccentricity minima ~9.7-9.3 Ma, $CaCO_3$ content minima are tuned to ET maxima (uncertainty up to ±40 kyr; Fig 4, Panel I.b).

We chose to tune to the La2004 solution (Laskar et al., 2004), as over the last 30 Myr the eccentricity components are essentially identical to the La2011_ecc3L solution (Laskar et al., 2011) used in Liebrand et al. (2016). Furthermore, the obliquity solution used in I.b) approach is currently only available in the La2004 solution.

There was potential to develop an astrochronology at precession-level, as the cyclostratigraphic analyses show that precession cycles are also imprinted in the $CaCO_3$ content younger than 14 Ma (see Section 4.1). However, as uncertainties in past tidal dissipation and dynamical ellipticity mean the phase of precession is also uncertain before 10 Ma (Lourens et al., 2004; Zeeden et al., 2013), we chose a conservative strategy of only tuning to eccentricity prior to 9.7 Ma.

### 4.2.2 late Miocene-mid Pliocene (8.0-3.3 Ma)

After 8 Ma, the ~110 kyr eccentricity imprint on $CaCO_3$ content decreases significantly, whilst the imprint of obliquity and precession is more prevalent between 8 and 2.5 Ma. We therefore apply a different tuning strategy between 8.0 and 3.3 Ma to accommodate the change in prevalent cyclicity in the $CaCO_3$ content from eccentricity/precession-driven (older than 8 Ma) to obliquity/precession-driven (younger than 8 Ma) (see Section 4.1). The contrasting relationship between benthic $\delta^{18}O$ and $CaCO_3$ content in the latest Miocene-Pleistocene compared to the Oligo-early Miocene also indicates that different tuning approaches are warranted. (6.0-3.3 Ma; Where latest Miocene-Pleistocene benthic $\delta^{18}O$ and $CaCO_3$ content are both available (6.0-3.3 Ma; Bell et al., 2014; Westerhold et al., 2020), the two proxies show an inverse relationship, with the obliquity- and precession-driven $CaCO_3$ content minima coinciding with benthic $\delta^{18}O$ maxima. This contrast to the Oligocene-early Miocene relationship between these proxies with the 110-kyr eccentricity-driven $CaCO_3$ content minima coinciding with benthic $\delta^{18}O$ minima (Liebrand et al., 2016). The late Miocene-Pleistocene phase relationship between benthic $\delta^{18}O$ and obliquity is well established, with benthic $\delta^{18}O$ minima coinciding with obliquity maxima (Shackleton et al., 1995; Shackleton and Hall, 1997; Hodell et al., 2001; Zeeden et al., 2013; Drury et al., 2017, 2018b). As the relationship between benthic $\delta^{18}O$ and $CaCO_3$ content is inverse after 6.0 Ma, we assume that $CaCO_3$ content maxima correlate to obliquity maxima for the entire 8 to 3.3 Ma interval. Precession and obliquity are the two prevalent cyclicities present in both the $CaCO_3$ content and benthic $\delta^{18}O$ data. The interference pattern observed in both benthic $\delta^{18}O$ and $CaCO_3$ content is most similar to an ET-P solution. We therefore generated an astrochronology by tuning $CaCO_3$ maxima to ET-P maxima guided by benthic $\delta^{18}O$ where these are available (uncertainty up to ±10 kyr; Fig 4, Panel II). Based on the shipboard biostratigraphy, there was some indication that there might be an unconformity of ~0.6 Myr in the late Miocene at the base of Core 1264A-7H (~76-77 rmcd) (Shipboard Scientific Party Leg 208, 2004a). However, we find excellent agreement between the $CaCO_3$ content and the ET-P solution, with 80 and 75 rmcd correlating well with 6.3-6.1 Ma (Fig 4, Panel II).

**4.2.3 mid Pliocene-Pleistocene (3.3-0.0 Ma)**

We used the original Bell et al. (2014) age model between 3.3 and 0 Ma, because no changes were made to the shipboard splice in the upper 27 rmcd (3.5 Myr). The Bell et al. (2014) age model was generated by correlating a benthic foraminiferal $\delta^{18}O$ stack comprising data from ODP Sites 1264 and 1267 to the LR04 $\delta^{18}O$ stack (Lisiecki and Raymo, 2005). We validated the Bell et al. (2014) age model by comparing the Site 1264 $\delta^{18}O$ record to the equatorial Atlantic CR17 $\delta^{18}O$ stack (Wilkens et al., 2017), which has an independent tuning based on MS and lightness. The agreement between the 1264 and CR $\delta^{18}O$ records is very good (Fig 4, Panel III), which further supports the accuracy of the original Bell et al. (2014) age model in this interval.

**5 Discussion**

**5.1 History of South Atlantic CaCO₃ deposition and its changing orbital pacing since the Oligocene**

Previous work at ODP Site 1264 shows that the recovered sediments record the orbital climate variability of the Southeast Atlantic for the last 30 Myr (Shipboard Scientific Party Leg 208, 2004a; Bell et al., 2014, 2015; Liebrand et al., 2016). Oligocene to early Miocene carbonate and benthic $\delta^{18}O$ records from Site 1264 show that the early Coolhouse was dominated by large ~110 kyr eccentricity driven variability in the Antarctic ice sheet (for full discussion, see Liebrand et al., 2016, 2017). By the Pliocene, Atlantic benthic $\delta^{13}C$ gradients indicate that North Atlantic Deep Water (NADW) heavily influenced Southeast Atlantic Site 1264 (for full discussion, see Bell et al., 2014, 2015). The new complete and continuous depth (~316 m; Section 3.1; Fig 3) and age (~30 Myr; Section 4; Fig 4) model presented here constitutes a reference framework for future palaeoclimatic and palaeoceanographic studies at Site 1264. Furthermore, the new data (Fig 5) enable investigation of how long-term climate trends and orbital-scale climate variability impacted this region, especially between 17 and 5 Ma, for which no high-resolution Southeast Atlantic records previously existed.

At Site 1264, CaCO₃ content is very high, with all long- and short-term variability occurring between 92-97.5% CaCO₃ (Fig 5). CaCO₃ content varied between 94-96% during the Oligocene-early Miocene (30-18.5 Ma), with MARs of ~1-2.5 $g/cm^2/kyr$ and are discussed in greater detail in Liebrand et al. (2016). The lowest CaCO₃ content (92-93.5%) and MARs (~0.3-0.7 $g/cm^2/kyr$) occur between ~18.5-14.4 Ma, which broadly coincides with the Miocene Climatic Optimum (MCO; 17-14.7 Ma; Shevenell et al., 2004; Holbourn et al., 2005) (see Section 5.2 for discussion). Broadly concurrent with cooling in the lead up to the mid Miocene climate Transition (mMCT; ~13.9 Ma), CaCO₃ content increases and remains between 94-96% during the early late Miocene (14.4-8.0 Ma), coincident with MARs of ~1-2.5 $g/cm^2/kyr$ (see Section 5.2 for discussion). The highest CaCO₃ content (96-97.5%) and MARs (2.5-4.5 $g/cm^2/kyr$) occur between 8-4 Ma (Fig 5), potentially indicating high carbonate productivity coincident with the known age of the global Late Miocene-early Pliocene Biogenic Bloom (LMBB; acronym from Lyle et al., 2019) (see Section 5.3 for further discussion).

Carbonate deposition is strongly affected by the balance between biogenic carbonate productivity (mostly in the surface water) and carbonate dissolution in the water column/at the sea floor. Sedimentary processes, such as dilution with terrigenous material and/or the removal of fine-grained material through winnowing, can affect both the amount and composition of the carbonate preserved. The relative importance of biogenic productivity versus dissolution is discussed in detail in Liebrand et al. (2016) for the Oligocene to early Miocene, in Section 5.2 for the early-mid Miocene, and in Section 5.3 for the late Miocene-early Pliocene. Over the last 30 Myr, detrital MARs are low, indicating that dilution with terrigenous material was not a major contributing factor in controlling carbonate deposition at Site 1264. Winnowing may have removed fine fraction material, including coccolith carbonate, thereby reducing carbonate deposition at Site 1264. By comparing MARs between nearby sites recovered during DSDP Leg 74, Shackleton et al. (1984) suggested that winnowing may have affected parts of the Walvis Ridge. They suggested that winnowing was especially pronounced at DSDP Site 526 (1054 m water depth) since the late Oligocene. Site 1264 is situated on a very gentle slope above the lysocline and carbonate compensation depth (palaeowater depths: 2-2.5 km). Winnowing likely had less effect on Site 1264 compared to Site 526, as Site 1264 is not positioned on the shallowest parts of the Walvis Ridge bathymetry. Nonetheless, Shackleton et al. (1984) also found some indication of winnowing at DSDP Site 525 (2467 m water depth) since the late Pliocene. Independent constraints on winnowing are not available for the entire 30 Myr interval; however, detailed fine fraction weights are available between 30 and 17 Ma (Liebrand et al., 2016; their Fig. 2). If these data are interpreted as a proxy for winnowing, this would suggest that winnowing is modest during the "mid" Oligocene, increasing during late Oligocene warming and relatively high across the Oligocene-Miocene Transition (Fig 5). During the early Miocene (post OMT, pre-mid Miocene) winnowing is comparable to late Oligocene values (Fig 5). There is evidence for winnowing to have increased towards the condensed middle Miocene part of the Site 1264 record, as there is an increase in both high-resolution and low-resolution percent >63 μm coarse fraction (%CF) (Liebrand et al., 2016; Keating-Bitonti and Peters, 2019) (Fig 5). However, between 18.5 and 8 Ma, the Site 1264 %CF varies within a 5% range, suggesting the amount of winnowing remained stable (Fig 5; Keating-Bitonti and Peters, 2019). After ~3 Ma, %CF gradually increases from 20 to 40% (Fig 5), which is the largest increase seen in the entire record and could indicate that Site 1264 is affected by winnowing at this time. The presence of winnowing is also supported by the fact that deeper Walvis Ridge Sites 1266 and 1267 both have higher sedimentation rates than Site 1264 in the last 3 Ma, whereas the opposite would be expected if deep-sea dissolution alone was considered (productivity should affect all sites similarly).

The influence of the long 405 kyr eccentricity on $CaCO_3$ deposition at Site 1264 is complicated. The imprint of the 405 kyr cycle on $CaCO_3$ content was not constant during the Oligocene interval at Site 1264 (Liebrand et al., 2016). This contrasts with the clearer imprint of a 405 kyr cycle on $CaCO_3$ deposition during the Miocene between ~21-5 Ma. For periodicities shorter than 405 kyr, we can recognise three distinctly different orbital imprints on the variability in $CaCO_3$ content (Fig 5; Fig 6):

1) ~110 kyr eccentricity is the dominant driver between 30 and ~13 Ma (Fig 6.A);
2) Precession-driven $\%CaCO_3$ oscillations appear ~14-13 Ma and are the main pacer of short-term variability until ~8 Ma (Fig 6.B);

3) Obliquity becomes a significant driver at Site 1264 after ~7.7 Ma, and together with precession imprints a characteristic interference pattern on %CaCO$_3$ (Fig 6.C);

Although different tuning strategies are used to generate a continuous astrochronology (Section 4.2), these shifts in imprinted cyclicity are also visible in the depth and bio- and magnetostratigraphic age model spectra, indicating that the shifts are independent of the changes in tuning strategy (see Section 4.1 and Supplementary Figure 9).

The three pacings observed in Southeastern Atlantic CaCO$_3$ deposition broadly coincide with major developments in climate, the cryosphere and/or the carbon cycle over the last 30 Myr. At Site 1264, the strong expression of ~110 kyr eccentricity-driven %CaCO$_3$ variability between 17-13 Ma parallels the dominant pacing of %CaCO$_3$ during the Oligo-early Miocene (30-17 Ma; Fig 5 and Fig 6.A; for further detail see Liebrand et al., 2016). The prevalence of ~110kyr eccentricity pacing at Site 1264 is in line with the wider understanding that Oligocene to mid Miocene unipolar Coolhouse climate was predominantly paced by short-term eccentricity during widespread global warmth (Pälike et al., 2006; Tian et al., 2013; Holbourn et al., 2014, 2015; Beddow et al., 2016, 2018; Voigt et al., 2016; Kochhann et al., 2016; Liebrand et al., 2016, 2017; De Vleeschouwer et al., 2017, 2020; Westerhold et al., 2020). The strong ~110-kyr cyclicity observed in marine archives is attributed to eccentricity-driven changes in ice volume and/or deep-sea temperature, likely associated with changes in atmospheric CO$_2$ (Pälike et al., 2006; Holbourn et al., 2015; Liebrand et al., 2017; Greenop et al., 2019).

The orbital imprint on CaCO$_3$ content shifts between 14 and 13 Ma, when eccentricity-modulated precession cycles progressively become more clearly superimposed on the larger ~110 kyr cycles. These precession cycles remain the main driver of carbonate deposition until ~8 Ma, although obliquity cycles are visible during the 2.4 Myr eccentricity minima from ~12.6 to 12.2 Ma and ~9.7 to 9.3 Ma, when the imprint of precession and ~110 kyr eccentricity is muted (Fig 5 and 6.B). Strong obliquity was also observed in benthic $\delta^{18}$O data from the South China Sea during the ~9.7-9.3 Ma node (Holbourn et al., 2013). The strong obliquity intervals that are observed across multiple marine archives, support that obliquity exerts greater control on the climate system as a whole when the orbital configuration is characterised by long-term eccentricity minima coincident with long-term obliquity maxima (Holbourn et al., 2013, 2018; Drury et al., 2017; Levy et al., 2019). The shift to stronger precession pacing occurs after global cooling and the reglaciation of Antarctica across the mid-Miocene Climate Transition (~13.9 Ma; mMCT; Holbourn et al., 2005). Some precession-driven %CaCO$_3$ cycles were observed, superimposed on larger ~110 kyr eccentricity cycles, between 23.5-19.5 Ma at Site 1264 (Fig 5; Liebrand et al., 2016). However, the relative amplitude of eccentricity and precession is different in the mid-late Miocene compared to the Oligocene-early Miocene. In the Oligocene-early Miocene, the amplitude of the ~110 kyr eccentricity cycles in CaCO$_3$ content were greater than the precession-driven CaCO$_3$ content cycles. In contrast, we observe a decrease in strength of the ~110 kyr eccentricity cycles concurrent with the strong precession-pacing of the CaCO$_3$ content between 14 and 8 Ma (Fig 5). The influence of early-mid Miocene climate evolution on Southeast Atlantic carbonate deposition is discussed further in Section 5.2.

Although some power remains in the ~110 kyr eccentricity bandwidth, the orbital imprint seen in CaCO$_3$ content changes around 7.7 Ma to a strong obliquity-precession interference pattern, which remains visible until ~3.3 Ma (Fig 5 and 6C). The onset of prevalent obliquity-precession pacing of %CaCO$_3$ observed at Site 1264 after ~7.7 Ma has been observed globally in

benthic $\delta^{18}$O records and is associated with increased influence of high-latitude processes, such as enhanced glacial activity and high-latitude cooling (Drury et al., 2016, 2017, 2018b; Holbourn et al., 2018; see also Section 5.3). Although benthic $\delta^{18}$O data are not available at Site 1264 between 8.0-6.0 Ma, the obliquity-precession interference pattern is visible in the benthic $\delta^{18}$O record between 6.0 and 3.3 Ma (Fig 4). Relative to the Oligocene-early late Miocene, the amplitude of the variability in CaCO$_3$ content is reduced during the latest Miocene and early Pliocene. Concurrent with the waning influence of ~110 kyr eccentricity at Site 1264, the highest CaCO$_3$ content (96-97.5%) of the entire record occur between 8 and 4 Ma. The influence of the complex late Miocene climate system on carbonate deposition is discussed in Section 5.3.

After 3.3 Ma, the short-term orbital imprint is more difficult to characterise. The wavelet analysis shows that ~110 kyr eccentricity influence increases in the Plio-Pleistocene compared to the latest Miocene (Fig 5). The influence of some obliquity and precession forcing on %CaCO$_3$ remains until ~0.9 Ma, when the ~110 kyr eccentricity pacing characteristic of the middle Pleistocene appears after the Mid Pleistocene Transition (MPT; Bell et al., 2014). Compared to Site 1264, the transition from ~40 kyr to ~110 kyr pacing is recorded more clearly at nearby Site 1267, where it is visible in the benthic $\delta^{18}$O data, composite core photos and physical property data (physical property data from Shipboard Scientific Party Leg 208, 2004c; benthic $\delta^{18}$O data from Bell et al., 2014; composite core photos from Westerhold et al., 2017). This difference in expression of the MPT may partly relate to water depth differences between the sites, as the deeper Site 1267 (4356 m water depth) may record a stronger deep-water signal compared to Site 1264 (2507 m water depth). Alternatively, winnowing may have obscured some of the cyclicity at Site 1264, considering the indication that both Sites 1264 and 525 (both ~2.4-2.5 km water depth) were affected by winnowing in the late Pliocene-early Pleistocene. Nonetheless, although the onset of the Pleistocene ~110-kyr cycles is not exceptionally clear at Site 1264, it is apparent that these cycles only appear after 0.9 Ma at both Sites 1264 and 1267 (Fig 5), which is considerably later than in the eastern equatorial Pacific, where ~110-kyr cycles first appear in carbonate records at 1.6 Ma (Lyle et al., 2019).

Benthic foraminiferal $\delta^{18}$O records are only available for the Oligocene-early Miocene (30-17 Ma; Liebrand et al., 2011, 2016) and the Plio-Pleistocene (5.3-0.0 Ma; Bell et al., 2014). It is therefore not yet possible to track the evolution of the relationship between the climate-cryosphere system (encompassed by benthic $\delta^{18}$O) and South Atlantic carbonate deposition over the last 30 Myr. However, in contrast to the in-phase %CaCO$_3$-benthic $\delta^{18}$O relationship on ~110-kyr eccentricity periodicities between 30 and 17 Ma (Liebrand et al., 2016), the new Site 1264 %CaCO$_3$ data has an inverse relationship on obliquity periodicities with benthic $\delta^{18}$O for the last 6.0 Myr (Fig 4; Supplementary Figure 11). This points to a considerably different relationship between the cryosphere and controls on carbonate deposition at Site 1264 in the late Miocene-Pleistocene compared to the Oligocene-early Miocene. The in-phase Oligocene to early Miocene %CaCO$_3$-benthic $\delta^{18}$O relationship on ~110 kyr periodicities observed at Site 1264 has been observed elsewhere for the Oligocene through to the mid Miocene, including across the mMCT (Holbourn et al., 2014, 2015; Kochhann et al., 2016; Liebrand et al., 2016; Beddow et al., 2018; Tian et al., 2018). It is possible that the %CaCO$_3$-benthic $\delta^{18}$O relationship changed from in-phase on 110-kyr eccentricity periodicities to anti-phase on obliquity periodicities, concurrent with the ~7.7 Ma shift in CaCO$_3$ deposition from a predominantly eccentricity/precession-paced system to one that is more controlled by obliquity/precession. Such an

interpretation would further support the notion that the Earth's system underwent a major shift in its response to orbital forcing in the late Miocene-early Pliocene, with Northern hemisphere high-latitude processes steadily growing in importance in the latest Miocene (Turner, 2014; Drury et al., 2017, 2018b ; De Vleeschouwer et al., 2020).

### 5.2 Eccentricity-Precession switch, low %$CaCO_3$ deposition and the early-mid Miocene warmth

The early-mid Miocene marks a warm interval where Antarctic ice volume underwent major change and climatic trends deviated from the overall Cenozoic cooling pattern (Miller et al., 1991; Shevenell et al., 2004; Holbourn et al., 2005, 2014, 2015; Tian et al., 2013, 2014; Super et al., 2018). The Miocene Climatic Optimum (MCO; defined in the benthic foraminiferal $\delta^{18}O$ as between 17 and 14.7 Ma; Holbourn et al., 2015), was characterised by pervasive global warmth and more humid conditions, together with lower meridional temperature gradients and greatly reduced continental ice sheets on Antarctica compared to the present-day (Lear et al., 2000, 2015; Billups and Schrag, 2002; Shevenell et al., 2004; John et al., 2011; Pound et al., 2012; Gasson et al., 2016; Levy et al., 2016). Distal marine records that track variations in land ice volume and deep-sea temperatures are marked by a strong ~110 kyr eccentricity pacing, coupled with large 400-kyr driven carbon cycle perturbations (Monterey Excursion) (Shevenell et al., 2008; Holbourn et al., 2014, 2015; Tian et al., 2014; Kochhann et al., 2016; Ohneiser and Wilson, 2018). The MCO warmth, ice volume decrease and carbon cycle perturbations have been hypothesized to be driven by increased atmospheric $CO_2$ levels associated with volcanic degassing from the Columbia River Flood Basalts, with the earliest eruptions occurring after ~17.2 Ma (Foster et al., 2012; Barry et al., 2013; Greenop et al., 2014; Kasbohm and Schoene, 2018; Moore et al., 2018, 2020; Super et al., 2018; Cahoon et al., 2020; Sosdian et al., 2020). The warm MCO conditions were reversed ~13.9 Ma during the mMCT when major continental ice sheets reappeared on Antarctica associated with a large decrease in atmospheric $CO_2$ and global temperatures (Shevenell et al., 2004; Holbourn et al., 2005; Foster et al., 2012; Pound et al., 2012; Badger et al., 2013; Lear et al., 2015; Gasson et al., 2016; Levy et al., 2016; Super et al., 2018, 2020).

Benthic foraminiferal $\delta^{18}O$ records are not yet available at Site 1264 for this interval, so it is not possible to recognise the MCO using this dataset. Nonetheless, the lowest %$CaCO_3$ content (92-93.5%) and MARs (~0.3-0.7 g/cm$^2$/kyr) occur between ~18.5-14.4 Ma, and broadly coincide with the MCO (Fig 5C and 7C). Low detrital MARs (bulk–$CaCO_3$ MARs), Si and K intensity indicates that biogenic silica and detrital input remains relatively constant and minimal across this interval. The low detrital MARs at Site 1264 (average 0.09 g/cm$^2$/kyr) are comparable to the non-carbonate MARs of nearby sites drilled during Leg 74, particularly DSDP Site 525 (Shackleton et al., 1984). Dilution was therefore not the main driving factor of the early-mid Miocene low %$CaCO_3$ content at Site 1264. Winnowing could have removed the <63 μm fraction at Site 1264 (Fig 5); however, such winnowing also tends to remove both small $CaCO_3$ and detrital particles, ultimately raising the overall $CaCO_3$ content but lowering the $CaCO_3$ MAR (Marcantonio et al., 2014). A 10% increase in the percent >63 μm coarse fraction (%CF) after ~18.5 Ma (Fig 5; Liebrand et al., 2016) indicates some winnowing occurred. However, between 18.5 and 8 Ma, the Site 1264 %CF varies within a 5% range, but never increases to the high %CF values seen in the Plio-Pleistocene (Fig 5; Keating-Bitonti and Peters, 2019). This indicates increased dissolution and/or decreased productivity likely also drove the early-mid

Miocene low CaCO$_3$ content at Site 1264. An increase of B/Ca concentration at Sites 1264 and 1266 after 15.5 Ma (Kender et al., 2014) indicates that dissolution influenced the early-mid Miocene low CaCO$_3$ content at Site 1264.

The recovery of %CaCO$_3$ content ~14.5 Ma agrees well with the end of the MCO ~14.7 Ma (Holbourn et al., 2015). However, at Site 1264, the decreasing CaCO$_3$ content starts ~18.5 Ma, which is ~1.5 Myr before the decrease in benthic $\delta^{18}$O normally associated with the onset of the MCO (Fig 5). During the early-mid Miocene, low %CaCO$_3$ and CaCO$_3$ MARs were observed at multiple sites in the eastern equatorial Pacific Ocean (EEP; DSDP Site 574; IODP Sites U1335-U1338), initially decreasing after 18-17.5 Ma, before recovering to early Miocene values by 15-14.5 Ma (Piela et al., 2012; Kochhann et al., 2016). Multiproxy evidence at these EEP sites indicates that the low %CaCO$_3$ and CaCO$_3$ MARs values were associated with increased deep-sea dissolution rather than decreased productivity, with the peak dissolution occurring at the onset of the MCO (Piela et al., 2012; Kochhann et al., 2016). This dissolution horizon has been traced regionally across the equatorial Pacific as the "Lavender" seismic unconformity, with the dissolution potentially linked to the intensification of proto-NADW formation leading to increased corrosive Antarctic Bottom Water (AABW) reaching the Pacific (Mayer et al., 1985). This hypothesis could not be tested at the time due to the absence of any comparable Atlantic carbonate records. However, the new evidence of low %CaCO$_3$ and CaCO$_3$ MARs at Site 1264 in the Southeast Atlantic indicates that dissolution occurred in the Atlantic and the Pacific during the early to mid-Miocene. Increased dissolution across ocean basins indicates a global forcing, supporting suggestions that the dissolution seen in the Pacific was associated with elevated atmospheric $p$CO$_2$, increased carbon storage in the deep ocean and shoaling of the carbonate compensation depth during the early-mid Miocene global warmth (Pälike et al., 2012; Piela et al., 2012; Kochhann et al., 2016).

Assuming that the low CaCO$_3$ content and MARs at Site 1264 is dissolution driven (e.g., see also Kender et al., 2014), rather than reflecting a decrease in carbonate rain, there is evidence that carbonate dissolution preceded the MCO by ~1.5 Myr in the Southeastern Atlantic (Site 1264) and ~1.0-0.5 Myr in the equatorial Pacific (Piela et al., 2012; Kochhann et al., 2016). Few early-mid Miocene atmospheric CO$_2$ or sea surface temperatures (SST) records extend back to 18.5 Ma, but long-term trends in early-mid Miocene TEX$_{86}$-derived SSTs from the North Atlantic Ocean indicate that SSTs may have been at MCO-levels since ~20 Ma (Super et al., 2018; Fig 7A). It is not yet clear whether elevated SSTs prior to the MCO are a global phenomenon. However, the likely dissolution-induced lows in Southeastern Atlantic and equatorial Pacific CaCO$_3$ deposition up to ~1.5 Myr before the MCO, indicate that the MCO itself was preconditioned by elevated temperatures and atmospheric $p$CO$_2$.

Shortly after the MCO, the overall Cenozoic cooling trend resumes across the mMCT with the reappearance of large ice sheets on Antarctica around 13.9 Ma. At Site 1264, the %CaCO$_3$ values increase after ~14.5 Ma, which could reflect decreased deep-sea dissolution and/or increased surface-ocean productivity. Between 14 and 13 Ma, the orbital imprint on CaCO$_3$ at Site 1264 progressively shifts from ~110 kyr eccentricity-dominated pacing to precession-dominated pacing superimposed on the ~110 kyr eccentricity cycles (Fig 6A, B and 7C). In comparison to the Oligocene-early Miocene, the ~110 kyr eccentricity cycles are more muted, and the precession cycles are of higher amplitude and more clearly expressed during the mid-late Miocene (Fig 5 and 6B). The change in orbital imprint at Site 1264 after the mMCT may indicate that productivity in this

region became more sensitive to precession forcing following changes to ocean circulation and/or the hydrological cycle driven by the reglaciation of Antarctica, global cooling and increased meridional temperature gradients. The shift towards strong precession-pacing occurs as carbonate content recovers after the MCO. Site 1264 likely experienced increased carbonate deposition during the MCO as indicated by low LSR and MARs between 18.5-14.4 Ma. The increased preservation of

precession cycles at Site 1264 after 14 Ma (i.e., after the mMCT) could also reflect a shift in deep-water circulation patterns bringing less corrosive deep-waters to Site 1264, which is supported by the increase in B/Ca at Sites 1264 and 1266 (Kender et al., 2014). This deep-water change would have enabled better preservation of precession-driven productivity cycles after the mMCT compared to the early-middle Miocene.

## 5.3 Late Miocene-early Pliocene Biogenic Bloom

The latest Miocene (~8-5.3 Ma) is a complicated and dynamic interval when climate and ecosystems recognisable to the present-day first appeared (Herbert et al., 2016). There is abundant evidence for a global and long-lasting increase in primary productivity in the global surface ocean during the late Miocene to early Pliocene (Farrell et al., 1995; Dickens and Owen, 1999; Diester-Haass et al., 2002, 2005). This Late Miocene-early Pliocene Biogenic Bloom (LMBB; as defined by Lyle et al., 2019) has been recognised between ~8 and 4 Ma in upwelling and oligotrophic areas of all major oceanic basins (Kroon et al.,

1991; Dickens and Owen, 1999; Hermoyian and Owen, 2001; Diester-Haass et al., 2002, 2004, 2005; Grant and Dickens, 2002; Liao and Lyle, 2014; Lyle and Baldauf, 2015; Lyle et al., 2019). At Sites 1264/1265, the highest $CaCO_3$ content (96-97.5%) occurs between ~8 and 4 Ma and the highest bulk and $CaCO_3$ MARs (~2-4.5 $g/cm^2/kyr$) are found between ~7.8 and 3.3 Ma (Fig. 5 and 8C), which falls within the broad timing associated with the LMBB. As %$CaCO_3$ is so high at Site 1264, $CaCO_3$ MARs account for most of the variability in the bulk MARs. Similarly, Lyle et al. (2019) showed that the LMBB is

expressed between ~8 and 4.4 Ma in the bulk and $CaCO_3$ MAR of 6 sites in the eastern equatorial Pacific (EEP), with $CaCO_3$ MARs accounting for most of the bulk MAR variability (Fig 8D). Despite the influence of palaeogeographical heterogeneity on the absolute EEP MARs, it becomes apparent after normalisation that common productivity patterns are visible across the EEP (Lyle et al., 2019; Fig 8B and D). The Site 1264 $CaCO_3$ MARs are generally higher than the EEP sites (Fig. 8C and D), except for ODP Sites 849 and 850, which are the two highest sedimentation EEP sites (Lyle et al., 2019).

The exact cause of the LMBB is poorly understood, however key hypotheses suggest the increased primary productivity was caused by 1) increased nutrient input into the surface ocean through increased weathering/dust input and/or 2) changes to the global distribution of nutrients through changes in atmospheric and oceanic circulation patterns. The widespread documentation of the LMBB shows that the expression and timing is regionally variable (Liao and Lyle, 2014; Lyle et al., 2019; Sutherland et al., 2019). However, most of these records are low-resolution and insufficient for accurately constraining

regional differences, so we cannot yet distinguish between global changes to the nutrient budget and changes to the regional distribution of nutrients in the ocean (e.g., changes in ocean circulation and/or upwelling). The availability of orbital-scale %$CaCO_3$ and MARs from Site 1264 and the EEP provide the opportunity to compare the LMBB at high-resolution for the first time. Based on increased $CaCO_3$ MARs, the timing of the LMBB at Site 1264 and the EEP generally agrees well (Fig 8B),

which corroborates the global nature of the LMBB. After increasing from 8 Ma onwards, MARs peak between 7.2-6.6 Ma at both Walvis Ridge and the equatorial Pacific (Fig. 8B), which supports a global LMBB optimum occurring between ~7.0-6.4 Ma (Lyle and Baldauf, 2015; Lyle et al., 2019). Site 1264 also has the highest absolute %CaCO$_3$ values between 7.2-6.6 Ma, indicating high productivity of carbonate producers at this time (Fig 5 and 8A). However, the LMBB extends to 3.3 Ma at Site 1264, in contrast to the western EEP, where the LMBB ends ~4.4 Ma (Fig 8B). In the far eastern equatorial Pacific near South America, high CaCO$_3$ MARs continue to ~3 Ma (Figure 8 in Lyle et al., 2019), which is further evidence for regional variability of the termination of the LMBB production interval. The recognition of global patterns and temporal heterogeneity in the expression of the LMBB between the Pacific and the Southeast Atlantic could reflect different regional responses to a single climatic forcing and/or multiple driving forces.

Constraining which primary producers drove the LMBB at different regions will be useful for disentangling regional and global patterns. During the latest Miocene-early Pliocene (~8-3 Ma), the new Site 1264 %CaCO$_3$ data displays an inverse relationship with the low-resolution record of the percent >63 μm coarse fraction (%CF) (Keating-Bitonti and Peters, 2019; adapted to this study's new composite depth and age model; Fig 5C and 8A). The %CF specifically shows the opposite trend to %CaCO$_3$ across the LMBB: decreasing %CF from 8 Ma, with the lowest %CF values occurring ~7 Ma, in line with the maximum values in %CaCO$_3$. Through the mid-late Miocene and early Pliocene, the LSR at Site 1264 are either similar or higher at Site 1264 (2505 m) relative to deeper Site 1266 (3806 m). The available %CF and %CaCO$_3$ from Site 1264 also do not display a strong relationship prior to 8 Ma. This suggests that any winnowing at Site 1264 was minimal and stable for the mid-late Miocene to early Pliocene. The inverse %CF-%CaCO$_3$ relationship therefore could indicate that the LMBB was predominantly driven by a change in the calcareous phytoplankton (coccolithophores) versus foraminifera ratio at Site 1264. Based on Si intensity, there is no evidence that biogenic silica producers play a major role in the LMBB at Site 1264 (Fig. 5). This contrasts to the EEP, which is upwelling dominated and where a combination of calcareous (coccolithophores) and siliceous (diatoms) phytoplankton drove the LMBB (Lyle and Baldauf, 2015; Lyle et al., 2019).

Although we cannot yet accurately distinguish global increases in nutrient delivery to the ocean versus the regional redistribution of nutrients causing localised increased primary productivity, we can consider links between this prolonged productivity event and the dynamic changes observed during late Miocene. Terrestrial and sea surface temperatures decreased rapidly during the late Miocene cooling between ~7.0-5.4 Ma (Pound et al., 2011, 2012; Herbert et al., 2016). There is evidence for dynamic ice sheet activity, although there is no major late Miocene increase in benthic δ$^{18}$O records suggesting that there was no long-term expansion in continental ice sheet extent or substantial deep-sea cooling (Hodell et al., 2001; Drury et al., 2016, 2017, 2018b; Holbourn et al., 2018; Tian et al., 2018). The carbon cycle underwent major change in the atmospheric, terrestrial and marine realms, with evidence for an atmospheric $p$CO$_2$ decrease around ~8-7 Ma (Bolton and Stoll, 2013; Herbert et al., 2016; Mejía et al., 2017; Tanner et al., 2020), the rise of terrestrial C$_4$ plants on land (Cerling et al., 1997; Behrensmeyer et al., 2007; Uno et al., 2016; Tauxe and Feakins, 2020), and the globally synchronous marine late Miocene carbon isotope shift (LMCIS; ~7.5-6.9 Ma) linked to global changes in oceanic circulation (Haq et al., 1980; Hodell and Venz-Curtis, 2006; Reghellin et al., 2015, 2020; Drury et al., 2017, 2018a).

At Site 1264, the onset of elevated $CaCO_3$ MARs (~7.8 Ma) roughly coincides with the shift from eccentricity-precession pacing to pervading obliquity-precession pacing of %$CaCO_3$, which infers an increased influence of high-latitude processes in the Southeast Atlantic (Fig. 5). The onset of strong obliquity pacing is also observed ~7.7 Ma as asymmetric (i.e., sawtooth-shaped) benthic $\delta^{18}O$ cycles, which have a characteristic "interglacial-glacial" anti-phase relationship with benthic $\delta^{13}C$ on obliquity timescales (Drury et al., 2017). The appearance of strong obliquity forcing in multiple systems shortly after 8 Ma implies increased influence of high-latitude climate processes, such as increased glacial activity and high-latitude cooling. There is widespread evidence that the late Miocene cooling was especially pronounced in the high latitudes and reached near-modern gradients around 5.4 Ma (Pound et al., 2012; Herbert et al., 2016). The growing importance of the high-latitudes in the latest Miocene is further supported by evidence that deep-sea stable $\delta^{13}C$ and $\delta^{18}O$ switched from in-phase to anti-phase on eccentricity timescales (Kirtland Turner, 2014; De Vleeschouwer et al., 2020), as a result of continental carbon reservoirs shrinking during cold periods due to increased extent of low-carbon Arctic biomes, such as ice sheets, polar deserts and tundra (De Vleeschouwer et al., 2020). There is also ice-proximal evidence for enhanced glacial activity in both the Northern and Southern Hemispheres, potentially indicating early transient bipolar cryosphere activity in the latest Miocene (Connell et al., 1996; Fronval and Jansen, 1996; Wolf-Welling et al., 1996; Kong et al., 2010; Williams et al., 2010). Increased glacial weathering after 8 Ma may have contributed to the onset of the LMBB through increased the nutrient influx into the ocean. An increased nutrient flux may also be driven by enhanced chemical weathering through Himalayan uplift and the intensification of the Indian and Asian Monsoon systems in the latest Miocene (Kroon et al., 1991; Filippelli, 1997; Zhisheng et al., 2001; Holbourn et al., 2018; Yang et al., 2019). Finally, the LMBB may be partly driven by increased nutrient input into the ocean as a result of widespread continental aridification coupled with trade wind intensification due to greater meridional gradients during the late Miocene cooling (7-5.4 Ma) (Hovan, 1995; Filippelli, 1997; Diester-Haass et al., 2006; Tipple and Pagani, 2007; Lyle et al., 2008; Pound et al., 2012; Herbert et al., 2016).

Regional variability in the LMBB may in turn be driven by regional differences in the extent of late Miocene cooling (Pound et al., 2012; Herbert et al., 2016), as well as regional diachrony in aridification (Molnar, 2005; Schuster et al., 2006; Lyle et al., 2008; Dupont et al., 2013). The LMCIS (~7.5-6.9 Ma) has been linked to the onset of near modern thermohaline circulation with NADW percolating further into the South Atlantic (Hodell and Venz-Curtis, 2006; Drury et al., 2017; Keating-Bitonti and Peters, 2019). A major shift in oceanic circulation would likewise affect the redistribution of nutrients around the globe, thereby potentially contributing to regional differences in nutrient supply. All these aspects go some way to explain why the LMBB began after ~8 Ma; however, it is unclear why the LMBB continued into the Pliocene, and especially why it continued until 3.3 Ma at Site 1264. Further work disentangling global versus regional productivity patterns is needed in future to explore causal links in greater detail.

## 6 Conclusions

We present a continuous Site-1264-encompassing depth (~316 m) and age (~30 Myr) model that constitutes a reference framework for future palaeoclimatic and palaeoceanographic studies. To achieve this framework, we generated new high-resolution (1-2 cm) XRF records between 17 and 0 Ma at ODP Site 1264 in the Southeastern Atlantic. We used the XRF data to revise the shipboard composite splice, especially in the late Miocene-early Pliocene interval. The new ln(Ca/Fe) records were integrated with previously published Oligocene-early Miocene XRF records and calibrated to shipboard %CaCO$_3$ data to obtain the first continuous Southeastern Atlantic carbonate record spanning the last 30 million years. Because of the variable orbital forcing imprint recorded in the Site 1264 CaCO$_3$ content, we employed three distinct tuning strategies to achieve a 30 Myr astrochronology: I.a) 30-9.7 Ma: CaCO$_3$ content(/benthic δ$^{18}$O) to eccentricity; I.b) 9.7-8.0 Ma: CaCO$_3$ content to E(T); II) 8.0-3.3 Ma: CaCO$_3$ content(/benthic δ$^{18}$O) to ET-P; and III) 3.3-0.0 Ma: benthic δ$^{18}$O to LR04.

The %CaCO$_3$ and CaCO$_3$ MARs were used to investigate carbonate deposition in the Southeastern Atlantic since the Oligocene. We recognise three distinct orbital pacings of the short-term %CaCO$_3$ variability, broadly related to major changes in climate, the cryosphere and/or the carbon cycle: 1) ~110 kyr eccentricity-driven pacing dominates from 30 to ~13 Ma during Oligo-Miocene global warmth; 2) eccentricity-modulated precession-driven pacing appears after the mMCT and prevails from 14-8 Ma; 3) increased obliquity/precession-driven pacing prevails between ~7.7-3.3 Ma, following increased influence of high-latitude processes.

The lowest CaCO$_3$ content (92-94%) occurs between 18.5-14.4 Ma, suggesting increased dissolution and/or decreased carbonate rain at Site 1264, potentially caused by the widespread global warmth associated with the MCO. However, the beginning of the low CaCO$_3$ content at Site 1264 precedes the MCO by ~1.5 Myr, in line with evidence for dissolution-induced %CaCO$_3$ lows in the equatorial Pacific Ocean 1.0-0.5 Myr before the MCO. This may indicate that the global warmth and Antarctic deglaciation across the MCO was preconditioned for up to ~1.5 Myr by a prolonged interval of early Miocene global warmth. The emergence of precession-driving pacing in the Site 1264 CaCO$_3$ content after ~14 Ma suggests that Antarctic ice sheet expansion and global cooling across the MMCT caused regional productivity to become more sensitive to precession forcing and/or signifies the appearance of less corrosive deep-waters at Site 1264 leading to better preservation of precession-driven productivity cycles.

In association with the late Miocene Biogenic Bloom (LMBB), the highest CaCO$_3$ content (95-97.5%) occurs between ~8-4 Ma and the highest CaCO$_3$ MARs (~2-4.5 g/cm$^2$/kyr) are found between ~7.8-3.3 Ma. The onset of elevated CaCO$_3$ MARs (~7.8 Ma) roughly coincides with the shift from eccentricity-precession pacing to pervading obliquity-precession pacing of %CaCO$_3$, which suggests a link between the onset of the LMBB and the increased influence of high-latitude processes, such as enhanced glacial activity and high-latitude cooling. The timing of the LMBB in the Site 1264 MARs agrees well with the onset in the eastern equatorial Pacific (EEP), although the LMBB lasts ~1 Myr longer in the South Atlantic (~3.3 Ma) than in the EEP (~4.4 Ma). Global patterns in the LMBB may be driven by increased nutrient input through increased late Miocene glacial weathering and/or increased weathering associated with Himalayan uplift/intensification of the monsoon. A global

increase in the oceanic nutrient flux may be related to increased dust input following increased continental aridification and enhanced trade winds due to the increased latitudinal temperature gradients that appeared during the late Miocene cooling (7-5.4 Ma). Regional differences in the expression of the LMBB most likely reflect changes in oceanic nutrients distribution driven by regional differences in the extent of the late Miocene cooling, diachrony in the spread of continental aridification and/or changes in oceanic circulation following the late Miocene carbon isotope shift.

**Author Contributions and Competing Interests**

AJD, DL, LL and TW designed the study. AJD, DL, TW, HB, LL, NR, RW, DK, DB and ML contributed to the data collection and analysis. AJD, DL, TW and LL contributed to the stratigraphy and astrochronology. AJD wrote the manuscript with input from all co-authors. The authors declare that they have no conflict of interest.

**Acknowledgements**

This research used samples and data provided by the Ocean Drilling Program (ODP), sponsored by the US National Science Foundation (NSF) and participating countries. This research used data acquired at the XRF Core Scanner Lab at the MARUM – Center for Marine Environmental Sciences, University of Bremen, Germany. We especially thank Ursula Röhl and Vera Lukies (MARUM) for assistance with XRF core scanning and Alex Wülbers, Walter Hale and Holger Kuhlmann (IODP Bremen Core Repository) for core handling, and Tim van Peer for valuable discussions. Funding for this research was provided by the Deutsche Forschungsgemeinschaft (DFG, German Research Foundation) to TW and AJD (Project number 242225091, 408101468). AJD and DL were postdoctoral researchers and HP was the principal investigator in ERC Consolidator grant "EARTHSEQUENCING" (grant agreement 617462). AJD is currently funded by the European Union's Horizon 2020 research and innovation programme under the Marie Sklodowska-Curie grant agreement No 796220. ML was funded by NSF grant OCE-1656960. LL's part of the research was carried out under the program of the Netherlands Earth System Science Centre (NESSC), financially supported by the Dutch Ministry of Education, Culture and Science (OCW).

**Data availability**

All data is archived on the open access database PANGAEA (https://doi.pangaea.de/10.1594/PANGAEA.919489). Further supplementary information is also available with the online version of this manuscript on the *Climate of the Past* website. A list of Supplemental Tables and Figures is provided here:

*Supplementary tables:*

1) Site 1264 XRF and Site 1264/1265 $CaCO_3$ data, including sedimentation rates and all MARs. The uncalibrated 1264 XRF datasets and full Site 1265 $CaCO_3$ data are also included.

2) Offsets/affine tables for 1264

3) Splice tie/interval tables for 1264

4) Mapping tables for 1264

5) 1264-1265 correlation to accommodate splice revisions

6) Selected (i.e. high-quality) bio- and magnetostratigraphic events for Site 1264

7) New astrochronology (with sedimentation rates) for Site 1264

*Supplementary figures:*

1) XRF intercalibration of the four measurement campaigns

2) Downcore intercalibrated XRF data from Site 1264, including ln(Ca/Fe), Si, Fe, K, Ti, and Mn

3) Splice revision panels for entire interval showing revisions.

4) Revisions to the offsplice mapping pairs of Core 1264B-29H

5) Generation of the composite core image of ODP Sites 1264 and 1265.

6) Calibration of ln(Ca/Fe) to shipboard $\%CaCO_3$

7) Calculation of bulk and $CaCO_3$ MARs

8) Polynomial fit through the selected (i.e. high-quality) bio- and magnetostratigraphic events for Site 1264

9) Spectral analysis of $\%CaCO_3$ on the polynomial age model

10) Oversized panels showing depth to age tie points and age model generation

11) Antiphase relationship between benthic $\delta^{18}O$ and $\%CaCO_3$

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

**Figure 1: (A)** Site Overview showing location of Site 1264 and Site 1265 on the Angola Basin side of Walvis Ridge, as well as **(B)** the differences between core box photos and line-scan images compiled with CODD.

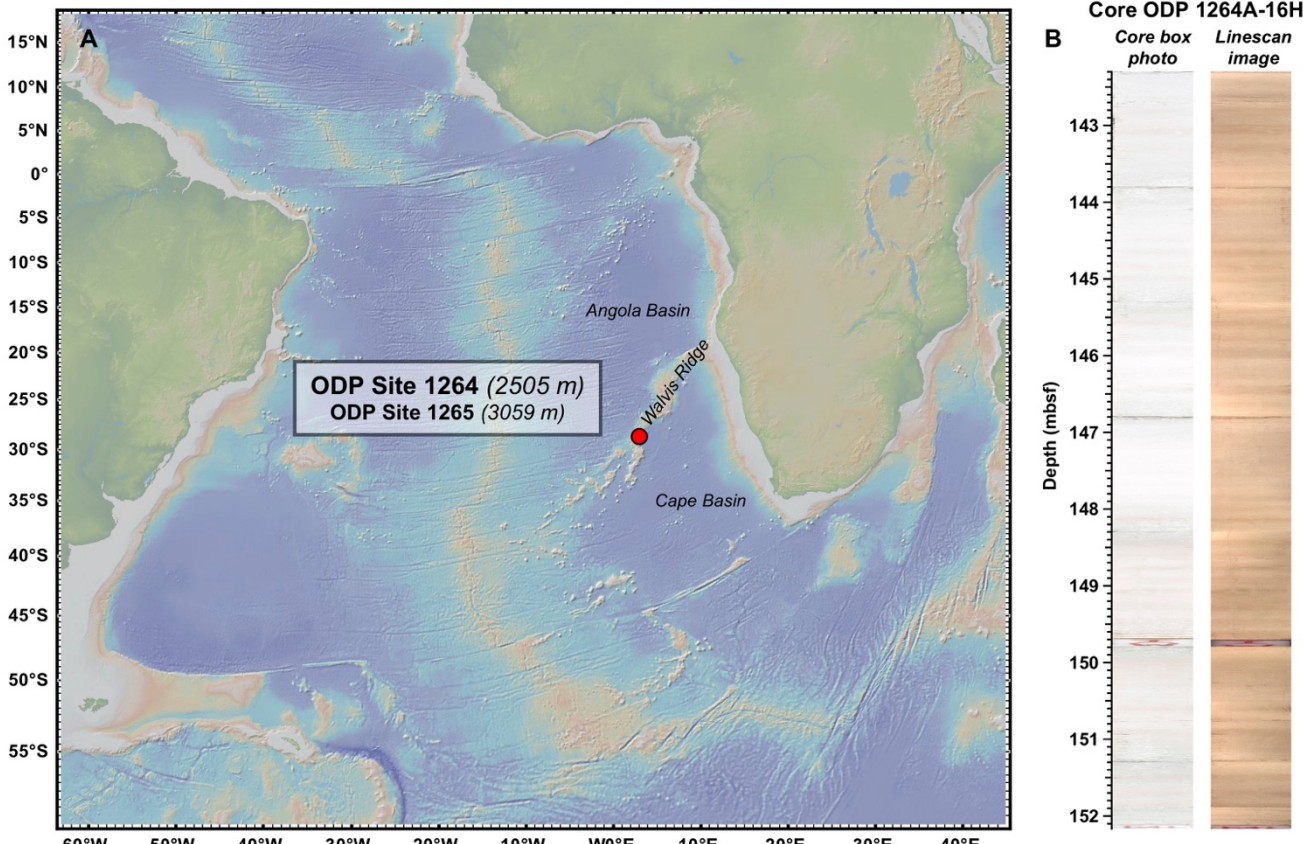

**Figure 2: Overview of main splice change between the shipboard splice (A) and the revised splice presented in this study (B). The interval between arrows on the splice was revised based on the ln(Ca/Fe) data. The shipboard magnetic susceptibility (MS) is too low amplitude in the late Miocene in particular to robustly revise the splice, whereas ln(Ca/Fe) data showed the sedimentary variability well. In certain high CaCO$_3$ intervals Ba counts were also used to revise the composite splice (Supplementary Figure 3). The individual holes are shown for the shipboard splice only, with splice intervals shown between consecutive turquoise (top) and purple (bottom) vertical lines.**

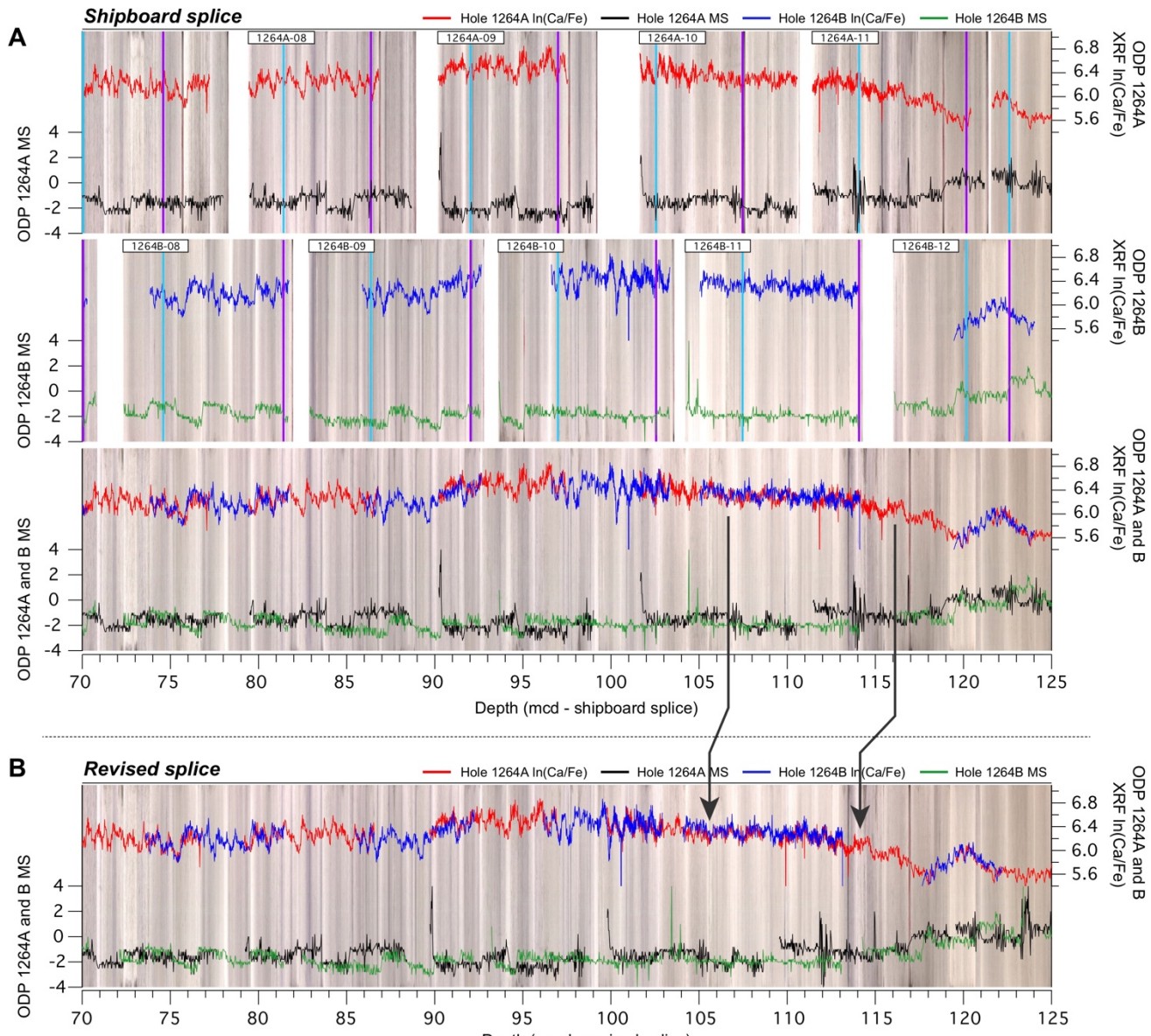

**Figure 3: On the new revised composite depth (rmcd) (A) Site 1264 XRF Si (green) and K (teal) intensities, (B) %CaCO₃ data derived from ln(Ca/Fe) for Sites 1264 (dark red) and 1265 (black), (C) bulk and CaCO₃ MARs for Sites 1264 (bulk = dark blue; CaCO₃ = light blue) and 1265 (bulk = black; CaCO₃ = dark grey), (D) sedimentation rates in m/Myr for Sites 1264 (light grey) and 1265 (black), and the combined composite core photo for Sites 1264 and 1265 compiled using line scan (E) and core box photo images (F). The depth-domain wavelet spectra are shown for the %CaCO₃ data after it was detrended to remove all cycles greater than 2 m (G) or greater than 40 m (H). The periods are highlighted in m. The wavelets were generated using the code from Torrence and Compo (1998) and Grinsted et al. (2004). The approximate stratigraphic location of the MCO and the LMBB are highlighted by shaded grey areas.**

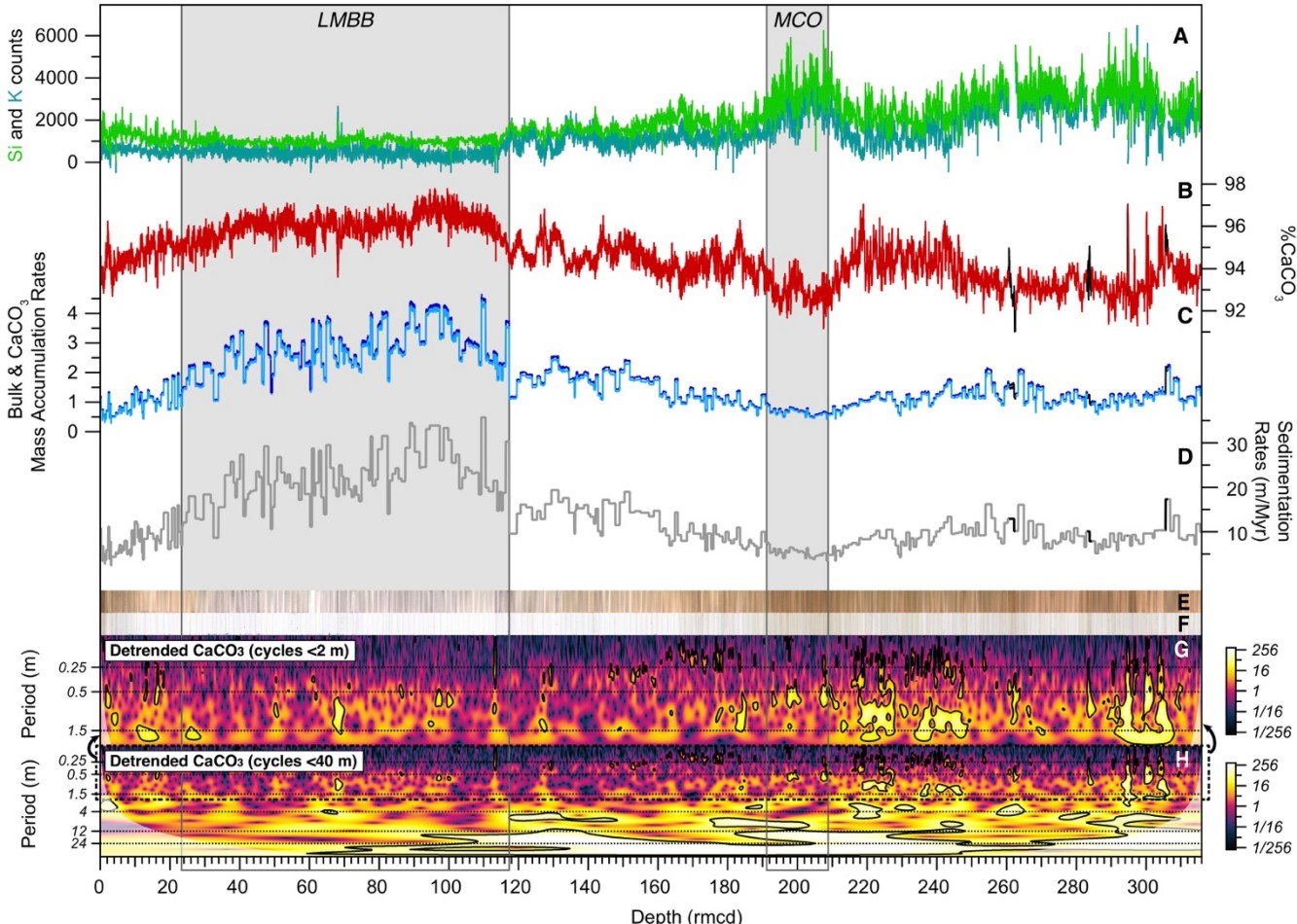

**Figure 4: Overview of the new astrochronology for the last 30 Myr. The four panels show the different tuning strategies employed. I.a) 30-9.7 Ma: CaCO₃(/benthic δ¹⁸O) to eccentricity; I.b) 9.7-8.0 Ma: CaCO₃ to E(T); II) 8.0-3.3 Ma: CaCO₃(/benthic δ¹⁸O) to ET-P; and III) 3.3-0.0 Ma: benthic δ¹⁸O to LR04. The composite core photo compiled from line scan images is used here as it highlights the sedimentological cyclicity best. Zoomed in figures showing the exact depth-age tie points are shown in Supplementary Information Figure 10.**

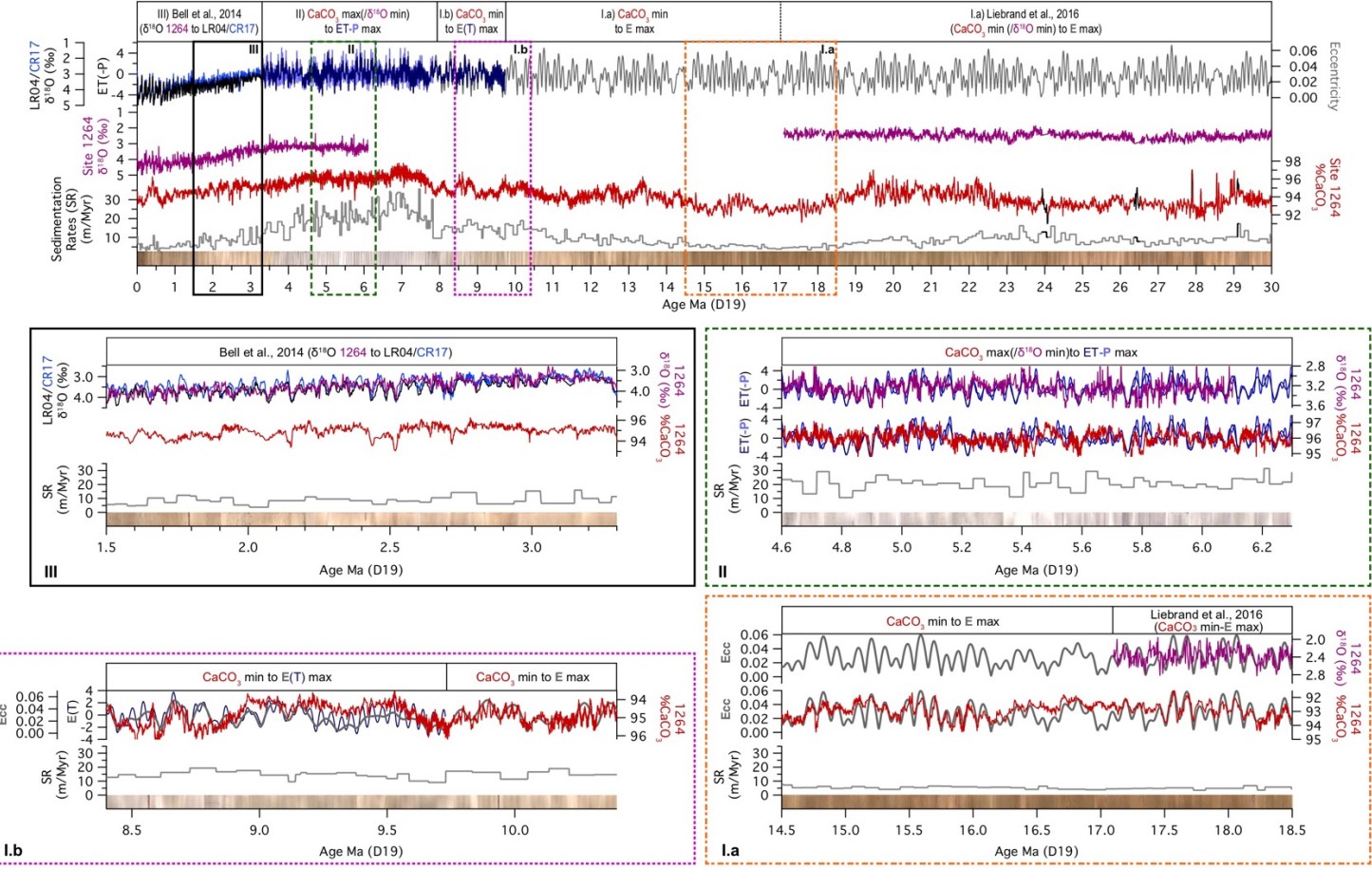

**Figure 5: New data from Sites 1264/1265 on the new astrochronology: A)** Site 1264 benthic foraminiferal δ[18]O (Bell et al., 2014; Liebrand et al., 2016) and the benthic δ[18]O Megasplice (De Vleeschouwer et al., 2017); **B)** Site 1264 Si intensity (counts); **C)** left axis: %>63µm coarse fraction (%CF) (Keating-Bitonti and Peters, 2019) and right axis: XRF-derived CaCO₃ data from Sites 1264 (dark red) and 1265 (black); **D)** bulk and CaCO₃ MARs for Sites 1264 (dark and light blue, respectively) and 1265 (black and grey, respectively); **E)** Eccentricity and **F)** obliquity solutions (Laskar et al., 2004); **G)** line scan and **H)** core box photo Site 1264/1265 composite core photos; **I)** wavelet spectra in the time domain of the CaCO₃ data detrended to remove cycles over 200 kyr; **J)** wavelet spectra in the time domain of the CaCO₃ data detrended to remove cycles over 4 Myr (Torrence and Compo, 1998; Grinsted et al., 2004). The MCO, mMCT and the LMBB are annotated.

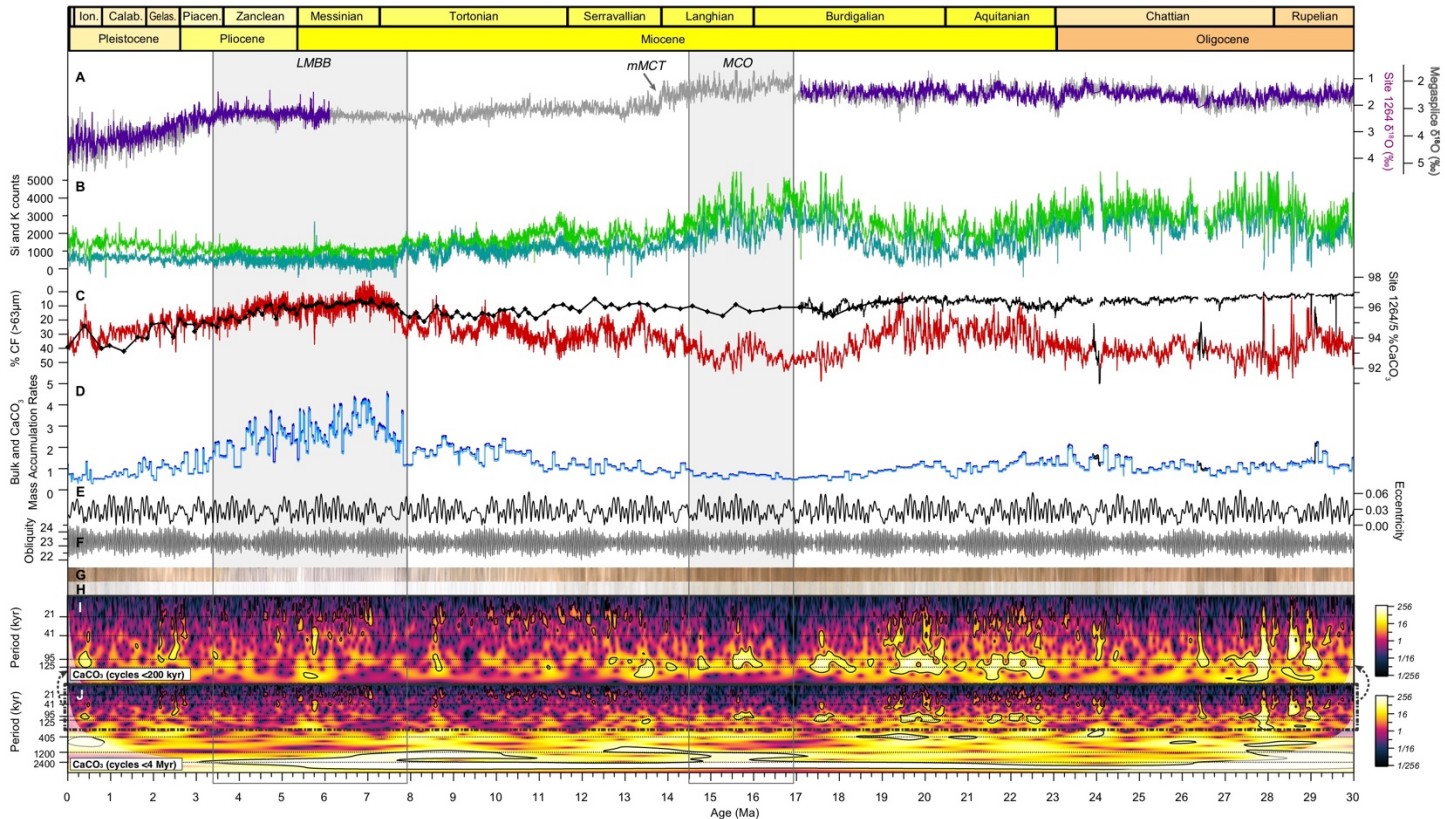

**Figure 6: Zoomed in panels highlighting the three distinctly different orbital controls on Southeast Atlantic CaCO₃ deposition. A) Example of strong eccentricity (E) pacing present between 30 and 13 Ma; B) Example of the prevalent eccentricity-modulated precession pacing present between 14 and 8 Ma; C) Example of the pervasive obliquity forcing present between 8 and ~3.3 Ma. An example of stronger obliquity appearing in a 2.4 Myr eccentricity minimum, when eccentricity-modulated precession is muted, is also shown in B. CaCO₃ minima correlate with eccentricity maxima between 30 and 8 Ma (A and B). Between 8 and 0 Ma, CaCO₃ maxima correlate with obliquity maxima (C).**

**Figure 7: Mid-late Miocene Sites 1264/1265 data on the new astrochronology:** A) BAYSPAR TEX$_{86}$ SSTs from Site 608 (blue dots = 50$^{th}$ percentile; medium blue = 65% CL; light blue = 95% CL; Super et al., 2018) B) benthic $\delta^{18}$O Megasplice (De Vleeschouwer et al., 2017); C) XRF-derived CaCO$_3$ data from Site 1264; D) bulk and CaCO$_3$ MARs for Sites 1264 (dark and light blue, respectively); E) Site 1264 line scan and F) core box composite core photos; G) wavelet spectra in the time domain of the CaCO$_3$ data detrended to remove cycles >200 kyr (Torrence and Compo, 1998; Grinsted et al., 2004). The approximate location of the MCO and the mMCT are also shown.

**Figure 8: Late Miocene to present data on the new astrochronology: A) left axis: XRF-derived CaCO₃ data from Sites 1264 (dark red) and right axis: >63μm coarse fraction (%CF) (Keating-Bitonti and Peters, 2019); B) normalised Site 1264 MARs (this study) and a normalised eastern equatorial Pacific (EEP) stack comprising data from ODP Sites 848, 849, 850 and 851, and IODP Sites U1335, U1337 and U1338; (Lyle et al., 2019); C) Site 1264 CaCO₃ MARs; D) MARS from EEP ODP Sites 848, 849, 850 and 851, and IODP Sites U1335 and U1338 (Lyle et al., 2019).  Site U1337 was not included as it was partly affected by winnowing (see Lyle et al., 2019 for details).**

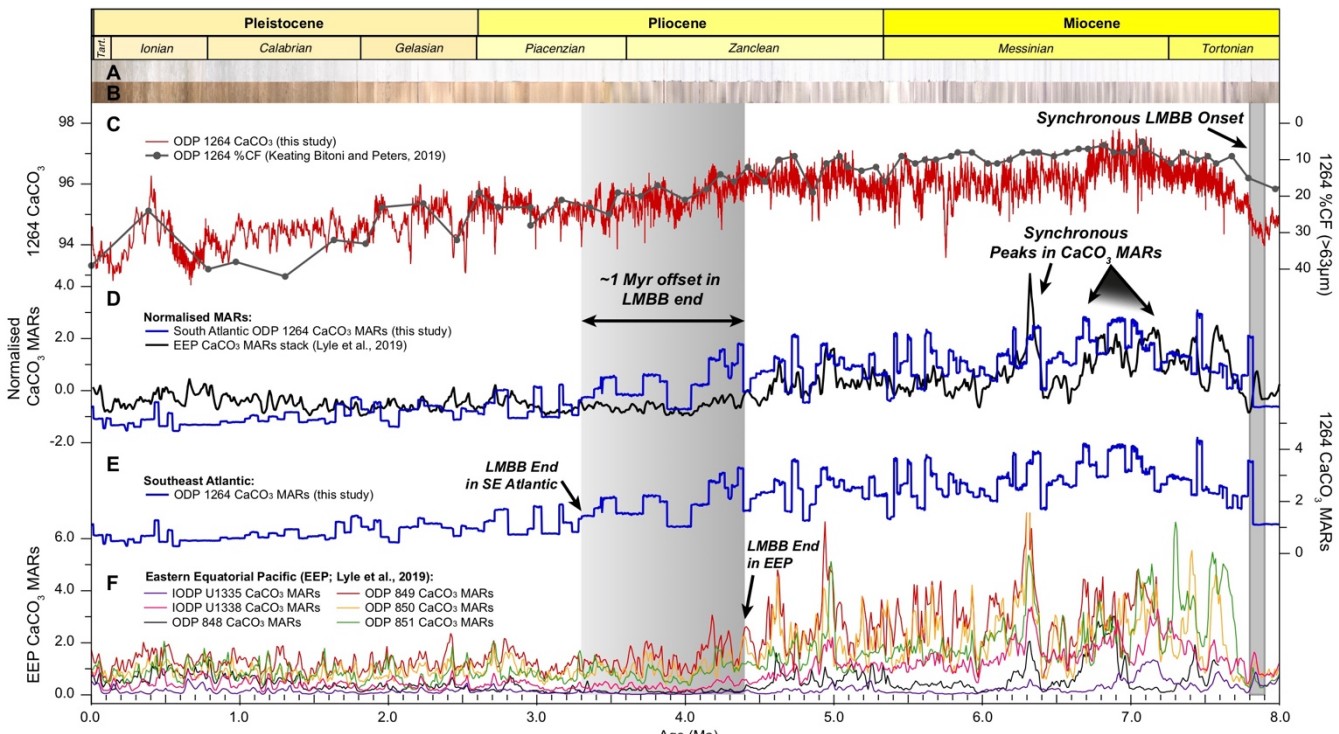