# Peer review of "Climate, cryosphere and carbon cycle controls on Southeast Atlantic orbital-scale carbonate deposition since the Oligocene (30-0 Ma)"

_Climate of the Past, 2020_

## Referee Comment (RC1) · Anonymous Referee #1 · 3 Oct 2020

Drury et al., present a new XRF (Ca/Fe) record that extends the previously published study of Liebrand et al., 2016 from early Miocene into Pleistocene. CaCO3 content estimated from this new dataset provides the first composite record in the South Atlantic with a continuous astronomical chronology for the last 30 Ma. Through wavelet analysis, Drury et al., find that the variability and dominant cyclicity in %CaCO3 content have evolved over time. Overall, 3 distinct stages are recognized: from 30 to 8 Ma, eccentricity paced cyclicity dominates %carbonate variability. After 8 Ma, obliquity and precession become more prevalent while eccentricity imprint is reduced. In the last 3 Ma, both precession and obliquity become hard to observe and the age model relies on previously published benthic d18O (Bell et al., 2014). The manuscript is well written,

and the dataset has the potential to make a great contribution to the community and serve as a framework for future palaeoclimatic and palaeoceanographic studies. I thus suggest acceptance of the manuscript with some minor revision.

My primary concern about the studied site is the potential complications of winnowing. Sites of Leg 208 were drilled to provide a depth transect in the South Atlantic to monitor changes in ocean chemistry as a function of time and depth. However, a submarine edifice such as the Walvis Ridge also forms a major obstacle to the flow of deep and intermediate waters. Sediments deposited on such a topographic high can be highly winnowed due to intensified flow of waters around and over the ridge. Shackleton et al. (1984) have also studied the accumulation rate of fine fraction along the depth transect of Leg 74 and concluded that winnowing has removed fine-grained material from topographic highs and deposited them on the flanks and in the basins.

I also take a look at the biostratigraphy of Site 1264 in the initial report. I find that 1) the highest occurrence of D. tamalis is ~18 mcd and 30 mcd at Site 1264 and 1266, respectively; 2) the highest occurrence of D. pentaradiatus is ~16 mcd and 26 mcd at Site 1264 and 1266, respectively; 3) the highest occurrence of D. brouweri is ~11 mcd and ~20 mcd at Site 1264 and 1266, respectively; the list can go on. The point here is that the deeper Site 1266 (3800 meters) has much higher carbonate accumulation rates (~ doubled) than the shallower Site 1264. This difference cannot be due to productivity and is unlikely due to dissolution. The most possible interpretation therefore is that strong winnowing has significantly affected the carbonation accumulation of Site 1264.

I do not rule out the possibility that changes in carbonate accumulation and %CF can be partially explained by changes in primary productivity at Site 1264. However, can authors provide some other evidence to support their interpretation of a change in primary productivity? Alternatively, if winnowing is important at Site 1264, could it compromise the %CaCO3 records and how it might affect the spectrum properties? For instance, could winnowing explain obscure cyclicity in the last 3 Ma at Site 1264? I think
these are open questions but the authors should be aware of the potential complication at this location.

ref: Shackleton, N. J. (1984). Accumulation rates in Leg 74 sediments. Initial Reports of the Deep Sea Drilling Project, 621-644.

Other comments: Figure 2 caption suggests that the black and green records are magnetic susceptibility. The label of y axis, however, is XRF (Ca/Fe).

Page 7, Line 10: a typo?

P14, the authors relate the recovery of %CaCO3 ~14.5 Ma to changes in dissolution. The study of Kender et al., 2014 (benthic foram B/Ca) can be helpful.

ref. Kender, S., Yu, J., & Peck, V. L. (2014). Deep ocean carbonate ion increase during mid Miocene CO 2 decline. Scientific reports, 4, 4187.

———————————————

---

## Referee Comment (RC2) · Anonymous Referee #2 · 13 Oct 2020

The authors present a revised composite depth scale and orbitally-tuned age model for Walvis Ridge ODP Site 1264 (S. Atlantic), based on XRF core-scanning ln(Ca/Fe) data which has been calibrated to shipboard %CaCO3. This is a substantial undertaking and will be extremely useful for further studies. The methods are described in detail and sufficiently illustrated, hence the new splice and age model appear to be robust. Overall the paper is well-written and organised, and I'm happy to see it published close to its present form.

Key methodological points that the authors address are, first, correcting the XRF results from over several years, and from different instruments/settings (Supplementary fig.

1 – it would be better if there was a bit more overlap of 2018 data with 2011 data, but access to rescanning some sections may not be possible at present? If not, I'm happy with the correction as it is), and second, calibrating the scanning Ca data to shipboard %CaCO3. The latter correlation shows some scatter, but outliers are ignored (I presume based on visual identification?) and the inferred %CaCO3 is within 2% at 1SD. This error isn't taken into account when discussing the %CaCO3 time-series nor when calculating CaCO3 MARs, and I'm not sure how much difference it will make to the conclusions because carbonate content is so high ($\sim$92-97%). But given that the discussion, especially about the Biogenic Bloom, hinges on accuracy of the CaCO3 MARs, which in turn hinge on accuracy of %CaCO3 according to the authors, then some mention of error on the inferred %CaCO3 & its MARs is perhaps warranted. Finally, I agree with the 3 different tuning strategies for the different intervals – this appears to be justified.

Regarding the wavelet analysis, I'm not entirely convinced I can see the cyclicity that the authors see, particularly the comment in 4.1.3, line 11. This may in part be due to the small reproduction of the wavelet figures, but I also think there's some ambivalence here. A related issue is how much can be inferred from the wavelet analyses without bias, given the orbital tuning methodology?

Section 5.1. The authors link orbital cycles at site 1264 to Antarctic ice-sheet variability and NADW, but there's no explanation of how this process-link is made. Maybe elaborate or be more speculative.

Minor edits. One of the authors with excellent English should proof-read the manuscript for grammar/spellings as there are several incorrect verb formats. I started noting them in section 3.2 (line 12: occur should read occurs or occurred, etc). Section 4.1, line 10: "four" intervals. Fig. 2: relabel 4 axes for mag sus. Fig. 3: (b) there are some lower spikes in %CaCO3 that look like outliers/cracks. . .??? (c) units? Also, I can't see the black & grey lines. . .is it because they exactly underlie the blue lines (seems a bit odd if they are identical)? Fig. 5: is it possible to show tie points? Fig. 6: (h) periods are

same as in Fig. 4c, but now we are in the age domain so I would expect ky periods on the left-hand axes.

---

## Author Comment (AC1) · 17 Jan 2021

**Anonymous Referee #1 (R1)**

We would like to take this opportunity to thank R1 for taking time to review our manuscript and for providing such constructive feedback that will help improve the overall manuscript. Please see our responses to this feedback below.

Drury et al., present a new XRF (Ca/Fe) record that extends the previously published study of Liebrand et al., 2016 from early Miocene into Pleistocene. CaCO3 content estimated from this new dataset provides the first composite record in the South Atlantic with a continuous astronomical chronology for the last 30 Ma. Through wavelet analysis, Drury et al., find that the variability and dominant cyclicity in %CaCO3 content have evolved over time. Overall, 3 distinct stages are recognized: from 30 to 8 Ma, eccentricity paced cyclicity dominates %carbonate variability. After 8 Ma, obliquity and precession become more prevalent while eccentricity imprint is reduced. In the last 3 Ma, both precession and obliquity become hard to observe and the age model relies on previously published benthic d18O (Bell et al., 2014). The manuscript is well written, and the dataset has the potential to make a great contribution to the community and serve as a framework for future palaeoclimatic and palaeoceanographic studies. I thus suggest acceptance of the manuscript with some minor revision.

We are pleased to hear that R1 values the progress we have made regarding the stratigraphy and understanding of the dominant cyclicities in carbonate content in the Angola basin for the past 30 Ma, and our dissemination of these results in the paper.

My primary concern about the studied site is the potential complications of winnowing. Sites of Leg 208 were drilled to provide a depth transect in the South Atlantic to monitor changes in ocean chemistry as a function of time and depth. However, a submarine edifice such as the Walvis Ridge also forms a major obstacle to the flow of deep and intermediate waters. Sediments deposited on such a topographic high can be highly winnowed due to intensified flow of waters around and over the ridge. Shackleton et al. (1984) have also studied the accumulation rate of fine fraction along the depth transect of Leg 74 and concluded that winnowing has removed fine-grained material from topographic highs and deposited them on the flanks and in the basins.

We agree with R1 that winnowing could have affected accumulation rates at Site 1264 and we will take this mechanism into account in the revised manuscript. However, we also note that Site 1264 is not positioned on the shallowest parts of the Walvis Ridge bathymetry (which are less than a km deep) and is situated on a very gentle slope. With palaeo-water depths between 2 and 2.5 km, the site was situated above the lysocline and CCD.

Unfortunately, we currently do not have independent constraints on winnowing throughout the entire 30 Myr interval (sample processing is still in progress), but if we interpret fine fraction weights for the interval between 30 and17 Ma (Liebrand et al. 2016, their Fig. 2) as a proxy for winnowing, this would suggest that winnowing is modest during the "mid" Oligocene, increasing during late Oligocene warming and relatively high across the Oligocene-Miocene Transition. During the early Miocene (post OMT, pre-mid Miocene) winnowing is comparable to late Oligocene values, and increase toward the more condensed middle Miocene part of the Site 1264 record. We will expand the results to include these observations and will take this mechanism into account in the discussion.

I also take a look at the biostratigraphy of Site 1264 in the initial report. I find that 1) the highest occurrence of D. tamalis is ~18 mcd and 30 mcd at Site 1264 and 1266, respectively; 2) the highest occurrence of D. pentaradiatus is ~16 mcd and 26 mcd at Site 1264 and 1266, respectively; 3) the highest occurrence of D. brouweri is ~11 mcd and ~20 mcd at Site 1264 and 1266, respectively; the list can go on. The point here is that the deeper Site 1266 (3800 meters) has much higher carbonate accumulation rates (~ doubled) than the shallower Site 1264. This difference cannot be due to productivity and is unlikely due to dissolution. The most possible interpretation therefore is that strong winnowing has significantly affected the carbonation accumulation of Site 1264.

We thank R1 for pointing out these differences between the 1264 and 1266 biostratigraphy in the upper succession. The discrepancy certainly indicates that the lower accumulation rates in the upper interval (certainly the upper 30 or 40 mcd) cannot/is unlikely to arise from productivity and/or dissolution. However, for the deeper intervals, the sedimentation rates increase at 1264 relative to 1266, which decreases the discrepancy between older biostratigraphic events. Certainly, for the late Miocene interval, the 1264 sedimentation rates are similar or higher than 1266, which would mean productivity remains a valid mechanism to explain these differences. Nonetheless, we agree with R1 that winnowing needs to be considered and that there are intervals where winnowing likely significantly influenced carbonate accumulation at Site 1264. We will therefore add a discussion of this biostratigraphic argument to the paper, especially in the context of what it reveals about how winnowing may act as an additional mechanism controlling CaCO3 deposition at 1264.

I do not rule out the possibility that changes in carbonate accumulation and %CF can be partially explained by changes in primary productivity at Site 1264. However, can authors provide some other evidence to support their interpretation of a change in primary productivity? Alternatively, if winnowing is important at Site 1264, could it compromise the %CaCO3 records and how it might affect the spectrum properties? For instance, could winnowing explain obscure cyclicity in the last 3 Ma at Site 1264? I think these are open questions but the authors should be aware of the potential complication at this location.

Our observation linking the inverse relationship between carbonate content/mass accumulation rates and %CF to productivity were only associated with the biogenic bloom interval in the late Miocene. We will make sure this is clearer in Section 5.3. As commented above, the sedimentation rates at 1264 increase relative to those at 1266 during the late Miocene and are either similar or higher at 1264 compared to 1266. We will certainly discuss the implications of the relative occurrence of biostratigraphic events between 1264 and 1266 across the 30 myr interval, but we think winnowing is less likely to have influenced carbonate deposition during the biogenic bloom interval compared to productivity. Although we do not have direct evidence of productivity during this time, we think increased productivity remains a valid interpretation considering the greatly increased sedimentation rates at Site 1264 and the presence of similar increases in carbonate deposition linked to increased productivity at other locations around the globe.
The effect of low sedimentation rates on spectral power at Site 1264 is well explained and explored for the Oligocene and early Miocene in Liebrand et al., 2016. Unfortunately, they did not suggest winnowing as a mechanism, but we agree with R1 that this topic should be considered and will expand our discussion as such. We will especially also consider how winnowing may explain the low carbonate values and lack of clear cyclicity in the last 3 Ma.

ref: Shackleton, N. J. (1984). Accumulation rates in Leg 74 sediments. Initial Reports of the Deep Sea Drilling Project, 621-644.

We will include this reference in the discussion about how winnowing may affect carbonate deposition at Site 1264.

Other comments:
Figure 2 caption suggests that the black and green records are magnetic susceptibility. The label of y axis, however, is XRF (Ca/Fe).

Thank you for spotting this error. The y-axis should be labelled shipboard magnetic susceptibility and we will correct this.

Page 7, Line 10: a typo?

Thank you, we have corrected "for" to "four".

P14, the authors relate the recovery of %CaCO3 ~14.5 Ma to changes in dissolution. The study of Kender et al., 2014 (benthic foram B/Ca) can be helpful.

ref. Kender, S., Yu, J., & Peck, V. L. (2014). Deep ocean carbonate ion increase during mid Miocene CO 2 decline. Scientific reports, 4, 4187.

Thank you, we were not aware of this study, it certainly supports that the appearance of precession ~13-14 Ma may in part be due to less seafloor dissolution. We will take their results into account in this discussion, as well as considering the potential influence that winnowing may have.

---

## Author Comment (AC2) · 17 Jan 2021

**Anonymous Referee #2 (R2)**

We would like to take this opportunity to thank R2 for taking time to review our manuscript and for providing such constructive feedback that will help improve the overall manuscript. Please see our responses to this feedback below.

The authors present a revised composite depth scale and orbitally-tuned age model for Walvis Ridge ODP Site 1264 (S. Atlantic), based on XRF core-scanning ln(Ca/Fe) data which has been calibrated to shipboard %CaCO3. This is a substantial undertaking and will be extremely useful for further studies. The methods are described in detail and sufficiently illustrated, hence the new splice and age model appear to be robust. Overall the paper is well-written and organised, and I'm happy to see it published close to its present form.

We are very pleased to hear that R2 values the progress represented by our revised depth and age model at Site 1264, and recognises the value for future research in this region.

Key methodological points that the authors address are, first, correcting the XRF results from over several years, and from different instruments/settings (Supplementary fig. 1 – it would be better if there was a bit more overlap of 2018 data with 2011 data, but access to rescanning some sections may not be possible at present? If not, I'm happy with the correction as it is), and second, calibrating the scanning Ca data to shipboard %CaCO3. The latter correlation shows some scatter, but outliers are ignored (I presume based on visual identification?) and the inferred %CaCO3 is within 2% at 1SD. This error isn't taken into account when discussing the %CaCO3 time-series nor when calculating CaCO3 MARs, and I'm not sure how much difference it will make to the conclusions because carbonate content is so high (~92-97%). But given that the discussion, especially about the Biogenic Bloom, hinges on accuracy of the CaCO3 MARs, which in turn hinge on accuracy of %CaCO3 according to the authors, then some mention of error on the inferred %CaCO3 & its MARs is perhaps warranted. Finally, I agree with the 3 different tuning strategies for the different intervals – this appears to be justified.

We are very happy that R2 agrees with the different tuning strategies we applied.
Unfortunately, it is not possible to increase the overlap between the 2018 and 2011 data, as this would involve rescanning material from all four measurement campaigns to generate a new 2021 calibration. We chose the overlap between the 2018 and 2011 data, as it was the same overlap scanned between the 2017 and 2011 and 2017 data. This provided the opportunity to compare data across three measurement campaigns, albeit over a short interval.
We agree with R2 that we should explain the %CaCO3 calibration process better and that we should propagate the error associated with it. We will include a brief explanation of why certain outliers were ignored, propagate the error for both the %CaCO3 and the MARs and discuss how this may influence our interpretations. However, the error is mostly relevant to the absolute carbonate content, rather than the cyclicity or the trends (e.g., identification of the biogenic bloom), which are both visible in the raw ln(Ca/Fe) timeseries. The calibration error and/or potential later changes to the calibration won't alter the underlying trends or cyclicities in the raw data. Nonetheless, we agree with R2 that the error should be mentioned and will revise the manuscript to reflect this.

Regarding the wavelet analysis, I'm not entirely convinced I can see the cyclicity that the authors see, particularly the comment in 4.1.3, line 11. This may in part be due to the small reproduction of the wavelet figures, but I also think there's some ambivalence here. A related issue is how much can be inferred from the wavelet analyses without bias, given the orbital tuning methodology?

We appreciate that the data and wavelets presented in Figure 4 are small, especially as we present the entire record in one figure. We plan to revise this figure to be comparable to those in Figure 6, but then in the depth domain. We will also change the text here to convey our main message more clearly, which is that the carbonate data itself shows clear cyclicity between 115 and 35 rmcd, but that the amplitude of this variability is reduced compared to the deeper interval. We appreciate that this is clearer in the $CaCO_3$ data itself, rather than the wavelets, which show this less well. We will also consider whether we can include a magnified insert of this interval to show the cyclicity or provide supplementary figures where expanded depth intervals can be shown across multiple panels.
The observations about the potential origin of the depth cyclicity are based on the data and wavelets in the depth domain compared to the data and wavelets in the age domain using the initial bio-magnetostratigraphic age model (supplementary figure 8). Both the depth and initial age model are independent of the astronomical tuning, which is only applied to the data in Figure 6, so the cyclicity observed and discussed in 4.1.3. is likewise independent from the orbital tuning methodology.

Section 5.1. The authors link orbital cycles at site 1264 to Antarctic ice-sheet variability and NADW, but there's no explanation of how this process-link is made. Maybe elaborate or be more speculative.

In the case of the early-mid Miocene and late Miocene, sections 5.2 and 5.3 are intended to provide greater detail about the observed carbonate trends and cyclicity, and what may drive these.
We mention that 110 kyr-driven cycles in benthic $\delta^{18}O$ and carbonate have been linked to Antarctic ice sheet variability in the 30-17 Ma interval by Liebrand et al., 2016/2017 and infer that a similar relationship may exist in the 17-13, or even 17-8 Ma interval; however, we do not have benthic stable isotope data to confirm that for those intervals. We will expand the discussion to discuss potential process-links. However, these will remain speculative based on carbonate content alone, so we will make sure the discussion reflects that tone.
We mentioned that benthic $\delta^{13}C$ records discussed in Bell et al., 2014/2015 indicate that Site 1264 was heavily influenced by NADW for the 5-0 Ma interval, but it wasn't our intention to specifically link the younger (late Miocene-recent) variability to the presence of NADW at 1264. We will adapt the discussion to reflect this.

Minor edits.

One of the authors with excellent English should proof-read the manuscript for grammar/spellings as there are several incorrect verb formats. I started noting them in section 3.2 (line 12: occur should read occurs or occurred, etc).

Thank you, we will make sure these issues are resolved during revision.

Section 4.1, line 10: "four" intervals.

Thank you, we will correct this.

Fig. 2: relabel 4 axes for mag sus.

Thank you for spotting this error, we will correct this.

Fig. 3: (b) there are some lower spikes in %CaCO3 that look like outliers/cracks: : :???

These lower spikes likely originate from abnomolously Fe counts in the ln(Ca/Fe) ratio, and should have been removed as outliers. We will do so and discuss the treatment of outliers in the methods.

Fig. 3: (c) units? Also, I can't see the black & grey lines: : :is it because they exactly underlie the blue lines (seems a bit odd if they are identical)?

We will add units and revise the caption text to be clearer. This panel contains two datasets: the bulk MARs and the carbonate MARs. The colours are used to denote both the type of data (bulk vs carbonate MAR) and the site at which they originated (1264 vs 1265). The dark and light blue were used to respectively denote bulk and carbonate MAR data from Site 1264. The Site 1264 data accounts for most of what is shown, except for 4 short intervals in the deepest section, where 1265 was used (this correlation was explained in detail by Liebrand et al., 2016). The black and grey colour was used to show the short intervals of bulk (black) and carbonate (grey) MARs from Site 1265. But in both the dark vs light blue and black vs grey, it is hard to see both records, because the bulk and carbonate MARs are near identical as carbonate is the main sedimentary component. We nonetheless wanted to show both datasets, and highlight which data originated from which site. However, we appreciate this is not clear from the current caption and figure, so will alter one or both to improve the clarity.

Fig. 5: is it possible to show tie points?

We are reluctant to include these in the main figure, as there are 368 ties in total and this would completely overcrowd the main panel. We are also concerned that including them in each sub panel could overcrowd the figure. We hoped to show the correlation by placing the data used to correlate over the target curve. Also, as this figure is presented in age only, the ties would not directly show how the data in depth relates to the tuning target in age. However, we will consider including expanded versions of this figure in the supplementary information, showing the data on depth with the tie points to the data on age.

Fig. 6: (h) periods are same as in Fig. 4c, but now we are in the age domain so I would expect ky periods on the left-hand axes.

The periods given for the wavelets are for kyr in Fig 6 and m in Fig 4c. We will change the axes to make this clear.

---

## Author Response (AR1)

**Overview of changes to Drury et al., "Climate, cryosphere and carbon cycle controls on Southeast Atlantic orbital-scale carbonate deposition since the Oligocene (30-0 Ma)"**

Dear Luc Beaufort,

We have completed the requested revisions to our manuscript, as outlined in our author comments in response to the two reviewer comments during the interactive discussion.

The revisions included carefully revisiting the manuscript language, and as such there are numerous small revisions to improve the overall clarity of the manuscript. We have summarised the main revisions to the scientific content here. We have also provided a full list of line-by-line insertions and deletions at the end of this document, and a word track changes file.

The original outcome of the manuscript remains largely unchanged, but we have clarified a few methodological reviewer questions, and expanded our discussed to incorporate some insightful suggestions from the reviewers.

Best wishes,
Anna Joy Drury
Submitted on behalf of all co-authors.

***Summary of main changes:***
1) **Overall manuscript clarity:**
   We have thoroughly proofread through the manuscript to correct previous issues with grammar and spelling. We have also made several revisions to improve the overall clarity of the manuscript. Several relevant publications came out since we submitted the original manuscript (Westerhold et al., 2020, Science; De Vleeschouwer et al., 2020, Nature Communications; Tanner et al., 2020, Paleoceanography and Paleoclimatology), and where relevant, we have included these in our discussion.

2) **Carbonate calibration, calibration uncertainty and treatment of outliers:**
   We found a small error in our initial calibration, so we have corrected this. This has resulted in a small change to the calibration; however, this only changes our absolute $CaCO_3$ by less than 0.07% and does not affect our interpretations.

   Following a helpful suggestion from the reviewer, we also clarified our outlier treatment process and discussed the uncertainty associated with the calibration and the MARs. The calibration uncertainty is ±2.2% at $2\sigma$. This uncertainty only pertains to the absolute %$CaCO_3$ values. The trends and cyclicity we observe in the calibrated $CaCO_3$ data are independent of this uncertainty, as these patterns are present in the raw $\ln(Ca/Fe)$ timeseries. Our interpretations are therefore not affected by this uncertainty.

3) **Improved presentation of the age model and cyclicity observed in the $CaCO_3$ data:**
   The reviewers raised concerns that they could not see the cyclicity we were referring to. We have now provided better wavelet figures in the main text and the manuscript, which we feel

highlight the cyclicity better. We have also added a new figure (new figure 6), where we highlight examples of the three main cyclicities discussed in the manuscript.

**4) Expanded discussion to consider winnowing and the processes that may drive the cyclicity we observe in our CaCO₃ data:**

We have strengthened our discussion to address reviewers' concerns that to our discussion did not sufficiently consider winnowing or the processes that might drive Site 1264 carbonate.

We now introduce sedimentary processes like winnowing and dilution in the introduction and consider the influence that these processes may have at 1264 throughout the discussion. We conclude that dilution is minimal, and that winnowing may have had some effect, but was likely not the main driver of the trends and cycles we see, with the exception of the last 3.3 Ma.

We also have expanded our discussion in several places to discuss which mechanisms may explain the trends and patterns in carbonate deposition that we observe. We especially focus the discussion on the changes in the previously unstudied interval between 17 and 5.3 Ma. Where appropriate, we have also referred to the original publications dealing with the Oligocene-early Miocene (Liebrand et al., 2016, 2017, 2018) and Plio-Pleistocene (Bell et al., 2014, 2015), as there is already very detailed discussion of these time periods there.

**5) Figures:**

We have made all the revisions requested by reviewers concerning the figures, in addition to the following changes:
- We merged Figures 3 and 4, especially improving the presentation of the wavelets.
- We added the K intensity data to new Figure 3 and Figure 5 (previously Figure 6). Both the Si and K are now also described in the results.
- We present a new figure (new Figure 6) to highlight how the three distinctly different cyclicities in carbonate deposition observed at Site 1264
- We have redone all wavelets presented in the main & supplementary figures to use a more colour-blind friendly scheme.
- We have added epoch and stages to Figures 7 and 8
- We have improved the annotations of Figures 3, 5, 7 and 8.
- Made sedimentation rates m/Myr to be consistent with the main text.
- We have added two supplementary figures:
  o one showing all calibrated XRF data now available at Site 1264
  o 4-part oversized figure showing the age-depth ties for the astrochronology in greater detail.

Main document changes and comments

**Page 1: Deleted  Anna Joy Drury   11/05/2021 14:38:00**
Icehouse

**Page 1: Inserted  Anna Joy Drury   11/05/2021 14:38:00**
cryosphere

**Page 1: Deleted  Anna Joy Drury   11/05/2021 15:56:00**
over the past 30 million years (Myr)

**Page 1: Deleted  Anna Joy Drury   24/05/2021 14:45:00**
a

**Page 1: Deleted  Anna Joy Drury   24/05/2021 14:45:00**
a

**Page 1: Deleted  Anna Joy Drury   24/05/2021 14:45:00**
world

**Page 1: Inserted  Anna Joy Drury   11/05/2021 15:56:00**
over the past 30 million years (Myr)

**Page 1: Deleted  Anna Joy Drury   11/05/2021 15:53:00**
; however, the exact development of orbital-scale climate variability is less well understood

**Page 1: Deleted  Anna Joy Drury   11/05/2021 15:57:00**
Angola Basin side of the

**Page 1: Inserted  Anna Joy Drury   11/05/2021 15:58:00**
. This

**Page 1: Deleted  Anna Joy Drury   11/05/2021 15:58:00**
, which

**Page 1: Deleted  Anna Joy Drury   11/05/2021 15:58:00**
site

**Page 1: Inserted  Anna Joy Drury   11/05/2021 15:58:00**
location

**Page 1: Deleted  Anna Joy Drury   28/05/2021 22:25:00**
/or

**Page 1: Deleted  Anna Joy Drury   28/05/2021 22:27:00**
strong

**Page 1: Inserted  Anna Joy Drury   28/05/2021 22:27:00**
pervasive

**Page 1: Deleted  Anna Joy Drury   28/05/2021 22:25:00**
* * *
the increasing influence

**Page 1: Inserted  Anna Joy Drury   28/05/2021 22:20:00**
greater importance

**Page 1: Inserted  Anna Joy Drury   28/05/2021 22:20:00**
, such as increased glacial activity and high-latitude cooling

**Page 1: Inserted  Anna Joy Drury   18/03/2021 13:36:00**
s

**Page 1: Inserted  Anna Joy Drury   11/05/2021 15:54:00**
between 14-13 Ma

**Page 1: Inserted  Anna Joy Drury   24/05/2021 14:48:00**
recurrent

**Page 1: Deleted  Anna Joy Drury   24/05/2021 14:48:00**
an

**Page 1: Deleted  Anna Joy Drury   24/05/2021 14:48:00**
es

**Page 1: Inserted  Anna Joy Drury   11/05/2021 15:50:00**

**Page 1: Deleted  Anna Joy Drury   11/05/2021 15:50:00**
-

**Page 1: Inserted  Anna Joy Drury   24/05/2021 14:45:00**
-early Pliocene

**Page 1: Deleted  Anna Joy Drury   24/05/2021 14:46:00**
, which is

**Page 1: Deleted  Anna Joy Drury   24/05/2021 14:46:00**
;

**Page 1: Deleted  Anna Joy Drury   24/05/2021 14:47:00**
, but not exactly,

**Page 1: Inserted  Anna Joy Drury   11/05/2021 15:54:00**
At Site 1264, the onset of the LMBB roughly coincides with appearance of strong obliquity pacing of %CaCO$_3$ reflecting increased high latitude forcing.

**Page 1: Deleted  Anna Joy Drury   28/05/2021 22:29:00**
an

**Page 1: Inserted  Anna Joy Drury   11/05/2021 16:02:00**
, due to enhanced glacial activity and increased meridional temperature gradients

**Page 2: Deleted  Anna Joy Drury   17/03/2021 12:17:00**
icehouse
* * *
**Page 2: Inserted  Anna Joy Drury   17/03/2021 12:17:00**
Coolhouse

**Page 2: Deleted  Anna Joy Drury   17/03/2021 12:17:00**
bipolar

**Page 2: Inserted  Anna Joy Drury   17/03/2021 12:17:00**
Icehouse

**Page 2: Deleted  Anna Joy Drury   24/05/2021 14:48:00**
sheets

**Page 2: Inserted  Anna Joy Drury   24/05/2021 14:48:00**
volumes (Liebrand et al., 2017)

**Page 2: Deleted  Anna Joy Drury   11/05/2021 14:36:00**
Unipolar

**Page 2: Inserted  Anna Joy Drury   11/05/2021 14:36:00**
unipolar

**Page 2: Deleted  Anna Joy Drury   11/05/2021 14:36:00**
icehouse

**Page 2: Inserted  Anna Joy Drury   24/05/2021 13:04:00**
Coolhouse

**Page 2: Deleted  Anna Joy Drury   11/05/2021 14:37:00**
 icehouse

**Page 2: Inserted  Anna Joy Drury   11/05/2021 14:37:00**

**Page 2: Deleted  Anna Joy Drury   17/03/2021 12:22:00**
The evolution of orbital-scale climate and carbon cycle dynamics across this interval remains relatively unscrutinised (Turner, 2014; De Vleeschouwer et al., 2017).

**Page 2: Deleted  Anna Joy Drury   24/05/2021 16:40:00**
the

**Page 2: Inserted  Anna Joy Drury   24/05/2021 16:40:00**
Earth's climate and

**Page 2: Inserted  Anna Joy Drury   17/03/2021 12:25:00**
, as well as sedimentary processes such as winnowing and dilution

**Page 2: Moved to page 2 (Move #5)         Anna Joy Drury   24/05/2021 16:42:00**
* * *
Understanding past changes in surface water productivity and deep-sea dissolution can inform about past climate development, and vice versa, how global processes affected regional production and deposition of biogenic carbonates.

**Page 2: Deleted  Anna Joy Drury   24/05/2021 16:42:00**
is a

**Page 2: Inserted  Anna Joy Drury   24/05/2021 16:42:00**
are

**Page 2: Inserted  Anna Joy Drury   24/05/2021 16:42:00**
s

**Page 2: Inserted  Anna Joy Drury   28/05/2021 16:24:00**
lysocline and

**Page 2: Inserted  Anna Joy Drury   28/05/2021 16:28:00**
 at greater depths and /or

**Page 2: Inserted  Anna Joy Drury   24/05/2021 16:43:00**
Changes in deep-sea currents can alter the composition of the sediment through processes like winnowing or dilution, which respectively remove fine-grained material or increase certain sedimentary components relative to others (e.g., increased dilution with terrigenous material).

**Page 2: Moved from page 2 (Move #5)         Anna Joy Drury   24/05/2021 16:42:00**
Understanding past changes in carbonate deposition surface water productivity and deep-sea dissolution can inform about past climate development by helping to disentangle , and vice versa, how global processes affected regional production and deposition of biogenic carbonates.

**Page 2: Inserted  Anna Joy Drury   24/05/2021 16:45:00**
carbonate deposition

**Page 2: Deleted  Anna Joy Drury   24/05/2021 16:48:00**
surface water productivity and deep-sea dissolution

**Page 2: Inserted  Anna Joy Drury   24/05/2021 16:48:00**
 by helping to disentangle

**Page 2: Deleted  Anna Joy Drury   24/05/2021 16:48:00**
, and vice versa,

**Page 2: Inserted  Anna Joy Drury   24/05/2021 15:03:00**
,

**Page 2: Deleted  Anna Joy Drury   24/05/2021 15:03:00**
both

**Page 2: Deleted  Anna Joy Drury   24/05/2021 15:04:00**

strategic

**Page 2: Inserted  Anna Joy Drury   24/05/2021 15:03:00**
was affected by

**Page 2: Deleted  Anna Joy Drury   24/05/2021 15:04:00**
has the potential to record

**Page 2: Deleted  Anna Joy Drury   24/05/2021 15:04:00**
changes

**Page 2: Inserted  Anna Joy Drury   24/05/2021 15:04:00**
conditions

**Page 3: Inserted  Anna Joy Drury   11/05/2021 15:57:00**
 the Angola Basin side of

**Page 3: Deleted  Anna Joy Drury   28/05/2021 16:45:00**
se

**Page 3: Inserted  Anna Joy Drury   17/03/2021 12:34:00**
 resulting

**Page 3: Deleted  Anna Joy Drury   28/05/2021 16:53:00**
, astronomically tuned carbonate

**Page 3: Deleted  Anna Joy Drury   28/05/2021 16:49:00**
allow us to

**Page 3: Inserted  Anna Joy Drury   28/05/2021 16:49:00**
help determine shifts in

**Page 3: Deleted  Anna Joy Drury   28/05/2021 16:42:00**
 investigate how

**Page 3: Deleted  Anna Joy Drury   28/05/2021 16:43:00**
se

**Page 3: Deleted  Anna Joy Drury   28/05/2021 16:48:00**
regimes

**Page 3: Inserted  Anna Joy Drury   28/05/2021 16:43:00**
of Southeast Atlantic CaCO$_3$ deposition in

**Page 3: Inserted  Anna Joy Drury   28/05/2021 16:43:00**
ion

**Page 3: Deleted  Anna Joy Drury   28/05/2021 16:43:00**
e

**Page 3: Inserted  Anna Joy Drury   28/05/2021 16:46:00**
We investigate how widespread Miocene warmth followed by Antarctic glaciation influenced the pacing and preservation of Southeast Atlantic carbonate deposition. Finally, we establish the relative timing of the late Miocene-early Pliocene Biogenic Bloom (LMBB; acronym from Lyle et al., 2019) in the Southeast Atlantic versus Pacific Oceans and explore what this reveals about the global and regional driving forces of this multi-million-year productivity event.

**Page 3: Deleted  Anna Joy Drury   17/03/2021 15:38:00**
composite section

**Page 3: Inserted  Anna Joy Drury   17/03/2021 15:38:00**
stratigraphic splice was developed

**Page 3: Deleted  Anna Joy Drury   17/03/2021 15:38:00**
was developed

**Page 3: Deleted  Anna Joy Drury   29/05/2021 17:32:00**
was

**Page 3: Inserted  Anna Joy Drury   29/05/2021 17:32:00**
were

**Page 3: Inserted  Anna Joy Drury   18/03/2021 13:37:00**
an

**Page 3: Deleted  Anna Joy Drury   29/05/2021 17:32:00**
are

**Page 3: Inserted  Anna Joy Drury   29/05/2021 17:32:00**
were exceptionally

**Page 3: Deleted  Anna Joy Drury   24/05/2021 15:45:00**

**Page 3: Inserted  Anna Joy Drury   24/05/2021 15:45:00**

**Page 3: Inserted  Anna Joy Drury   24/05/2021 15:50:00**
; LSR =

**Page 3: Deleted  Anna Joy Drury   24/05/2021 15:50:00**
,

**Page 3: Deleted  Anna Joy Drury   24/05/2021 15:42:00**

**Page 3: Inserted  Anna Joy Drury   24/05/2021 15:43:00**
3.9

**Page 3: Deleted  Anna Joy Drury   24/05/2021 15:42:00**

**Page 3: Inserted  Anna Joy Drury   24/05/2021 15:48:00**
5.4

**Page 3: Inserted  Anna Joy Drury   24/05/2021 15:51:00**
, average of 4.7 m/Myr

**Page 3: Deleted  Anna Joy Drury   24/05/2021 15:50:00**
 and early Plio-Pleistocene intervals (3-0 Ma, 4-8 m/Myr)

**Page 3: Inserted  Anna Joy Drury   17/03/2021 12:38:00**
Higher shipboard

**Page 3: Deleted  Anna Joy Drury   17/03/2021 12:38:00**
are higher

**Page 3: Inserted  Anna Joy Drury   17/03/2021 12:38:00**
occurred

**Page 3: Inserted  Anna Joy Drury   24/05/2021 15:49:00**

**Page 3: Deleted  Anna Joy Drury   24/05/2021 15:49:00**

**Page 3: Inserted  Anna Joy Drury   24/05/2021 15:51:00**
; LSR = 5.3-9.3 m/Myr, average of 7.1m/Myr

**Page 3: Deleted  Anna Joy Drury   17/03/2021 12:38:00**
, 6-22 m/Myr

**Page 3: Inserted  Anna Joy Drury   24/05/2021 15:51:00**
and early Plio-Pleistocene (3-0 Ma; LSR = 4.5-7.4 m/Myr, average of 6.0 m/Myr). The

**Page 3: Deleted  Anna Joy Drury   24/05/2021 15:52:00**
and are

**Page 3: Inserted  Anna Joy Drury   24/05/2021 15:52:00**
shipboard LSR occurred

**Page 3: Inserted  Anna Joy Drury   24/05/2021 15:52:00**
LSR average 15.9 m/Myr

**Page 3: Deleted  Anna Joy Drury   24/05/2021 15:52:00**
they range between

**Page 3: Inserted  Anna Joy Drury   24/05/2021 15:52:00**
(7.7-30.5

**Page 3: Deleted  Anna Joy Drury   24/05/2021 15:53:00**
8 and 31

**Page 3: Inserted  Anna Joy Drury   17/03/2021 12:38:00**
shipboard LSR for the

**Page 3: Deleted  Anna Joy Drury   17/03/2021 12:38:00**
shipboard LSR

**Page 3: Moved to page 4 (Move #4)       Anna Joy Drury   24/05/2021 15:57:00**
(Liebrand et al., 2011, 2016, 2017, 2018; Bell et al., 2014, 2015)

**Page 4: Moved from page 3 (Move #4)       Anna Joy Drury   24/05/2021 15:57:00**
(Liebrand et al., 2011, 2016, 2017, 2018; Bell et al., 2014, 2015)

**Page 4: Inserted  Anna Joy Drury   17/03/2021 15:39:00**
composite

**Page 4: Deleted  Anna Joy Drury   17/03/2021 15:39:00**
composite

**Page 4: Deleted  Anna Joy Drury   17/03/2021 13:52:00**
each

**Page 4: Inserted  Anna Joy Drury   17/03/2021 13:52:00**
s

**Page 4: Deleted  Anna Joy Drury   17/03/2021 13:52:00**
was

**Page 4: Inserted  Anna Joy Drury   17/03/2021 13:52:00**
were

**Page 4: Deleted  Anna Joy Drury   17/03/2021 12:39:00**
composite

**Page 4: Inserted  Anna Joy Drury   17/03/2021 12:39:00**
single

**Page 4: Deleted  Anna Joy Drury   17/03/2021 13:53:00**
core

**Page 4: Inserted  Anna Joy Drury   17/03/2021 13:53:00**
 for each core

**Page 4: Inserted  Anna Joy Drury   17/03/2021 14:05:00**
"

**Page 4: Inserted  Anna Joy Drury   17/03/2021 14:05:00**

"

**Page 4: Inserted Anna Joy Drury 17/03/2021 13:53:00**
-

**Page 4: Deleted Anna Joy Drury 17/03/2021 13:53:00**

**Page 4: Deleted Anna Joy Drury 17/03/2021 13:56:00**
composite

**Page 4: Inserted Anna Joy Drury 17/03/2021 13:56:00**
individual core

**Page 4: Deleted Anna Joy Drury 17/03/2021 13:57:00**
core

**Page 4: Inserted Anna Joy Drury 17/03/2021 13:56:00**
-

**Page 4: Inserted Anna Joy Drury 17/03/2021 13:56:00**
core-box

**Page 4: Deleted Anna Joy Drury 17/03/2021 13:57:00**
CODD (

**Page 4: Inserted Anna Joy Drury 17/03/2021 13:57:00**
the "

**Page 4: Inserted Anna Joy Drury 17/03/2021 14:05:00**
"

**Page 4: Deleted Anna Joy Drury 17/03/2021 13:57:00**
)

**Page 4: Deleted Anna Joy Drury 17/03/2021 13:57:00**
core

**Page 4: Inserted Anna Joy Drury 17/03/2021 13:57:00**
-

**Page 4: Deleted Anna Joy Drury 17/03/2021 13:57:00**
composite

**Page 4: Inserted Anna Joy Drury 17/03/2021 13:57:00**
core-box

**Page 4: Deleted Anna Joy Drury 17/03/2021 13:57:00**
reflect

**Page 4: Inserted Anna Joy Drury 17/03/2021 13:57:00**
represent

**Page 4: Inserted Anna Joy Drury 17/03/2021 13:59:00**
-

**Page 4: Deleted Anna Joy Drury 17/03/2021 13:59:00**

**Page 4: Deleted Anna Joy Drury 17/03/2021 13:59:00**
-

**Page 4: Inserted Anna Joy Drury 17/03/2021 13:59:00**
-

**Page 4: Deleted Anna Joy Drury 17/03/2021 13:59:00**
composite

**Page 4: Inserted Anna Joy Drury 17/03/2021 14:02:00**
preferentially

**Page 4: Deleted Anna Joy Drury 17/03/2021 13:59:00**
composite

**Page 4: Inserted Anna Joy Drury 17/03/2021 14:00:00**
,

**Page 4: Deleted Anna Joy Drury 17/03/2021 14:00:00**
they visually highlight

**Page 4: Deleted Anna Joy Drury 17/03/2021 14:00:00**
better

**Page 4: Inserted Anna Joy Drury 17/03/2021 14:00:00**
is better visible in these images

**Page 4: Deleted Anna Joy Drury 17/03/2021 15:39:00**
composite

**Page 4: Inserted Anna Joy Drury 17/03/2021 15:39:00**
stratigraphic

**Page 4: Inserted Anna Joy Drury 17/03/2021 14:03:00**
The individual core-box/line-scan core images were then combined into a single composite image along the revised Site 1264 splice using the "SpliceImages" function.

**Page 4: Deleted Anna Joy Drury 17/03/2021 14:03:00**
composite

**Page 4: Inserted Anna Joy Drury 17/03/2021 14:03:00**
individual

**Page 4: Inserted Anna Joy Drury 17/03/2021 14:02:00**
-

**Page 4: Deleted Anna Joy Drury 17/03/2021 14:02:00**

**Page 4: Inserted Anna Joy Drury 17/03/2021 14:02:00**
-

**Page 4: Deleted Anna Joy Drury 17/03/2021 14:02:00**

**Page 4: Inserted Anna Joy Drury 17/03/2021 14:09:00**
in the 1264 splice image

**Page 4: Deleted Anna Joy Drury 17/03/2021 14:10:00**
to form a continuous composite core image spanning the early Oligocene to present day

**Page 4: Deleted Anna Joy Drury 17/03/2021 14:10:00**
,

**Page 4: Inserted Anna Joy Drury 17/03/2021 14:10:00**
This resulted in a continuous spliced image of the sedimentary succession at Site 1264-1265 spanning the early Oligocene to present day.

**Page 4: Deleted Anna Joy Drury 17/03/2021 14:28:00**
195

**Page 4: Inserted Anna Joy Drury 17/03/2021 14:28:00**
205

**Page 4: Deleted Anna Joy Drury 17/03/2021 14:29:00**
195

**Page 4: Inserted Anna Joy Drury 17/03/2021 14:29:00**
205

**Page 4: Deleted Anna Joy Drury 29/05/2021 17:33:00**
was

**Page 4: Inserted Anna Joy Drury 29/05/2021 17:33:00**
were

**Page 4: Deleted Anna Joy Drury 17/03/2021 14:31:00**
three

**Page 4: Inserted Anna Joy Drury 17/03/2021 14:31:00**
four

**Page 4: Inserted Anna Joy Drury 17/03/2021 14:31:00**
2011 (195-205 rmcd),

**Page 4: Inserted Anna Joy Drury 17/03/2021 15:02:00**
directly at the core surface of Site 1264 archive halves

**Page 4: Deleted Anna Joy Drury 17/03/2021 15:02:00**
during

**Page 4: Inserted Anna Joy Drury 17/03/2021 15:02:00**
using

**Page 4: Inserted Anna Joy Drury 17/03/2021 15:01:00**
.2

**Page 4: Deleted Anna Joy Drury 17/03/2021 15:02:00**
down-core

**Page 4: Inserted Anna Joy Drury 17/03/2021 15:02:00**
a

**Page 4: Inserted Anna Joy Drury 17/03/2021 15:01:00**
mm down-core and

**Page 4: Inserted Anna Joy Drury 17/03/2021 15:01:00**
-

**Page 4: Inserted Anna Joy Drury 17/03/2021 15:02:00**
cross-core

**Page 4: Deleted Anna Joy Drury 17/03/2021 15:02:00**
directly at the core surface of Site 1264 archive halves

**Page 4: Inserted Anna Joy Drury 17/03/2021 15:03:00**
with the same slit conditions

**Page 4: Deleted Anna Joy Drury 17/03/2021 15:12:00**
during

**Page 4: Inserted Anna Joy Drury 17/03/2021 15:12:00**
using

**Page 4: Deleted Anna Joy Drury 17/03/2021 15:12:00**
relatively

**Page 4: Deleted Anna Joy Drury 17/03/2021 15:13:00**
splice

**Page 4: Inserted  Anna Joy Drury   17/03/2021 15:13:00**
accurately correlate between holes

**Page 4: Inserted  Anna Joy Drury   17/03/2021 14:32:00**
2011) MARUM XRF III, 10 kV/0.15 mA/10 s count time/Cl-Rh filter (see also

**Page 4: Deleted  Anna Joy Drury   17/03/2021 14:41:00**
(

**Page 4: Inserted  Anna Joy Drury   17/03/2021 14:40:00**
(Liebrand et al., 2016);

**Page 4: Inserted  Anna Joy Drury   17/03/2021 15:06:00**
/Cl-Rh filter

**Page 4: Inserted  Anna Joy Drury   17/03/2021 15:08:00**
/no filter

**Page 4: Deleted  Anna Joy Drury   29/05/2021 17:33:00**

**Page 4: Inserted  Anna Joy Drury   17/03/2021 15:08:00**
/no filter

**Page 5: Inserted  Anna Joy Drury   11/05/2021 16:43:00**
All data were inspected directly following collection and outliers were removed if they were clearly associated with cracks and/or uneven sediment surface. Following this, the ln(Ca/Fe) data were additionally despiked using the CODD editing functions.

**Page 5: Deleted  Anna Joy Drury   29/05/2021 17:33:00**
was

**Page 5: Inserted  Anna Joy Drury   29/05/2021 17:33:00**
were

**Page 5: Inserted  Anna Joy Drury   17/03/2021 15:08:00**
including

**Page 5: Inserted  Anna Joy Drury   29/05/2021 14:26:00**
 and 2

**Page 5: Inserted  Anna Joy Drury   17/03/2021 15:52:00**

**Page 5: Deleted  Anna Joy Drury   11/05/2021 16:29:00**

**Page 5: Inserted  Anna Joy Drury   11/05/2021 12:38:00**
80.238

**Page 5: Deleted  Anna Joy Drury   11/05/2021 12:38:00**
79.642

**Page 5: Inserted  Anna Joy Drury   10/05/2021 18:10:00**
±1.069

**Page 5: Deleted  Anna Joy Drury   11/05/2021 12:38:00**

**Page 5: Inserted  Anna Joy Drury   11/05/2021 12:40:00**
2.526 ± 0.188

**Page 5: Deleted  Anna Joy Drury   11/05/2021 12:40:00**
2.6441

**Page 5: Inserted  Anna Joy Drury   11/05/2021 12:47:00**
0.622

**Page 5: Deleted  Anna Joy Drury   11/05/2021 12:47:00**
0.7572

**Page 5: Deleted  Anna Joy Drury   29/05/2021 14:27:00**

**Page 5: Inserted  Anna Joy Drury   29/05/2021 14:27:00**

**Page 5: Deleted  Anna Joy Drury   11/05/2021 12:52:00**
This calibration is within the 2σ uncertainty of the

**Page 5: Inserted  Anna Joy Drury   11/05/2021 12:51:00**
The

**Page 5: Inserted  Anna Joy Drury   11/05/2021 12:52:00**
 is within the 2σ uncertainty of the new %CaCO₃ calibration, which equates to ±2.2% in the calibrated %CaCO₃ dataset.

**Page 5: Inserted  Anna Joy Drury   11/05/2021 12:55:00**
    The uncertainty in the calibration likely originates from the scatter of the shipboard coulometry-derived %CaCO₃ data that were used in the calibration. This uncertainty only pertains to the absolute %CaCO₃ values. The trends and cyclicity observed in the calibrated CaCO₃ data are independent of this uncertainty, as these patterns are present in the raw ln(Ca/Fe) timeseries.

**Page 5: Inserted  Anna Joy Drury   11/05/2021 12:53:00**
e new and recalibrated %CaCO₃

**Page 5: Deleted  Anna Joy Drury   11/05/2021 12:53:00**
is

**Page 5: Inserted  Anna Joy Drury   11/05/2021 12:54:00**
from Site 1264

**Page 5: Deleted  Anna Joy Drury   29/05/2021 17:33:00**
was

**Page 5: Inserted  Anna Joy Drury   29/05/2021 17:33:00**
were

**Page 5: Inserted  Anna Joy Drury   11/05/2021 13:01:00**

$$MAR_{detrital} = MAR_{Bulk} - MAR_{CaCO_3} \qquad (3)$$

**Page 5: Deleted  Anna Joy Drury   29/05/2021 17:34:00**
using t

**Page 5: Inserted  Anna Joy Drury   29/05/2021 17:34:00**
T

**Page 5: Inserted  Anna Joy Drury   29/05/2021 17:34:00**
was

**Page 5: Inserted  Anna Joy Drury   29/05/2021 17:34:00**
.

**Page 5: Deleted  Anna Joy Drury   29/05/2021 17:34:00**
and d

**Page 5: Inserted  Anna Joy Drury   29/05/2021 17:34:00**
D

**Page 5: Inserted  Anna Joy Drury   29/05/2021 17:34:00**
was

**Page 5: Deleted  Anna Joy Drury   29/05/2021 14:27:00**

**Page 5: Inserted  Anna Joy Drury   29/05/2021 14:27:00**

**Page 5: Inserted  Anna Joy Drury   17/03/2021 15:31:00**
    The uncertainty in the MARs is difficult to quantify. The largest uncertainties affecting bulk, CaCO₃ and detrital MARs arise from uncertainties in the $\rho_{dry}$, which was calculated using shipboard GRA and discrete dry density data, and the LSR, both of which are difficult to estimate. CaCO₃ MARs additionally have ±2.2% 2σ calibration uncertainty. However, as %CaCO₃ is so high at Site 1264, the %CaCO₃ calibration uncertainty will have a smaller affect compared with the changes in LSR. Because detrital MARs are low and calculated using the difference between bulk and CaCO₃ MARs, changes in detrital MARs should be treated cautiously.

**Page 6: Deleted  Anna Joy Drury   17/03/2021 15:33:00**
composite

**Page 6: Inserted  Anna Joy Drury   17/03/2021 15:33:00**
core

**Page 6: Inserted  Anna Joy Drury   17/03/2021 15:33:00**
, leading to duplicated and/or missing intervals in the shipboard

**Page 6: Deleted  Anna Joy Drury   17/03/2021 15:33:00**
 and

**Page 6: Inserted  Anna Joy Drury   17/03/2021 15:34:00**
ese misalignments

**Page 6: Deleted  Anna Joy Drury   17/03/2021 15:34:00**
is

**Page 6: Inserted  Anna Joy Drury   17/03/2021 15:34:00**
are

**Page 6: Deleted  Anna Joy Drury   17/03/2021 15:34:00**
is

**Page 6: Deleted  Anna Joy Drury   17/03/2021 15:35:00**
Using

**Page 6: Inserted  Anna Joy Drury   17/03/2021 15:36:00**
Predominantly

**Page 6: Inserted  Anna Joy Drury   17/03/2021 15:36:00**
using

**Page 6: Deleted  Anna Joy Drury   18/03/2021 13:59:00**
196.13

**Page 6: Inserted  Anna Joy Drury   18/03/2021 13:59:00**
205

**Page 6: Deleted  Anna Joy Drury   29/05/2021 14:27:00**

**Page 6: Inserted  Anna Joy Drury   29/05/2021 14:27:00**

**Page 6: Deleted  Anna Joy Drury   29/05/2021 17:34:00**
was

**Page 6: Inserted  Anna Joy Drury   29/05/2021 17:34:00**

were

**Page 6: Deleted** Anna Joy Drury   29/05/2021 14:27:00

**Page 6: Inserted** Anna Joy Drury   29/05/2021 14:27:00

**Page 6: Deleted** Anna Joy Drury   17/03/2021 15:36:00

and splice

**Page 6: Deleted** Anna Joy Drury   24/05/2021 16:16:00

Hole

**Page 6: Inserted** Anna Joy Drury   24/05/2021 16:16:00

Core

**Page 6: Inserted** Anna Joy Drury   24/05/2021 16:16:00

H

**Page 6: Deleted** Anna Joy Drury   24/05/2021 16:16:00

Hole

**Page 6: Inserted** Anna Joy Drury   24/05/2021 16:16:00

Core

**Page 6: Inserted** Anna Joy Drury   24/05/2021 16:16:00

H

**Page 6: Deleted** Anna Joy Drury   17/03/2021 15:37:00

in order

**Page 6: Deleted** Anna Joy Drury   17/03/2021 15:37:00

composite

**Page 6: Deleted** Anna Joy Drury   17/03/2021 15:37:00

and

**Page 6: Inserted** Anna Joy Drury   17/03/2021 15:37:00

 new

**Page 6: Inserted** Anna Joy Drury   17/03/2021 15:37:00

 and

**Page 6: Deleted** Anna Joy Drury   17/03/2021 15:37:00

, together with the

**Page 6: Deleted** Anna Joy Drury   17/03/2021 15:37:00

composite

**Page 6: Inserted** Anna Joy Drury   17/03/2021 15:37:00

stratigraphic

**Page 6: Inserted** Anna Joy Drury   17/03/2021 15:40:00

ary

**Page 6: Deleted** Anna Joy Drury   17/03/2021 15:40:00

column

**Page 6: Inserted** Anna Joy Drury   17/03/2021 15:40:00

succession (0-205 rmcd)

**Page 6: Inserted** Anna Joy Drury   17/03/2021 15:41:00

Site 1264

**Page 6: Inserted** Anna Joy Drury   17/03/2021 15:41:00

, stratigraphic

**Page 6: Deleted** Anna Joy Drury   17/03/2021 15:41:00

/

**Page 6: Deleted** Anna Joy Drury   29/05/2021 14:27:00

**Page 6: Inserted** Anna Joy Drury   29/05/2021 14:27:00

**Page 6: Inserted** Anna Joy Drury   11/05/2021 14:39:00

, which were

**Page 6: Deleted** Anna Joy Drury   11/05/2021 14:39:00

.

**Page 6: Inserted** Anna Joy Drury   17/03/2021 15:44:00

filled with new isotope data (Westerhold et al., 2020).

**Page 6: Inserted** Anna Joy Drury   12/05/2021 11:51:00

**XRF intensities,**

**Page 6: Deleted** Anna Joy Drury   11/05/2021 13:15:00

**Page 6: Inserted** Anna Joy Drury   18/03/2021 13:22:00

The range of observed %CaCO$_3$ variability is close to the 2.2% uncertainty associated with the calibration. However, we are confident that both the long-term trends and short-term variability discussed below represent true changes in carbonate content, as these patterns originate in the original ln(Ca/Fe) ratio. The calibration uncertainty is most relevant to the absolute carbonate content.

**Page 7: Inserted** Anna Joy Drury   18/03/2021 13:39:00

span 93-96%

**Page 7: Deleted** Anna Joy Drury   18/03/2021 13:39:00

span 93-96%

**Page 7: Deleted** Anna Joy Drury   18/03/2021 13:39:00

is

**Page 7: Inserted** Anna Joy Drury   18/03/2021 13:39:00

agrees

**Page 7: Deleted** Anna Joy Drury   18/03/2021 13:39:00

of

**Page 7: Inserted** Anna Joy Drury   18/03/2021 13:39:00

with

**Page 7: Inserted** Anna Joy Drury   18/03/2021 13:39:00

s

**Page 7: Deleted** Anna Joy Drury   18/03/2021 13:40:00

especially

**Page 7: Inserted** Anna Joy Drury   18/03/2021 13:40:00

especially

**Page 7: Inserted** Anna Joy Drury   18/03/2021 13:40:00

s

**Page 7: Inserted** Anna Joy Drury   18/03/2021 13:41:00

s

**Page 7: Inserted** Anna Joy Drury   18/03/2021 13:41:00

s

**Page 7: Inserted** Anna Joy Drury   24/05/2021 17:53:00

 (mid Miocene)

**Page 7: Inserted** Anna Joy Drury   18/03/2021 13:42:00

then

**Page 7: Inserted** Anna Joy Drury   18/03/2021 13:42:00

es

**Page 7: Inserted** Anna Joy Drury   24/05/2021 17:54:00

0

**Page 7: Deleted** Anna Joy Drury   18/03/2021 13:42:00

%

**Page 7: Inserted** Anna Joy Drury   18/03/2021 13:42:00

the

**Page 7: Inserted** Anna Joy Drury   18/03/2021 13:42:00

content

**Page 7: Deleted** Anna Joy Drury   18/03/2021 13:42:00

remains

**Page 7: Inserted** Anna Joy Drury   18/03/2021 13:42:00

decreases slightly to

**Page 7: Inserted** Anna Joy Drury   24/05/2021 17:54:00

 (early Pliocene)

**Page 7: Inserted** Anna Joy Drury   24/05/2021 17:54:00

-early Pliocene

**Page 7: Deleted** Anna Joy Drury   29/05/2021 17:35:00

,

**Page 7: Inserted** Anna Joy Drury   24/05/2021 17:58:00

s

**Page 7: Inserted** Anna Joy Drury   24/05/2021 17:54:00

 (Pleistocene)

**Page 7: Inserted** Anna Joy Drury   12/05/2021 11:52:00

 The Si and K intensities are comparable throughout the record, although Si is generally slightly higher than K (Fig 3). Both elements, together with Fe and Ti intensities, display the same short-term variability and long-term trends (Fig 3 and Supplementary Figure 2), indicating that these elements reflect changes in aluminosilicates. As the trends of Si and K are inverse to those seen in the CaCO$_3$ content, this supports that Site 1264 is predominantly composed of carbonate and clay, with minimal influence of biogenic silica. The amplitude of changes in Si and K becomes much smaller relative to CaCO$_3$ content changes between ~115-0 rmcd compared to ~315-115 rmcd.

**Page 7: Inserted** Anna Joy Drury   18/03/2021 13:50:00

Because

**Page 7: Deleted** Anna Joy Drury   18/03/2021 13:50:00

. As a result

**Page 7: Deleted** Anna Joy Drury   11/05/2021 14:00:00

sedimentation rates

**Page 7: Inserted** Anna Joy Drury   11/05/2021 14:00:00

LSR

**Page 7: Inserted  Anna Joy Drury    11/05/2021 14:00:00**
LSR also strongly affect detrital MARs; however, these remain low throughout at Site 1264 (0.01-0.2 g/cm²/kyr).

**Page 7: Inserted  Anna Joy Drury    18/03/2021 13:52:00**
CaCO₃

**Page 7: Deleted  Anna Joy Drury    11/05/2021 14:07:00**
variability

**Page 7: Inserted  Anna Joy Drury    11/05/2021 14:07:00**
changes

**Page 7: Inserted  Anna Joy Drury    11/05/2021 14:06:00**
; however, this variability is smaller than that variability

**Page 7: Deleted  Anna Joy Drury    11/05/2021 14:07:00**
superimposed upon variability

**Page 7: Inserted  Anna Joy Drury    11/05/2021 14:07:00**
(see section 4.2)

**Page 7: Inserted  Anna Joy Drury    29/05/2021 14:36:00**
(Fig 3)

**Page 7: Deleted  Anna Joy Drury    18/03/2021 14:00:00**
record

**Page 7: Deleted  Anna Joy Drury    18/03/2021 14:00:00**
sequence recovered

**Page 7: Inserted  Anna Joy Drury    18/03/2021 14:00:00**
succession

**Page 7: Inserted  Anna Joy Drury    18/03/2021 14:00:00**

**Page 7: Deleted  Anna Joy Drury    18/03/2021 14:00:00**

**Page 8: Deleted  Anna Joy Drury    29/05/2021 17:35:00**
.

**Page 8: Inserted  Anna Joy Drury    18/03/2021 14:01:00**
u

**Page 8: Deleted  Anna Joy Drury    29/05/2021 17:36:00**
.

**Page 8: Deleted  Anna Joy Drury    29/05/2021 17:36:00**
.

**Page 8: Deleted  Anna Joy Drury    24/05/2021 18:18:00**
kyr

**Page 8: Inserted  Anna Joy Drury    24/05/2021 18:18:00**
Myr

**Page 8: Deleted  Anna Joy Drury    29/05/2021 17:36:00**
125

**Page 8: Inserted  Anna Joy Drury    29/05/2021 17:36:00**
110

**Page 8: Deleted  Anna Joy Drury    12/05/2021 17:18:00**
cyclicity

**Page 8: Inserted  Anna Joy Drury    12/05/2021 17:18:00**
variability

**Page 8: Deleted  Anna Joy Drury    18/03/2021 14:05:00**
e.g.,

**Page 8: Inserted  Anna Joy Drury    18/03/2021 14:05:00**
which shows

**Page 8: Inserted  Anna Joy Drury    21/03/2021 14:36:00**
~

**Page 8: Inserted  Anna Joy Drury    21/03/2021 14:36:00**
(e.g. the ~95 and ~125 kyr cycles) with

**Page 8: Inserted  Anna Joy Drury    18/03/2021 14:06:00**
longer

**Page 8: Deleted  Anna Joy Drury    24/05/2021 20:45:00**
, 4

**Page 8: Deleted  Anna Joy Drury    24/05/2021 20:46:00**

**Page 8: Inserted  Anna Joy Drury    24/05/2021 20:46:00**

**Page 8: Deleted  Anna Joy Drury    29/05/2021 17:36:00**
.

**Page 8: Inserted  Anna Joy Drury    18/03/2021 14:26:00**
respectively

**Page 8: Inserted  Anna Joy Drury    18/03/2021 14:06:00**
~

**Page 8: Deleted  Anna Joy Drury    29/05/2021 14:28:00**

**Page 8: Inserted  Anna Joy Drury    29/05/2021 14:28:00**

**Page 8: Deleted  Anna Joy Drury    29/05/2021 14:28:00**

**Page 8: Inserted  Anna Joy Drury    29/05/2021 14:28:00**

**Page 8: Inserted  Anna Joy Drury    29/05/2021 14:37:00**
; Fig 4 and Supplementary Figure 10

**Page 8: Deleted  Anna Joy Drury    29/05/2021 17:36:00**
.

**Page 8: Deleted  Anna Joy Drury    18/03/2021 14:02:00**
Lithological cycles broadly varying around 2 and 0.5 m length are present in the

**Page 8: Inserted  Anna Joy Drury    18/03/2021 14:02:00**
The depth-domain

**Page 8: Inserted  Anna Joy Drury    18/03/2021 14:25:00**
between 205 and 190 rmcd highlights the lithological cycles in %CaCO₃, which broadly varies around 2 and 0.5 m in length

**Page 8: Deleted  Anna Joy Drury    18/03/2021 14:03:00**
for the interval

**Page 8: Deleted  Anna Joy Drury    18/03/2021 14:25:00**
between 205 and 190 rmcd

**Page 8: Deleted  Anna Joy Drury    24/05/2021 20:44:00**

**Page 8: Inserted  Anna Joy Drury    24/05/2021 20:44:00**

**Page 8: Deleted  Anna Joy Drury    18/03/2021 14:04:00**
decreases to low values that

**Page 8: Deleted  Anna Joy Drury    24/05/2021 18:17:00**
0.

**Page 8: Deleted  Anna Joy Drury    24/05/2021 18:18:00**
c

**Page 8: Deleted  Anna Joy Drury    18/03/2021 14:06:00**
about

**Page 8: Deleted  Anna Joy Drury    18/03/2021 14:06:00**
approximately

**Page 8: Inserted  Anna Joy Drury    18/03/2021 14:06:00**
~

**Page 8: Deleted  Anna Joy Drury    18/03/2021 14:07:00**
in the range of

**Page 8: Inserted  Anna Joy Drury    18/03/2021 14:07:00**
from ~

**Page 8: Inserted  Anna Joy Drury    18/03/2021 14:07:00**
~

**Page 8: Inserted  Anna Joy Drury    18/03/2021 14:26:00**
in the depth-domain wavelet analysis of the CaCO₃ data

**Page 8: Deleted  Anna Joy Drury    24/05/2021 20:45:00**

**Page 8: Inserted  Anna Joy Drury    24/05/2021 20:45:00**

**Page 8: Inserted  Anna Joy Drury    18/03/2021 14:27:00**
gradually shifting,

**Page 8: Deleted  Anna Joy Drury    18/03/2021 14:07:00**
are resultant from

**Page 8: Inserted  Anna Joy Drury    18/03/2021 14:07:00**
reflect

**Page 8: Deleted  Anna Joy Drury    18/03/2021 14:07:00**
, that vary

**Page 8: Inserted  Anna Joy Drury    24/05/2021 18:18:00**
0

**Page 8: Deleted  Anna Joy Drury    24/05/2021 18:18:00**
c

**Page 8: Inserted  Anna Joy Drury    24/05/2021 18:18:00**
M

**Page 8: Deleted  Anna Joy Drury    24/05/2021 18:18:00**

**Page 8: Deleted Anna Joy Drury 24/05/2021 18:18:00**

k

**Page 8: Deleted Anna Joy Drury 24/05/2021 18:18:00**

.

**Page 8: Deleted Anna Joy Drury 24/05/2021 18:18:00**

c

**Page 8: Inserted Anna Joy Drury 24/05/2021 18:18:00**

M

**Page 8: Deleted Anna Joy Drury 24/05/2021 18:18:00**

k

**Page 8: Inserted Anna Joy Drury 18/03/2021 14:27:00**

,

**Page 8: Deleted Anna Joy Drury 18/03/2021 14:08:00**

can

**Page 8: Inserted Anna Joy Drury 18/03/2021 14:08:00**

~

**Page 8: Inserted Anna Joy Drury 18/03/2021 14:08:00**

~

**Page 8: Inserted Anna Joy Drury 18/03/2021 14:08:00**

~

**Page 8: Inserted Anna Joy Drury 18/03/2021 14:08:00**

~

**Page 8: Deleted Anna Joy Drury 29/05/2021 14:29:00**

**Page 8: Inserted Anna Joy Drury 29/05/2021 14:29:00**

**Page 8: Deleted Anna Joy Drury 18/03/2021 14:27:00**

cycle

**Page 8: Inserted Anna Joy Drury 24/05/2021 18:01:00**

of these cycles

**Page 8: Inserted Anna Joy Drury 18/03/2021 14:08:00**

,

**Page 8: Deleted Anna Joy Drury 18/03/2021 14:08:00**

is still

**Page 8: Inserted Anna Joy Drury 18/03/2021 14:08:00**

remains the

**Page 8: Inserted Anna Joy Drury 18/03/2021 14:09:00**

cycle

**Page 8: Inserted Anna Joy Drury 18/03/2021 14:10:00**

in line with the strong ~110-kyr eccentricity cycles observed

**Page 8: Deleted Anna Joy Drury 18/03/2021 14:10:00**

similar to the older interval

**Page 8: Inserted Anna Joy Drury 18/03/2021 14:10:00**

~110-kyr

**Page 9: Deleted Anna Joy Drury 29/05/2021 17:36:00**

.

**Page 9: Inserted Anna Joy Drury 18/03/2021 14:12:00**

[revised manuscript text omitted]

Page 9: Inserted  Anna Joy Drury   18/03/2021 14:39:00
etween 55 and 35 rmcd

Page 9: Inserted  Anna Joy Drury   18/03/2021 14:40:00
w

Page 9: Deleted  Anna Joy Drury   18/03/2021 14:40:00
and w

Page 9: Inserted  Anna Joy Drury   18/03/2021 14:39:00
CaCO₃ content data and benthic foraminiferal δ¹⁸O data

Page 9: Deleted  Anna Joy Drury   18/03/2021 14:41:00
these two proxy records

Page 9: Deleted  Anna Joy Drury   29/05/2021 14:29:00

Page 9: Inserted  Anna Joy Drury   29/05/2021 14:29:00

Page 9: Deleted  Anna Joy Drury   29/05/2021 17:36:00
.

Page 9: Inserted  Anna Joy Drury   29/05/2021 17:36:00
.

Page 9: Deleted  Anna Joy Drury   18/03/2021 14:41:00
In general, clear

Page 9: Inserted  Anna Joy Drury   18/03/2021 14:42:00
At Site 1264, clear

Page 9: Inserted  Anna Joy Drury   18/03/2021 14:41:00
generally

Page 9: Inserted  Anna Joy Drury   18/03/2021 14:42:00
depth-domain CaCO₃ content

Page 9: Deleted  Anna Joy Drury   18/03/2021 14:42:00
of the Site 1264 CaCO₃ content

Page 9: Inserted  Anna Joy Drury   18/03/2021 14:42:00
apart from

Page 9: Inserted  Anna Joy Drury   18/03/2021 14:42:00
except for

Page 9: Deleted  Anna Joy Drury   18/03/2021 14:42:00
somewhat

Page 9: Inserted  Anna Joy Drury   18/03/2021 14:42:00
occasional

Page 9: Inserted  Anna Joy Drury   18/03/2021 14:43:00
~1.0-1.5 m

Page 9: Deleted  Anna Joy Drury   18/03/2021 14:43:00
with periodicities of 1.0 to 1.5 m

Page 9: Deleted  Anna Joy Drury   18/03/2021 14:43:00
are able to

Page 9: Inserted  Anna Joy Drury   18/03/2021 14:43:00
can

Page 9: Deleted  Anna Joy Drury   18/03/2021 14:43:00
se

Page 9: Inserted  Anna Joy Drury   18/03/2021 14:43:00
CaCO₃ content

Page 9: Deleted  Anna Joy Drury   18/03/2021 14:43:00
ir

Page 9: Inserted  Anna Joy Drury   18/03/2021 14:43:00
of these cycles

Page 9: Deleted  Anna Joy Drury   18/03/2021 14:44:00
not as pronounced

Page 9: Inserted  Anna Joy Drury   18/03/2021 14:44:00
muted

Page 9: Deleted  Anna Joy Drury   18/03/2021 14:44:00
interval

Page 9: Inserted  Anna Joy Drury   18/03/2021 14:44:00
cycles observed

Page 9: Moved to page 9 (Move #1)        Anna Joy Drury   18/03/2021 14:51:00
We derive averaged LSR of <1 cm/kyr for this interval based on the initial bio-/magnetostratigraphic age model.

Page 9: Inserted  Anna Joy Drury   18/03/2021 14:51:00
appear to

Page 9: Inserted  Anna Joy Drury   18/03/2021 14:51:00
in the upper 35 m

Page 9: Moved to page 9 (Move #2)        Anna Joy Drury   18/03/2021 14:53:00
Based on the initial age model we note absence of clear precession and obliquity paced cyclicity in both benthic foraminiferal δ¹⁸O and CaCO₃ content records during the last 2.5 Ma (Supplementary Figure 8).

Page 9: Moved from page 9 (Move #1)        Anna Joy Drury   18/03/2021 14:51:00
We derive averaged LSR of <10 cm/Mkyr for this0-35 rmcd interval based on the initial bio-/magnetostratigraphic age model.

Page 9: Inserted  Anna Joy Drury   24/05/2021 18:19:00
0

Page 9: Deleted  Anna Joy Drury   24/05/2021 18:19:00
c

Page 9: Inserted  Anna Joy Drury   24/05/2021 18:19:00
M

Page 9: Deleted  Anna Joy Drury   24/05/2021 18:19:00
k

Page 9: Deleted  Anna Joy Drury   18/03/2021 14:52:00
this

Page 9: Inserted  Anna Joy Drury   18/03/2021 14:52:00
0-35 rmcd

Page 9: Deleted  Anna Joy Drury   18/03/2021 14:52:00
interval

Page 9: Inserted  Anna Joy Drury   18/03/2021 14:52:00
observed

Page 9: Deleted  Anna Joy Drury   18/03/2021 14:52:00
periodicity

Page 9: Inserted  Anna Joy Drury   18/03/2021 14:52:00
cycles

Page 9: Deleted  Anna Joy Drury   18/03/2021 14:52:00
is

Page 9: Inserted  Anna Joy Drury   18/03/2021 14:52:00
are

Page 9: Deleted  Anna Joy Drury   18/03/2021 14:52:00
either

Page 9: Inserted  Anna Joy Drury   24/05/2021 18:28:00
(Bailey et al., 2013)

Page 9: Moved from page 9 (Move #2)        Anna Joy Drury   18/03/2021 14:53:00
Based on the initial age model we note absence of clear precession and obliquity paced cyclicity in both benthic foraminiferal δ¹⁸O and CaCO₃ content records during the last 2.5 Ma (Supplementary Figure 89).

Page 9: Deleted  Anna Joy Drury   29/05/2021 14:32:00

Page 9: Inserted  Anna Joy Drury   29/05/2021 14:32:00

Page 9: Deleted  Anna Joy Drury   29/05/2021 17:37:00
.

Page 9: Inserted  Anna Joy Drury   21/03/2021 14:23:00
(

**Page 9: Deleted** Anna Joy Drury 21/03/2021 14:23:00
between

**Page 9: Deleted** Anna Joy Drury 21/03/2021 14:23:00
and

**Page 9: Inserted** Anna Joy Drury 21/03/2021 14:23:00
to

**Page 9: Inserted** Anna Joy Drury 21/03/2021 14:23:00
)

**Page 9: Deleted** Anna Joy Drury 21/03/2021 14:22:00
%

**Page 9: Inserted** Anna Joy Drury 21/03/2021 14:22:00
content

**Page 9: Deleted** Anna Joy Drury 21/03/2021 14:23:00
spanning

**Page 9: Inserted** Anna Joy Drury 21/03/2021 14:23:00
(

**Page 9: Inserted** Anna Joy Drury 21/03/2021 14:23:00
)

**Page 9: Inserted** Anna Joy Drury 21/03/2021 14:23:00
a

**Page 9: Inserted** Anna Joy Drury 24/05/2021 18:26:00
on

**Page 9: Deleted** Anna Joy Drury 24/05/2021 18:26:00
ng

**Page 9: Inserted** Anna Joy Drury 21/03/2021 14:23:00
between

**Page 10: Inserted** Anna Joy Drury 21/03/2021 14:24:00
Because of the splice revisions between 27 and 149 rmcd at Site 1264, we re-evaluated t

**Page 10: Deleted** Anna Joy Drury 21/03/2021 14:24:00
T

**Page 10: Deleted** Anna Joy Drury 21/03/2021 14:26:00
has to be re-evaluated

**Page 10: Deleted** Anna Joy Drury 21/03/2021 14:26:00

, especially

**Page 10: Inserted** Anna Joy Drury 21/03/2021 14:26:00
/

**Page 10: Deleted** Anna Joy Drury 21/03/2021 14:26:00
(

**Page 10: Deleted** Anna Joy Drury 21/03/2021 14:26:00
), resulting from the splice revisions between 27 and 149 rmcd at Site 1264

**Page 10: Inserted** Anna Joy Drury 21/03/2021 14:28:00
we updated

**Page 10: Deleted** Anna Joy Drury 21/03/2021 14:28:00
were updated

**Page 10: Inserted** Anna Joy Drury 21/03/2021 14:28:00
cumulative

**Page 10: Inserted** Anna Joy Drury 21/03/2021 14:28:00
shift in the revised

**Page 10: Deleted** Anna Joy Drury 21/03/2021 14:28:00
cumulative

**Page 10: Inserted** Anna Joy Drury 21/03/2021 14:28:00
composite

**Page 10: Deleted** Anna Joy Drury 21/03/2021 14:29:00
shift due to depth model/splice revisions in

**Page 10: Inserted** Anna Joy Drury 21/03/2021 14:29:00
of

**Page 10: Inserted** Anna Joy Drury 21/03/2021 14:30:00
in the depth-domain

**Page 10: Deleted** Anna Joy Drury 21/03/2021 14:29:00
%

**Page 10: Deleted** Anna Joy Drury 21/03/2021 14:29:00
record

**Page 10: Inserted** Anna Joy Drury 21/03/2021 14:29:00
content record

**Page 10: Inserted** Anna Joy Drury 11/05/2021 14:32:00
using the flexible best-practice guidelines outlined in Sinnesael et al. (2019)

**Page 10: Deleted** Anna Joy Drury 21/03/2021 14:30:00
However,

**Page 10: Inserted** Anna Joy Drury 21/03/2021 14:30:00
B

**Page 10: Deleted** Anna Joy Drury 21/03/2021 14:30:00
b

**Page 10: Deleted** Anna Joy Drury 21/03/2021 14:34:00
orbital forcing

**Page 10: Inserted** Anna Joy Drury 21/03/2021 14:34:00
of eccentricity (E), obliquity (T) and precession (P)

**Page 10: Deleted** Anna Joy Drury 21/03/2021 14:30:00
single and

**Page 10: Deleted** Anna Joy Drury 21/03/2021 14:30:00
to

**Page 10: Inserted** Anna Joy Drury 21/03/2021 14:30:00
for

**Page 10: Inserted** Anna Joy Drury 21/03/2021 14:31:00
we employed

**Page 10: Deleted** Anna Joy Drury 21/03/2021 14:31:00
were employed

**Page 10: Inserted** Anna Joy Drury 29/05/2021 14:33:00
and Figure 10

**Page 10: Inserted** Anna Joy Drury 21/03/2021 14:31:00
tuned

**Page 10: Inserted** Anna Joy Drury 21/03/2021 14:33:00
;

**Page 10: Deleted** Anna Joy Drury 21/03/2021 14:33:00
, with o

**Page 10: Inserted** Anna Joy Drury 21/03/2021 14:33:00
o

**Page 10: Inserted** Anna Joy Drury 21/03/2021 14:33:00

**Page 10: Deleted** Anna Joy Drury 21/03/2021 14:35:00

**Page 10: Inserted** Anna Joy Drury 21/03/2021 14:33:00
is

**Page 10: Inserted** Anna Joy Drury 21/03/2021 14:32:00
(

**Page 10: Inserted** Anna Joy Drury 21/03/2021 14:32:00
)

**Page 10: Inserted** Anna Joy Drury 24/05/2021 18:29:00
tuned

**Page 10: Deleted** Anna Joy Drury 21/03/2021 14:33:00
eccentricity

**Page 10: Inserted** Anna Joy Drury 21/03/2021 14:33:00
E

**Page 10: Inserted** Anna Joy Drury 24/05/2021 18:29:00
tuned

**Page 10: Deleted** Anna Joy Drury 21/03/2021 14:32:00
(/benthic δ$^{18}$O)

**Page 10: Inserted** Anna Joy Drury 24/05/2021 18:29:00
tuned

**Page 10: Inserted** Anna Joy Drury 21/03/2021 14:32:00
(visually aided by δ$^{18}$O, where available)

**Page 10: Inserted** Anna Joy Drury 24/05/2021 18:29:00
tuned

**Page 10: Inserted** Anna Joy Drury 21/03/2021 14:35:00
Between 30-17 Ma,

**Page 10: Deleted** Anna Joy Drury 21/03/2021 14:35:00
between 30-17 Ma

**Page 10: Inserted** Anna Joy Drury 21/03/2021 14:36:00
are both antiphase

**Page 10: Deleted** Anna Joy Drury 21/03/2021 14:36:00
in turn have an inverse relationship

**Page 10: Deleted** Anna Joy Drury 21/03/2021 14:36:00
(e.g. the ~95 and ~125 kyr cycles)

**Page 10: Inserted** Anna Joy Drury 21/03/2021 14:39:00

As

| Page 10: Deleted | Anna Joy Drury | 21/03/2021 14:39:00 |

T

| Page 10: Inserted | Anna Joy Drury | 21/03/2021 14:39:00 |

t

| Page 10: Inserted | Anna Joy Drury | 21/03/2021 14:39:00 |

,

| Page 10: Deleted | Anna Joy Drury | 21/03/2021 14:39:00 |

.

| Page 10: Inserted | Anna Joy Drury | 21/03/2021 14:39:00 |

w

| Page 10: Deleted | Anna Joy Drury | 21/03/2021 14:39:00 |

W

| Page 10: Deleted | Anna Joy Drury | 21/03/2021 14:39:00 |

therefore

| Page 10: Deleted | Anna Joy Drury | 21/03/2021 14:44:00 |

across the

| Page 10: Inserted | Anna Joy Drury | 21/03/2021 14:39:00 |

between

| Page 10: Inserted | Anna Joy Drury | 21/03/2021 14:40:00 |

.

| Page 10: Deleted | Anna Joy Drury | 21/03/2021 14:40:00 |

interval and

| Page 10: Inserted | Anna Joy Drury | 21/03/2021 14:40:00 |

We therefore also employ the Liebrand et al. (2016)

| Page 10: Inserted | Anna Joy Drury | 21/03/2021 14:40:00 |

ing strategy of

| Page 10: Deleted | Anna Joy Drury | 21/03/2021 14:40:00 |

e

| Page 10: Inserted | Anna Joy Drury | 29/05/2021 17:37:00 |

e

| Page 10: Deleted | Anna Joy Drury | 29/05/2021 17:37:00 |

E

| Page 10: Inserted | Anna Joy Drury | 21/03/2021 14:45:00 |

between 17-8 Ma

| Page 10: Deleted | Anna Joy Drury | 21/03/2021 14:42:00 |

(La2004)

| Page 10: Deleted | Anna Joy Drury | 24/05/2021 20:46:00 |

| Page 10: Inserted | Anna Joy Drury | 24/05/2021 20:46:00 |

| Page 10: Inserted | Anna Joy Drury | 24/05/2021 18:31:00 |

When benthic foraminiferal stable isotope records become available for the interval between 17-8 Ma,

| Page 10: Deleted | Anna Joy Drury | 24/05/2021 18:31:00 |

Future work can independently test whether

| Page 10: Inserted | Anna Joy Drury | 24/05/2021 18:32:00 |

stability of the

| Page 10: Deleted | Anna Joy Drury | 24/05/2021 18:30:00 |

early

| Page 10: Inserted | Anna Joy Drury | 24/05/2021 18:30:00 |

late

| Page 10: Deleted | Anna Joy Drury | 24/05/2021 18:30:00 |

derived

| Page 10: Inserted | Anna Joy Drury | 24/05/2021 18:31:00 |

can be tested.

| Page 10: Deleted | Anna Joy Drury | 24/05/2021 18:31:00 |

remains stable until 8 Ma,

| Page 10: Deleted | Anna Joy Drury | 21/03/2021 14:55:00 |

Our

| Page 10: Inserted | Anna Joy Drury | 21/03/2021 14:55:00 |

The CaCO₃ content to eccentricity

| Page 10: Inserted | Anna Joy Drury | 21/03/2021 14:56:00 |

is

| Page 10: Deleted | Anna Joy Drury | 21/03/2021 14:56:00 |

e

| Page 10: Deleted | Anna Joy Drury | 21/03/2021 14:56:00 |

of the ~110 kyr eccentricity cycles are

| Page 10: Inserted | Anna Joy Drury | 21/03/2021 14:56:00 |

is

| Page 10: Deleted | Anna Joy Drury | 21/03/2021 14:56:00 |

more

| Page 10: Inserted | Anna Joy Drury | 21/03/2021 14:56:00 |

The imprint of

| Page 10: Deleted | Anna Joy Drury | 21/03/2021 14:56:00 |

O

| Page 10: Inserted | Anna Joy Drury | 21/03/2021 14:56:00 |

o

| Page 10: Deleted | Anna Joy Drury | 21/03/2021 14:58:00 |

becomes

| Page 10: Inserted | Anna Joy Drury | 21/03/2021 14:58:00 |

is apparent

| Page 10: Deleted | Anna Joy Drury | 21/03/2021 14:56:00 |

more prevalent

| Page 10: Deleted | Anna Joy Drury | 21/03/2021 15:03:00 |

prior to

| Page 10: Inserted | Anna Joy Drury | 21/03/2021 15:03:00 |

before

| Page 10: Inserted | Anna Joy Drury | 24/05/2021 20:46:00 |

La)

| Page 10: Deleted | Anna Joy Drury | 24/05/2021 20:46:00 |

La)

| Page 11: Inserted | Anna Joy Drury | 24/05/2021 20:46:00 |

La)

| Page 11: Deleted | Anna Joy Drury | 24/05/2021 20:46:00 |

La)

| Page 11: Inserted | Anna Joy Drury | 21/03/2021 15:01:00 |

the obliquity solution used in I.b) approach is currently

| Page 11: Inserted | Anna Joy Drury | 21/03/2021 15:01:00 |

available in the

| Page 11: Deleted | Anna Joy Drury | 21/03/2021 15:01:00 |

the

| Page 11: Deleted | Anna Joy Drury | 21/03/2021 15:01:00 |

has the obliquity solution used in I.b) approach

| Page 11: Inserted | Anna Joy Drury | 24/05/2021 18:33:00 |

solution

| Page 11: Deleted | Anna Joy Drury | 21/03/2021 15:01:00 |

has the obliquity solution used in I.b) approach

| Page 11: Inserted | Anna Joy Drury | 21/03/2021 15:01:00 |

There was potential to develop an astrochronology at precession-level, as

| Page 11: Deleted | Anna Joy Drury | 21/03/2021 15:02:00 |

T

| Page 11: Inserted | Anna Joy Drury | 21/03/2021 15:02:00 |

t

| Page 11: Deleted | Anna Joy Drury | 21/03/2021 15:03:00 |

record

| Page 11: Inserted | Anna Joy Drury | 21/03/2021 15:03:00 |

CaCO₃ content

| Page 11: Inserted | Anna Joy Drury | 21/03/2021 15:01:00 |

(see Section 4.1)

| Page 11: Deleted | Anna Joy Drury | 21/03/2021 15:03:00 |

, so there was potential to develop an astrochronology at precession-level

| Page 11: Deleted | Anna Joy Drury | 21/03/2021 15:03:00 |

older than

| Page 11: Inserted | Anna Joy Drury | 21/03/2021 15:03:00 |

before

| Page 11: Inserted | Anna Joy Drury | 21/03/2021 15:04:00 |

chose a

| Page 11: Deleted | Anna Joy Drury | 21/03/2021 15:04:00 |

were

| Page 11: Inserted | Anna Joy Drury | 21/03/2021 15:04:00 |

strategy of

| Page 11: Deleted | Anna Joy Drury | 21/03/2021 15:04:00 |

in

**Page 11: Inserted**     Anna Joy Drury   21/03/2021 15:04:00
only

**Page 11: Deleted**     Anna Joy Drury   21/03/2021 15:04:00
only

**Page 11: Inserted**     Anna Joy Drury   21/03/2021 15:05:00
After 8 Ma, t

**Page 11: Deleted**     Anna Joy Drury   21/03/2021 15:05:00
T

**Page 11: Deleted**     Anna Joy Drury   21/03/2021 15:08:00
strong

**Page 11: Deleted**     Anna Joy Drury   21/03/2021 15:06:00
the %

**Page 11: Deleted**     Anna Joy Drury   21/03/2021 15:06:00
record

**Page 11: Inserted**     Anna Joy Drury   21/03/2021 15:06:00
content

**Page 11: Deleted**     Anna Joy Drury   21/03/2021 15:05:00
is

**Page 11: Inserted**     Anna Joy Drury   21/03/2021 15:05:00
decreases

**Page 11: Inserted**     Anna Joy Drury   21/03/2021 15:06:00
, whilst the imprint of

**Page 11: Deleted**     Anna Joy Drury   21/03/2021 15:05:00
reduced after 8 Ma. Spectral analyses show that

**Page 11: Deleted**     Anna Joy Drury   21/03/2021 15:06:00
are

**Page 11: Inserted**     Anna Joy Drury   21/03/2021 15:06:00
is

**Page 11: Deleted**     Anna Joy Drury   21/03/2021 15:06:00
after 8 Ma until around

**Page 11: Inserted**     Anna Joy Drury   21/03/2021 15:06:00
between 8 and

**Page 11: Inserted**     Anna Joy Drury   21/03/2021 15:07:00

therefore

**Page 11: Inserted**     Anna Joy Drury   21/03/2021 15:08:00
and

**Page 11: Deleted**     Anna Joy Drury   21/03/2021 15:08:00
-

**Page 11: Inserted**     Anna Joy Drury   21/03/2021 15:07:00
to accommodate the change in

**Page 11: Deleted**     Anna Joy Drury   21/03/2021 15:07:00
, because the

**Page 11: Deleted**     Anna Joy Drury   21/03/2021 15:07:00
%

**Page 11: Deleted**     Anna Joy Drury   21/03/2021 15:07:00
data

**Page 11: Inserted**     Anna Joy Drury   21/03/2021 15:07:00
content

**Page 11: Deleted**     Anna Joy Drury   21/03/2021 15:07:00
changes

**Page 11: Deleted**     Anna Joy Drury   21/03/2021 15:10:00
A change in the relationship between t

**Page 11: Inserted**     Anna Joy Drury   21/03/2021 15:10:00
T

**Page 11: Inserted**     Anna Joy Drury   21/03/2021 15:11:00
contrasting relationship between

**Page 11: Deleted**     Anna Joy Drury   21/03/2021 15:10:00
%

**Page 11: Deleted**     Anna Joy Drury   21/03/2021 15:10:00
data

**Page 11: Inserted**     Anna Joy Drury   21/03/2021 15:10:00
content

**Page 11: Deleted**     Anna Joy Drury   21/03/2021 15:12:00
Plio

**Page 11: Inserted**     Anna Joy Drury   21/03/2021 15:12:00
latest Miocene

**Page 11: Deleted**     Anna Joy Drury   21/03/2021 15:11:00
a change in

**Page 11: Inserted**     Anna Joy Drury   21/03/2021 15:11:00
different

**Page 11: Inserted**     Anna Joy Drury   21/03/2021 15:11:00
es

**Page 11: Deleted**     Anna Joy Drury   21/03/2021 15:11:00
is

**Page 11: Inserted**     Anna Joy Drury   21/03/2021 15:12:00
are

**Page 11: Inserted**     Anna Joy Drury   21/03/2021 15:13:00
(6.0-3.3 Ma;

**Page 11: Inserted**     Anna Joy Drury   21/03/2021 15:18:00
latest Miocene-Pleistocene

**Page 11: Deleted**     Anna Joy Drury   21/03/2021 15:17:00
overlapping

**Page 11: Deleted**     Anna Joy Drury   21/03/2021 15:12:00
%

**Page 11: Inserted**     Anna Joy Drury   21/03/2021 15:12:00
content

**Page 11: Deleted**     Anna Joy Drury   21/03/2021 15:18:00
is

**Page 11: Inserted**     Anna Joy Drury   21/03/2021 15:18:00
are both

**Page 11: Inserted**     Anna Joy Drury   21/03/2021 15:15:00
6.0

**Page 11: Deleted**     Anna Joy Drury   21/03/2021 15:15:00
5.3

**Page 11: Deleted**     Anna Joy Drury   21/03/2021 15:17:00
(

**Page 11: Deleted**     Anna Joy Drury   21/03/2021 15:19:00
positive

**Page 11: Inserted**     Anna Joy Drury   21/03/2021 15:19:00
Oligocene-early Miocene

**Page 11: Inserted**     Anna Joy Drury   21/03/2021 15:19:00
between

**Page 11: Deleted**     Anna Joy Drury   21/03/2021 15:19:00
display in the Oligocene-early Miocene

**Page 11: Inserted**     Anna Joy Drury   21/03/2021 15:19:00
late Miocene-Pleistocene

**Page 11: Deleted**     Anna Joy Drury   21/03/2021 15:22:00
for this time interval

**Page 11: Deleted**     Anna Joy Drury   21/03/2021 15:31:00
tuned

**Page 11: Inserted**     Anna Joy Drury   21/03/2021 15:31:00
coinciding with

**Page 11: Inserted**     Anna Joy Drury   21/03/2021 15:31:00
to

**Page 11: Inserted**     Anna Joy Drury   21/03/2021 15:32:00
As

**Page 11: Deleted**     Anna Joy Drury   21/03/2021 15:32:00
Considering

**Page 11: Inserted**     Anna Joy Drury   21/03/2021 15:32:00
relationship between

**Page 11: Deleted**     Anna Joy Drury   21/03/2021 15:32:00
inverse

**Page 11: Deleted**     Anna Joy Drury   21/03/2021 15:32:00
-

**Page 11: Inserted**     Anna Joy Drury   21/03/2021 15:32:00
and

**Page 11: Deleted**     Anna Joy Drury   21/03/2021 15:34:00
relationship

**Page 11: Inserted**     Anna Joy Drury   21/03/2021 15:32:00
is inverse

**Page 11: Deleted**     Anna Joy Drury   21/03/2021 15:32:00
5.3

**Page 11: Inserted**     Anna Joy Drury   21/03/2021 15:32:00

6.0

| Page 11: Deleted | Anna Joy Drury | 21/03/2021 15:33:00 |

interval

| Page 11: Inserted | Anna Joy Drury | 21/03/2021 15:33:00 |

interval

| Page 11: Inserted | Anna Joy Drury | 21/03/2021 15:34:00 |

benthic δ$^{18}$O and CaCO$_3$ content

| Page 11: Deleted | Anna Joy Drury | 21/03/2021 15:34:00 |

datasets

| Page 11: Deleted | Anna Joy Drury | 21/03/2021 15:34:00 |

As such, we

| Page 11: Inserted | Anna Joy Drury | 21/03/2021 15:34:00 |

We therefore

| Page 11: Deleted | Anna Joy Drury | 11/05/2021 14:35:00 |

(i.e., Southern Hemisphere insolation minima) (uncertainty up to ±10 kyr),

| Page 11: Inserted | Anna Joy Drury | 21/03/2021 15:35:00 |

uncertainty up to ±10 kyr;

[revised manuscript text omitted]

T

| Page 13: Deleted | Anna Joy Drury | 12/05/2021 12:34:00 |
|---|---|---|

Liebrand et al. (2016) could not observe a uniform imprint of t

| Page 13: Inserted | Anna Joy Drury | 12/05/2021 12:31:00 |
|---|---|---|

imprint of the

| Page 13: Inserted | Anna Joy Drury | 12/05/2021 12:32:00 |
|---|---|---|

was not constant

| Page 13: Inserted | Anna Joy Drury | 12/05/2021 12:33:00 |
|---|---|---|

(Liebrand et al., 2016)

| Page 13: Inserted | Anna Joy Drury | 21/03/2021 16:12:00 |
|---|---|---|

For periodicities shorter than 405 kyr,

| Page 13: Deleted | Anna Joy Drury | 21/03/2021 16:12:00 |
|---|---|---|

W

| Page 13: Inserted | Anna Joy Drury | 21/03/2021 16:12:00 |
|---|---|---|

w

| Page 13: Inserted | Anna Joy Drury | 21/03/2021 16:12:00 |
|---|---|---|

variability in

| Page 13: Deleted | Anna Joy Drury | 21/03/2021 16:12:00 |
|---|---|---|

short-term %

| Page 13: Inserted | Anna Joy Drury | 21/03/2021 16:12:00 |
|---|---|---|

content

| Page 13: Deleted | Anna Joy Drury | 21/03/2021 16:12:00 |
|---|---|---|

variability

| Page 13: Deleted | Anna Joy Drury | 24/05/2021 20:56:00 |
|---|---|---|

| Page 13: Inserted | Anna Joy Drury | 24/05/2021 20:56:00 |
|---|---|---|

5; Fig 6

| Page 13: Deleted | Anna Joy Drury | 21/03/2021 16:13:00 |
|---|---|---|

prevails as the dominant forcing

| Page 13: Inserted | Anna Joy Drury | 21/03/2021 16:13:00 |
|---|---|---|

is the dominant driver

| Page 13: Inserted | Anna Joy Drury | 29/05/2021 17:23:00 |
|---|---|---|

(Fig 6.A)

| Page 13: Deleted | Anna Joy Drury | 21/03/2021 16:13:00 |
|---|---|---|

,

| Page 13: Inserted | Anna Joy Drury | 21/03/2021 16:14:00 |
|---|---|---|

are the main

| Page 13: Inserted | Anna Joy Drury | 21/03/2021 16:14:00 |
|---|---|---|

r

| Page 13: Deleted | Anna Joy Drury | 21/03/2021 16:14:00 |
|---|---|---|

the main

| Page 13: Inserted | Anna Joy Drury | 21/03/2021 16:14:00 |
|---|---|---|

of

| Page 13: Inserted | Anna Joy Drury | 29/05/2021 17:23:00 |
|---|---|---|

(Fig 6.B)

| Page 14: Deleted | Anna Joy Drury | 21/03/2021 16:14:00 |
|---|---|---|

the

| Page 14: Inserted | Anna Joy Drury | 21/03/2021 16:14:00 |
|---|---|---|

a

| Page 14: Inserted | Anna Joy Drury | 29/05/2021 17:23:00 |
|---|---|---|

(Fig 6.C)

| Page 14: Deleted | Anna Joy Drury | 12/05/2021 12:35:00 |
|---|---|---|

see

| Page 14: Inserted | Anna Joy Drury | 21/03/2021 16:14:00 |
|---|---|---|

in imprinted cyclicity

| Page 14: Deleted | Anna Joy Drury | 21/03/2021 16:17:00 |
|---|---|---|

(see Section 4.1)

| Page 14: Inserted | Anna Joy Drury | 12/05/2021 12:35:00 |
|---|---|---|

that the shifts

| Page 14: Inserted | Anna Joy Drury | 12/05/2021 12:35:00 |
|---|---|---|

they

| Page 14: Inserted | Anna Joy Drury | 21/03/2021 16:17:00 |
|---|---|---|

(see Section 4.1 and Supplementary Figure 9)

| Page 14: Deleted | Anna Joy Drury | 21/03/2021 16:18:00 |
|---|---|---|

the

| Page 14: Inserted | Anna Joy Drury | 21/03/2021 16:18:00 |
|---|---|---|

over the last 30 Myr

| Page 14: Inserted | Anna Joy Drury | 10/05/2021 13:53:00 |
|---|---|---|

At Site 1264,

| Page 14: Inserted | Anna Joy Drury | 10/05/2021 13:54:00 |
|---|---|---|

T

| Page 14: Inserted | Anna Joy Drury | 10/05/2021 13:54:00 |
|---|---|---|

t

| Page 14: Deleted | Anna Joy Drury | 10/05/2021 13:54:00 |
|---|---|---|

at Site 1264

| Page 14: Inserted | Anna Joy Drury | 10/05/2021 13:53:00 |
|---|---|---|

parallels

| Page 14: Deleted | Anna Joy Drury | 10/05/2021 13:53:00 |
|---|---|---|

is in line with

| Page 14: Inserted | Anna Joy Drury | 10/05/2021 13:54:00 |
|---|---|---|

pacing of

| Page 14: Deleted | Anna Joy Drury | 10/05/2021 13:54:00 |
|---|---|---|

variability at Site 1264

**Page 14: Deleted**     Anna Joy Drury    24/05/2021 20:56:00

**Page 14: Inserted**     Anna Joy Drury    24/05/2021 20:56:00
5 and Fig 6.A

**Page 14: Inserted**     Anna Joy Drury    28/05/2021 18:38:00
for further detail see

**Page 14: Inserted**     Anna Joy Drury    28/05/2021 18:31:00
e prevalence of ~110kyr eccentricity pacing at Site 1264

**Page 14: Deleted**     Anna Joy Drury    28/05/2021 18:31:00
is

**Page 14: Deleted**     Anna Joy Drury    12/05/2021 12:36:00
that

**Page 14: Inserted**     Anna Joy Drury    28/05/2021 18:35:00
wider understanding that

**Page 14: Deleted**     Anna Joy Drury    05/05/2021 14:54:00
ice house

**Page 14: Inserted**     Anna Joy Drury    05/05/2021 14:54:00
Coolhouse

**Page 14: Deleted**     Anna Joy Drury    12/05/2021 12:36:00
was

**Page 14: Inserted**     Anna Joy Drury    28/05/2021 18:35:00
was

**Page 14: Inserted**     Anna Joy Drury    28/05/2021 17:59:00
The strong ~110-kyr cyclicity observed in marine archives is attributed to eccentricity-driven changes in ice volume and/or deep-sea temperature, likely associated with changes in atmospheric $CO_2$ (Pälike et al., 2006; Holbourn et al., 2015; Liebrand et al., 2017; Greenop et al., 2019).

**Page 14: Deleted**     Anna Joy Drury    21/03/2021 16:25:00
However, t

**Page 14: Inserted**     Anna Joy Drury    21/03/2021 16:25:00
T

**Page 14: Deleted**     Anna Joy Drury    21/03/2021 16:25:00
with

**Page 14: Inserted**     Anna Joy Drury    21/03/2021 16:25:00
when

**Page 14: Inserted**     Anna Joy Drury    21/03/2021 16:25:00
e

**Page 14: Deleted**     Anna Joy Drury    21/03/2021 16:25:00
ing

**Page 14: Inserted**     Anna Joy Drury    21/03/2021 16:40:00
These precession cycles remain the main driver of carbonate deposition until ~8 Ma, although obliquity cycles are visible

**Page 14: Moved from page 14 (Move #3)**     Anna Joy Drury    21/03/2021 16:40:00
Dduring the 2.4 Myr eccentricity minima from ~12.6- to 12.2 Ma and ~9.7- to 9.3 Ma, when the imprint of precession and ~110 kyr eccentricity imprint is muted (Fig 5 and 6.B), and obliquity paces %$CaCO_3$ variability.

**Page 14: Deleted**     Anna Joy Drury    21/03/2021 16:41:00
D

**Page 14: Inserted**     Anna Joy Drury    21/03/2021 16:41:00
d

**Page 14: Deleted**     Anna Joy Drury    21/03/2021 16:41:00
-

**Page 14: Inserted**     Anna Joy Drury    21/03/2021 16:41:00
to

**Page 14: Deleted**     Anna Joy Drury    21/03/2021 16:41:00
-

**Page 14: Inserted**     Anna Joy Drury    21/03/2021 16:41:00
to

**Page 14: Inserted**     Anna Joy Drury    21/03/2021 16:42:00
when

**Page 14: Inserted**     Anna Joy Drury    21/03/2021 16:42:00
imprint of

**Page 14: Inserted**     Anna Joy Drury    21/03/2021 16:41:00
and ~110 kyr eccentricity

**Page 14: Deleted**     Anna Joy Drury    21/03/2021 16:42:00
imprint

**Page 14: Inserted**     Anna Joy Drury    29/05/2021 17:51:00
(Fig 5 and 6.B)

**Page 14: Deleted**     Anna Joy Drury    21/03/2021 16:42:00
, and obliquity paces %$CaCO_3$ variability

**Page 14: Inserted**     Anna Joy Drury    28/05/2021 18:56:00
Strong obliquity was also observed in benthic $\delta^{18}O$ data from the South China Sea during the ~9.7-9.3 Ma node (Holbourn et al., 2013). The strong obliquity intervals observed across multiple marine archives support that obliquity exerts greater control on the climate system as a whole when the orbital configuration is characterised by long-term eccentricity minima coincident with long-term obliquity maxima

**Page 14: Inserted**     Anna Joy Drury    28/05/2021 19:12:00
(Holbourn et al., 2013, 2018; Drury et al., 2017; Levy et al., 2019).

**Page 14: Inserted**     Anna Joy Drury    21/03/2021 16:43:00
e

**Page 14: Deleted**     Anna Joy Drury    21/03/2021 16:43:00
is

**Page 14: Inserted**     Anna Joy Drury    21/03/2021 16:43:00
to stronger precession pacing

**Page 14: Deleted**     Anna Joy Drury    21/03/2021 16:25:00
are

**Page 14: Inserted**     Anna Joy Drury    21/03/2021 16:25:00
were

**Page 14: Deleted**     Anna Joy Drury    21/03/2021 16:25:00
previously

**Page 14: Inserted**     Anna Joy Drury    12/05/2021 12:36:00
,

**Page 14: Inserted**     Anna Joy Drury    21/03/2021 16:31:00
superimposed on larger ~110 kyr eccentricity cycles,

**Page 14: Deleted**     Anna Joy Drury    21/03/2021 16:31:00
at Site 1264

**Page 14: Deleted**     Anna Joy Drury    21/03/2021 16:26:00
,

**Page 14: Inserted**     Anna Joy Drury    21/03/2021 16:31:00
at Site 1264

**Page 14: Deleted**     Anna Joy Drury    21/03/2021 16:31:00
superimposed on larger ~110 kyr eccentricity cycles

**Page 14: Deleted**     Anna Joy Drury    24/05/2021 20:56:00

**Page 14: Inserted**     Anna Joy Drury    24/05/2021 20:56:00

**Page 14: Inserted**     Anna Joy Drury    21/03/2021 16:29:00
relative amplitude of eccentricity and precession is different in the mid-late Miocene compared to the Oligocene-early Miocene. In the Oligocene-early Miocene, the amplitude of the

**Page 14: Inserted**     Anna Joy Drury    21/03/2021 16:27:00
in $CaCO_3$ content

**Page 14: Deleted**     Anna Joy Drury    21/03/2021 16:26:00
are

**Page 14: Inserted**     Anna Joy Drury    21/03/2021 16:26:00
were

**Page 14: Deleted**     Anna Joy Drury    21/03/2021 16:27:00
in amplitude

**Page 14: Inserted**     Anna Joy Drury    21/03/2021 16:27:00
-driven $CaCO_3$ content

**Page 14: Deleted**     Anna Joy Drury    21/03/2021 16:28:00
concurrent with the increase in precession power in the %$CaCO_3$ data after 14 Ma,

**Page 14: Inserted**     Anna Joy Drury    21/03/2021 16:28:00
concurrent with the strong precession-pacing of the $CaCO_3$ content between 14 and 8 Ma

**Page 14: Deleted**     Anna Joy Drury    24/05/2021 20:56:00

**Page 14: Inserted**     Anna Joy Drury    24/05/2021 20:56:00

**Page 14: Inserted**     Anna Joy Drury    28/05/2021 19:18:00
The influence of early-mid Miocene climate evolution on Southeast Atlantic carbonate deposition is discussed further in Section 5.2.

**Page 14: Moved to page 14 (Move #3)**     Anna Joy Drury    21/03/2021 16:40:00
During the 2.4 Myr eccentricity minima from ~12.6-12.2 Ma and ~9.7-9.3 Ma, the precession imprint is muted, and obliquity paces %$CaCO_3$ variability.

**Page 14: Deleted**     Anna Joy Drury    10/05/2021 13:57:00
on

**Page 14: Inserted**     Anna Joy Drury    10/05/2021 13:57:00
seen in

**Page 14: Deleted**     Anna Joy Drury    21/03/2021 16:48:00

%

**Page 14: Deleted**  Anna Joy Drury  21/03/2021 16:48:00
variability

**Page 14: Inserted**  Anna Joy Drury  21/03/2021 16:48:00
content

**Page 14: Inserted**  Anna Joy Drury  29/05/2021 17:52:00
C

**Page 14: Inserted**  Anna Joy Drury  10/05/2021 13:57:00
of %CaCO₃

**Page 14: Inserted**  Anna Joy Drury  10/05/2021 13:57:00
that we

**Page 14: Inserted**  Anna Joy Drury  10/05/2021 13:58:00
d

**Page 14: Inserted**  Anna Joy Drury  10/05/2021 13:58:00
at Site 1264

**Page 14: Deleted**  Anna Joy Drury  10/05/2021 13:58:00
in the Site 1264 %CaCO₃ record

**Page 15: Inserted**  Anna Joy Drury  28/05/2021 21:02:00
, such as enhanced glacial activity and high-latitude cooling

**Page 15: Deleted**  Anna Joy Drury  28/05/2021 20:55:00
)

**Page 15: Inserted**  Anna Joy Drury  28/05/2021 20:55:00
; see also Section 5.3)

**Page 15: Deleted**  Anna Joy Drury  21/03/2021 16:59:00
5.3

**Page 15: Inserted**  Anna Joy Drury  21/03/2021 16:59:00
6.0

**Page 15: Inserted**  Anna Joy Drury  21/03/2021 16:59:00
6.0

**Page 15: Deleted**  Anna Joy Drury  21/03/2021 16:59:00
5.3

**Page 15: Deleted**  Anna Joy Drury  24/05/2021 20:47:00

**Page 15: Inserted**  Anna Joy Drury  24/05/2021 20:47:00

**Page 15: Inserted**  Anna Joy Drury  21/03/2021 17:00:00
in CaCO₃ content

**Page 15: Deleted**  Anna Joy Drury  21/03/2021 17:00:00
%

**Page 15: Deleted**  Anna Joy Drury  21/03/2021 17:00:00
values

**Page 15: Inserted**  Anna Joy Drury  21/03/2021 17:00:00
content

**Page 15: Inserted**  Anna Joy Drury  10/05/2021 13:59:00
After 3.3 Ma, t

**Page 15: Deleted**  Anna Joy Drury  10/05/2021 13:59:00
T

**Page 15: Deleted**  Anna Joy Drury  10/05/2021 13:59:00
 after 3.3 Ma

**Page 15: Inserted**  Anna Joy Drury  21/03/2021 17:02:00
s

**Page 15: Deleted**  Anna Joy Drury  24/05/2021 20:56:00

**Page 15: Inserted**  Anna Joy Drury  24/05/2021 20:56:00

**Page 15: Inserted**  Anna Joy Drury  05/05/2021 15:11:00
at nearby Site 1267, where it is

**Page 15: Deleted**  Anna Joy Drury  05/05/2021 15:11:00
at nearby Site 1267

**Page 15: Inserted**  Anna Joy Drury  12/05/2021 17:02:00
Alternatively, winnowing may have obscured some of the cyclicity at Site 1264, considering the indication that both Sites 1264 and 525 (both ~2.4-2.5 km water depth) were affected by winnowing in the late Pliocene-early Pleistocene.

**Page 15: Deleted**  Anna Joy Drury  10/05/2021 14:02:00
However

**Page 15: Commented [AJD1]**  Anna Joy Drury  05/05/2021 15:18:00
Improve sentence link.

**Page 15: Inserted**  Anna Joy Drury  10/05/2021 14:02:00

Nonetheless

**Page 15: Inserted**  Anna Joy Drury  24/05/2021 18:39:00
~110

**Page 15: Deleted**  Anna Joy Drury  24/05/2021 18:39:00
100

**Page 15: Deleted**  Anna Joy Drury  24/05/2021 20:57:00

**Page 15: Inserted**  Anna Joy Drury  24/05/2021 20:57:00

**Page 15: Deleted**  Anna Joy Drury  12/05/2021 12:40:00
has been seen

**Page 15: Inserted**  Anna Joy Drury  24/05/2021 18:39:00
~110

**Page 15: Deleted**  Anna Joy Drury  24/05/2021 18:39:00
100

**Page 15: Inserted**  Anna Joy Drury  10/05/2021 14:05:00
6.0

**Page 15: Deleted**  Anna Joy Drury  10/05/2021 14:05:00
5.3

**Page 15: Deleted**  Anna Joy Drury  24/05/2021 20:47:00

**Page 15: Inserted**  Anna Joy Drury  24/05/2021 20:47:00

**Page 15: Deleted**  Anna Joy Drury  29/05/2021 14:32:00

**Page 15: Inserted**  Anna Joy Drury  29/05/2021 14:32:00

**Page 15: Deleted**  Anna Joy Drury  12/05/2021 12:40:00
at

**Page 15: Inserted**  Anna Joy Drury  12/05/2021 12:40:00
to

**Page 15: Deleted**  Anna Joy Drury  10/05/2021 14:05:00
Plio

**Page 15: Inserted**  Anna Joy Drury  10/05/2021 14:05:00
late Miocene

**Page 15: Deleted**  Anna Joy Drury  10/05/2021 14:05:00
and

**Page 15: Inserted**  Anna Joy Drury  10/05/2021 14:05:00
compared to

**Page 15: Inserted**  Anna Joy Drury  10/05/2021 14:06:00
on ~110 kyr periodicities

**Page 15: Deleted**  Anna Joy Drury  10/05/2021 14:06:00
seen

**Page 15: Inserted**  Anna Joy Drury  10/05/2021 14:06:00
observed

**Page 15: Deleted**  Anna Joy Drury  10/05/2021 14:10:00
 Considering the three phases with distinctly different orbital controls on CaCO₃ deposition at Site 1264,

**Page 15: Inserted**  Anna Joy Drury  10/05/2021 14:10:00
I

**Page 15: Deleted**  Anna Joy Drury  10/05/2021 14:10:00
i

**Page 16: Inserted**  Anna Joy Drury  05/05/2021 15:19:00
, with Northern hemisphere high-latitude processes steadily growing in importance in the latest Miocene

**Page 16: Inserted**  Anna Joy Drury  11/05/2021 15:44:00
; De Vleeschouwer et al., 2020

**Page 16: Deleted**  Anna Joy Drury  11/05/2021 14:37:00
icehouse

**Page 16: Deleted**  Anna Joy Drury  29/05/2021 17:40:00
s

**Page 16: Inserted**  Anna Joy Drury  24/05/2021 18:44:00
yet

**Page 16: Deleted**  Anna Joy Drury  12/05/2021 12:41:00
at Site 1264

**Page 16: Deleted**  Anna Joy Drury  10/05/2021 16:37:00
, which display strong ~110 kyr eccentricity pacing,

**Page 16: Deleted**  Anna Joy Drury  10/05/2021 16:37:00
CaCO₃

| Page 16: Deleted | Anna Joy Drury | 24/05/2021 20:57:00 |

6C

| Page 16: Inserted | Anna Joy Drury | 24/05/2021 20:57:00 |

5C

| Page 16: Deleted | Anna Joy Drury | 10/05/2021 18:04:00 |

Little change in

| Page 16: Inserted | Anna Joy Drury | 10/05/2021 18:04:00 |

Low

[revised manuscript text omitted]

| Page 18: Deleted | Anna Joy Drury | 10/05/2021 17:30:00 |
allowing for the

| Page 18: Deleted | Anna Joy Drury | 10/05/2021 17:28:00 |
than

| Page 18: Inserted | Anna Joy Drury | 10/05/2021 17:31:00 |
after the mMCT compared to

| Page 18: Deleted | Anna Joy Drury | 10/05/2021 17:28:00 |
during

| Page 18: Inserted | Anna Joy Drury | 10/05/2021 17:28:00 |
early-

| Page 18: Deleted | Anna Joy Drury | 12/05/2021 11:25:00 |
sedimentary and geochemical

| Page 18: Deleted | Anna Joy Drury | 12/05/2021 11:26:00 |
event is referred to as the

| Page 18: Inserted | Anna Joy Drury | 24/05/2021 18:52:00 |
 as defined by

| Page 18: Deleted | Anna Joy Drury | 24/05/2021 18:53:00 |
and

| Page 18: Inserted | Anna Joy Drury | 24/05/2021 18:53:00 |
s

| Page 18: Deleted | Anna Joy Drury | 24/05/2021 20:58:00 |

| Page 18: Inserted | Anna Joy Drury | 24/05/2021 20:58:00 |

| Page 18: Deleted | Anna Joy Drury | 12/05/2021 11:26:00 |
which means

| Page 18: Inserted | Anna Joy Drury | 12/05/2021 11:26:00 |
so

| Page 19: Deleted | Anna Joy Drury | 11/05/2021 14:59:00 |
at this time

| Page 19: Inserted | Anna Joy Drury | 11/05/2021 14:59:00 |
between 7.2-6.6 Ma

| Page 19: Inserted | Anna Joy Drury | 24/05/2021 20:58:00 |

| Page 19: Deleted | Anna Joy Drury | 24/05/2021 20:58:00 |

| Page 19: Inserted | Anna Joy Drury | 11/05/2021 15:00:00 |
During the latest Miocene-early Pliocene (~8-3 Ma), t

| Page 19: Deleted | Anna Joy Drury | 11/05/2021 15:00:00 |
T

| Page 19: Deleted | Anna Joy Drury | 11/05/2021 15:33:00 |
has

| Page 19: Inserted | Anna Joy Drury | 11/05/2021 15:33:00 |
displays

| Page 19: Deleted | Anna Joy Drury | 11/05/2021 15:00:00 |
during the late Miocene-early Pliocene

| Page 19: Inserted | Anna Joy Drury | 24/05/2021 20:58:00 |

| Page 19: Deleted | Anna Joy Drury | 24/05/2021 20:58:00 |

| Page 19: Deleted | Anna Joy Drury | 11/05/2021 15:00:00 |
 Specifically,

| Page 19: Inserted | Anna Joy Drury | 11/05/2021 15:01:00 |
T

| Page 19: Deleted | Anna Joy Drury | 11/05/2021 15:01:00 |
t

| Page 19: Inserted | Anna Joy Drury | 11/05/2021 15:01:00 |
specifically

| Page 19: Inserted | Anna Joy Drury | 11/05/2021 15:31:00 |
Through the mid-late Miocene and early Pliocene, the LSR at Site 1264 are either similar or higher at Site 1264 (2505 m) relative to deeper Site 1266 (3806 m). The available %CF and %CaCO₃ from Site 1264 also do not display a strong relationship prior to 8 Ma. This suggests that any winnowing at Site 1264 was minimal and stable for the mid-late Miocene to early Pliocene.

| Page 19: Deleted | Anna Joy Drury | 11/05/2021 15:38:00 |
This

| Page 19: Inserted | Anna Joy Drury | 11/05/2021 15:38:00 |
The

| Page 19: Inserted | Anna Joy Drury | 11/05/2021 15:34:00 |
%CF-%CaCO₃

| Page 19: Inserted | Anna Joy Drury | 11/05/2021 15:35:00 |
therefore could

| Page 19: Deleted | Anna Joy Drury | 11/05/2021 15:35:00 |
s

| Page 19: Inserted | Anna Joy Drury | 24/05/2021 20:58:00 |

| Page 19: Deleted | Anna Joy Drury | 24/05/2021 20:58:00 |

| Page 19: Deleted | Anna Joy Drury | 28/05/2021 21:26:00 |
Conversely,

| Page 19: Inserted | Anna Joy Drury | 28/05/2021 21:27:00 |
There is evidence for dynamic ice sheet activity, although

| Page 19: Deleted | Anna Joy Drury | 28/05/2021 21:27:00 |
relative to those seen mid Miocene and Pleistocene, indicating

| Page 19: Inserted | Anna Joy Drury | 28/05/2021 21:57:00 |
suggesting

| Page 19: Inserted | Anna Joy Drury | 28/05/2021 21:57:00 |
long-term

| Page 19: Deleted | Anna Joy Drury | 28/05/2021 21:58:00 |
large

| Page 19: Inserted | Anna Joy Drury | 11/05/2021 15:40:00 |
; Tanner et al., 2020

| Page 20: Inserted | Anna Joy Drury | 11/05/2021 15:56:00 |
 in the Southeast Atlantic

| Page 20: Deleted | Anna Joy Drury | 11/05/2021 15:56:00 |
at Site 1264

| Page 20: Inserted | Anna Joy Drury | 24/05/2021 20:58:00 |

| Page 20: Deleted | Anna Joy Drury | 24/05/2021 20:58:00 |

| Page 20: Inserted | Anna Joy Drury | 28/05/2021 21:36:00 |
shortly after 8 Ma

| Page 20: Inserted | Anna Joy Drury | 28/05/2021 21:35:00 |
climate

| Page 20: Inserted | Anna Joy Drury | 28/05/2021 21:35:00 |
, such as increased glacial activity and high-latitude cooling

| Page 20: Deleted | Anna Joy Drury | 28/05/2021 21:36:00 |
 shortly after 8 Ma

| Page 20: Deleted | Anna Joy Drury | 28/05/2021 22:05:00 |
This increased high-latitude influence may be caused by

| Page 20: Inserted | Anna Joy Drury | 28/05/2021 22:05:00 |
There is widespread evidence that

| Page 20: Deleted | Anna Joy Drury | 28/05/2021 22:06:00 |
, which

| Page 20: Inserted | Anna Joy Drury | 28/05/2021 22:06:00 |

| Page 20: Inserted | Anna Joy Drury | 28/05/2021 21:56:00 |
 The growing importance of the high-latitudes in the latest Miocene is further supported by evidence that deep-sea stable δ¹³C and δ¹⁸O switched from in-phase on anti-phase on eccentricity timescales (Kirtland Turner, 2014; De Vleeschouwer et al., 2020), as a result of continental carbon reservoirs shrinking during cold periods due to increased extent of low-carbon Arctic biomes, such as ice sheets, polar deserts and tundra (De Vleeschouwer et al., 2020).

| Page 20: Inserted | Anna Joy Drury | 28/05/2021 21:38:00 |
,

| Page 20: Inserted | Anna Joy Drury | 28/05/2021 22:08:00 |
 in the latest Miocene

| Page 20: Deleted | Anna Joy Drury | 12/05/2021 12:48:00 |
in explaining

| Page 20: Inserted | Anna Joy Drury | 12/05/2021 12:48:00 |
to explain

| Page 20: Deleted | Anna Joy Drury | 12/05/2021 12:55:00 |
will be

| Page 20: Inserted | Anna Joy Drury | 12/05/2021 12:55:00 |
is

| Page 21: Inserted | Anna Joy Drury | 12/05/2021 12:52:00 |
|---|---|---|

achieve

| Page 21: Deleted | Anna Joy Drury | 12/05/2021 12:52:00 |
|---|---|---|

aim

| Page 21: Inserted | Anna Joy Drury | 12/05/2021 12:52:00 |
|---|---|---|

framework

| Page 21: Deleted | Anna Joy Drury | 12/05/2021 12:53:00 |
|---|---|---|

the

| Page 21: Inserted | Anna Joy Drury | 24/05/2021 18:53:00 |
|---|---|---|

beginning of the

| Page 21: Inserted | Anna Joy Drury | 12/05/2021 12:53:00 |
|---|---|---|

s

| Page 21: Deleted | Anna Joy Drury | 11/05/2021 16:09:00 |
|---|---|---|

cooler,

| Page 21: Inserted | Anna Joy Drury | 12/05/2021 12:54:00 |
|---|---|---|

leading to

| Page 21: Inserted | Anna Joy Drury | 12/05/2021 12:54:00 |
|---|---|---|

ation of

| Page 21: Deleted | Anna Joy Drury | 12/05/2021 12:54:00 |
|---|---|---|

ing

| Page 21: Inserted | Anna Joy Drury | 12/05/2021 12:53:00 |
|---|---|---|

content

| Page 21: Inserted | Anna Joy Drury | 12/05/2021 12:54:00 |
|---|---|---|

s

[revised manuscript text omitted]

------Column Break------

**Page 38: Inserted**   **Anna Joy Drury**  **25/05/2021 14:17:00**

The approximate location of the MCO and the mMCT are also shown.

**Page 38: Inserted**   **Anna Joy Drury**  **25/05/2021 14:16:00**

[Figure]

[Figure]

[Figure]

Header and footer changes
Text Box changes
Header and footer text box changes
Footnote changes
Endnote changes

---

## Author Response (AR2)

**Updated Author Response 05.07.2021**

Dear Luc Beaufort,

Thank you for accepting the publication following minor revision. We made very small modifications to the final files to more accurately describe the data included in the supplementary information. We also noticed that Supplementary Tables 1 and 6 were accidentally missing from the uploaded Supplementary Information.zip, so these are now included.

Best wishes, Anna Joy Drury Submitted on behalf of all co-authors.

**Overview of changes to Drury et al., "Climate, cryosphere and carbon cycle controls on Southeast Atlantic orbital-scale carbonate deposition since the Oligocene (30-0 Ma)"**

Dear Luc Beaufort,

We have completed the requested revisions to our manuscript, as outlined in our author comments in response to the two reviewer comments during the interactive discussion.

The revisions included carefully revisiting the manuscript language, and as such there are numerous small revisions to improve the overall clarity of the manuscript. We have summarised the main revisions to the scientific content here. We have also provided a full list of line-by-line insertions and deletions at the end of this document, and a word track changes file.

The original outcome of the manuscript remains largely unchanged, but we have clarified a few methodological reviewer questions, and expanded our discussed to incorporate some insightful suggestions from the reviewers.

Best wishes, Anna Joy Drury Submitted on behalf of all co-authors.

**Summary of main changes:**

**1) Overall manuscript clarity:**

We have thoroughly proofread through the manuscript to correct previous issues with grammar and spelling. We have also made several revisions to improve the overall clarity of the manuscript. Several relevant publications came out since we submitted the original manuscript (Westerhold et al., 2020, Science; De Vleeschouwer et al., 2020, Nature Communications; Tanner et al., 2020, Paleoceanography and Paleoclimatology), and where relevant, we have included these in our discussion.

**2) Carbonate calibration, calibration uncertainty and treatment of outliers:**

We found a small error in our initial calibration, so we have corrected this. This has resulted in a small change to the calibration; however, this only changes our absolute CaCO3 by less than 0.07% and does not affect our interpretations.

Following a helpful suggestion from the reviewer, we also clarified our outlier treatment process and discussed the uncertainty associated with the calibration and the MARs. The calibration uncertainty is  $\pm 2.2\%$  at  $2\sigma$ . This uncertainty only pertains to the absolute %CaCO3 values. The trends and cyclicity we observe in the calibrated CaCO3 data are independent of this uncertainty, as these patterns are present in the raw ln(Ca/Fe) timeseries. Our interpretations are therefore not affected by this uncertainty.

**3) Improved presentation of the age model and cyclicity observed in the CaCO3 data:**

The reviewers raised concerns that they could not see the cyclicity we were referring to. We have now provided better wavelet figures in the main text and the manuscript, which we feel

highlight the cyclicity better. We have also added a new figure (new figure 6), where we highlight examples of the three main cyclicities discussed in the manuscript.

**4) Expanded discussion to consider winnowing and the processes that may drive the cyclicity we observe in our CaCO3 data:**

We have strengthened our discussion to address reviewers' concerns that to our discussion did not sufficiently consider winnowing or the processes that might drive Site 1264 carbonate.

We now introduce sedimentary processes like winnowing and dilution in the introduction and consider the influence that these processes may have at 1264 throughout the discussion. We conclude that dilution is minimal, and that winnowing may have had some effect, but was likely not the main driver of the trends and cycles we see, with the exception of the last 3.3 Ma.

We also have expanded our discussion in several places to discuss which mechanisms may explain the trends and patterns in carbonate deposition that we observe. We especially focus the discussion on the changes in the previously unstudied interval between 17 and 5.3 Ma. Where appropriate, we have also referred to the original publications dealing with the Oligocene-early Miocene (Liebrand et al., 2016, 2017, 2018) and Plio-Pleistocene (Bell et al., 2014, 2015), as there is already very detailed discussion of these time periods there.

**5) Figures:**

We have made all the revisions requested by reviewers concerning the figures, in addition to the following changes:

- We merged Figures 3 and 4, especially improving the presentation of the wavelets.
- We added the K intensity data to new Figure 3 and Figure 5 (previously Figure 6). Both the Si and K are now also described in the results.
- We present a new figure (new Figure 6) to highlight how the three distinctly different cyclicities in carbonate deposition observed at Site 1264
- We have redone all wavelets presented in the main & supplementary figures to use a more colour-blind friendly scheme.
- We have added epoch and stages to Figures 7 and 8
- We have improved the annotations of Figures 3, 5, 7 and 8.
- Made sedimentation rates m/Myr to be consistent with the main text.
- We have added two supplementary figures:
  - o one showing all calibrated XRF data now available at Site 1264
  - 4-part oversized figure showing the age-depth ties for the astrochronology in greater detail.

Revised\_Drury\_et\_al\_CPD\_Site\_1264\_CaCO3\_30Myr\_(29.05.2021)\_Tracked

| Page 1: Deleted    | Anna Joy Drury    | 11/05/2021 14:38:00                                          |
|--------------------|-------------------|--------------------------------------------------------------|
| Icehouse           |                   |                                                              |
| Down 1. Townshid   | Anna Jau Duum     | 11/05/2021 14:20:00                                          |
| cryosphere         | Anna Joy Drury    | 11/05/2021 14:58:00                                          |
| eryosphere         |                   |                                                              |
| Page 1: Deleted    | Anna Joy Drury    | 11/05/2021 15:56:00                                          |
| over the past 30   | million years (My | r)                                                           |
| Page 1: Deleted    | Anna Joy Drury    | 24/05/2021 14:45:00                                          |
| a                  |                   |                                                              |
| Page 1: Deleted    | Anno Joy Druny    | 24/05/2021 14:45:00                                          |
| a a                | Anna Joy Drury    | 24/03/2021 14:43:00                                          |
| -                  |                   |                                                              |
| Page 1: Deleted    | Anna Joy Drury    | 24/05/2021 14:45:00                                          |
| world              |                   |                                                              |
| Page 1: Inserted   | Anna Joy Drury    | 11/05/2021 15:56:00                                          |
| over the past 30   | million years (My | r)                                                           |
| Page 1: Deleted    | Anna Joy Drury    | 11/05/2021 15:53:00                                          |
| ; however, the ex  | act development   | of orbital-scale climate variability is less well understood |
|                    |                   |                                                              |
| Angel: Deleted     | Anna Joy Drury    | 11/05/2021 15:57:00                                          |
| Aligoia Basili sid | ie of the         |                                                              |
| Page 1: Inserted   | Anna Joy Drury    | 11/05/2021 15:58:00                                          |
| . This             |                   |                                                              |
| Page 1: Deleted    | Anna Joy Drury    | 11/05/2021 15:58:00                                          |
| , which            |                   |                                                              |
| Page 1: Deleted    | Anno Joy Druny    | 11/05/2021 15:59:00                                          |
| site               | Anna SSy Drury    | 1100/2011 200000                                             |
|                    |                   |                                                              |
| Page 1: Inserted   | Anna Joy Drury    | 11/05/2021 15:58:00                                          |
| location           |                   |                                                              |
| Page 1: Deleted    | Anna Joy Drury    | 28/05/2021 22:25:00                                          |
| /or                |                   |                                                              |
| Page 1: Deleted    | Anna Joy Drury    | 28/05/2021 22:27:00                                          |
| strong             |                   |                                                              |
| Page 1: Inserted   | Anna Joy Drury    | 28/05/2021 22:27:00                                          |
| pervasive          |                   |                                                              |
| Page 1: Deleted    | Anna Joy Drury    | 28/05/2021 22:25:00                                          |
| Fage 1. Deleteu    |                   |                                                              |

| the increasing influence              |                                                                                     |
|---------------------------------------|-------------------------------------------------------------------------------------|
| Page 1: Inserted Anna Joy Drury       | 28/05/2021 22:20:00                                                                 |
| greater importance                    |                                                                                     |
| Page 1: Inserted Anna Joy Drury       | 28/05/2021 22:20:00                                                                 |
| , such as increased glacial activity  | and high-latitude cooling                                                           |
| Page 1: Inserted Anna Joy Drury       | 18/03/2021 13:36:00                                                                 |
| s                                     |                                                                                     |
| Page 1: Inserted Anna Joy Drury       | 11/05/2021 15:54:00                                                                 |
| between 14-13 Ma                      |                                                                                     |
| Page 1: Inserted Anna Joy Drury       | 24/05/2021 14:48:00                                                                 |
| recurrent                             |                                                                                     |
| Page 1: Deleted Anna Joy Drury        | 24/05/2021 14:48:00                                                                 |
| an                                    |                                                                                     |
| Page 1: Inserted Anna Joy Drury       | 24/05/2021 14:48:00                                                                 |
| es                                    |                                                                                     |
| Page 1: Inserted Anna Joy Drury       | 11/05/2021 15:50:00                                                                 |
|                                       |                                                                                     |
| Page 1: Deleted Anna Joy Drury        | 11/05/2021 15:50:00                                                                 |
| -                                     |                                                                                     |
| Page 1: Inserted Anna Joy Drury       | 24/05/2021 14:45:00                                                                 |
| -early Pliocene                       |                                                                                     |
| Page 1: Deleted Anna Joy Drury        | 24/05/2021 14:46:00                                                                 |
| , which is                            |                                                                                     |
| Page 1: Inserted Anna Joy Drury       | 24/05/2021 14:46:00                                                                 |
| ;                                     |                                                                                     |
| Page 1: Deleted Anna Joy Drury        | 24/05/2021 14:47:00                                                                 |
| , but not exactly,                    |                                                                                     |
| Page 1: Inserted Anna Joy Drury       | 11/05/2021 15:54:00                                                                 |
| At Site 1264, the onset of the LMBI   | 3 roughly coincides with appearance of strong obliquity pacing of %CaCO3 reflecting |
| increased high latitude forcing.      |                                                                                     |
| Page 1: Deleted Anna Joy Drury        | 28/05/2021 22:29:00                                                                 |
| an                                    |                                                                                     |
| Page 1: Inserted Anna Joy Drury       | 11/05/2021 16:02:00                                                                 |
| , due to enhanced glacial activity ar | d increased meridional temperature gradients                                        |
| Page 2: Deleted Anna Joy Drury        | 17/03/2021 12:17:00                                                                 |
| icehouse                              |                                                                                     |

| Page 2: Inserted   | Anna Joy Drury     | 17/03/2021 12:17:00                                                                 |
|--------------------|--------------------|-------------------------------------------------------------------------------------|
| Coolhouse          |                    |                                                                                     |
|                    |                    |                                                                                     |
| Page 2: Deleted    | Anna Joy Drury     | 17/03/2021 12:17:00                                                                 |
| bipolar            |                    |                                                                                     |
|                    |                    |                                                                                     |
| Page 2: Inserted   | Anna Joy Drury     | 17/03/2021 12:17:00                                                                 |
| Icehouse           |                    |                                                                                     |
| Page 2: Deleted    | Anno Joy Druny     | 24/05/2021 14:49:00                                                                 |
| sheets             | Anna Joy Drary     | 24/05/2021 14:40:00                                                                 |
| succes             |                    |                                                                                     |
| Page 2: Inserted   | Anna Joy Drury     | 24/05/2021 14:48:00                                                                 |
| volumes (Liebrai   | nd et al., 2017)   |                                                                                     |
|                    |                    |                                                                                     |
| Page 2: Deleted    | Anna Joy Drury     | 11/05/2021 14:36:00                                                                 |
| Unipolar           |                    |                                                                                     |
|                    |                    |                                                                                     |
| Page 2: Inserted   | Anna Joy Drury     | 11/05/2021 14:36:00                                                                 |
| unipolar           |                    |                                                                                     |
| Page 2: Deleted    | Anna Joy Drury     | 11/05/2021 14-36-00                                                                 |
| icehouse           | Anna Soy Brary     | 11/03/2021 14/30/00                                                                 |
|                    |                    |                                                                                     |
| Page 2: Inserted   | Anna Joy Drury     | 24/05/2021 13:04:00                                                                 |
| Coolhouse          |                    |                                                                                     |
|                    |                    |                                                                                     |
| Page 2: Deleted    | Anna Joy Drury     | 11/05/2021 14:37:00                                                                 |
| icehouse           |                    |                                                                                     |
| Page 2: Incerted   | Anna Joy Drury     | 11/05/2021 14-37-00                                                                 |
| ruge zi inserteu   | Anna Soy Brary     | 11/03/2021 14:57:00                                                                 |
|                    |                    |                                                                                     |
| Page 2: Deleted    | Anna Joy Drury     | 17/03/2021 12:22:00                                                                 |
| The evolution of   | orbital-scale clim | ate and carbon cycle dynamics across this interval remains relatively unscrutinised |
| (Turner, 2014; D   | e Vleeschouwer e   | t al., 2017).                                                                       |
|                    |                    |                                                                                     |
| Page 2: Deleted    | Anna Joy Drury     | 24/05/2021 16:40:00                                                                 |
| the                |                    |                                                                                     |
|                    |                    |                                                                                     |
| Fage 2: Inserted   | nd                 | 24/05/2021 10:40:00                                                                 |
| Larur 5 clillate a | ilu                |                                                                                     |
| Page 2: Inserted   | Anna Joy Drury     | 17/03/2021 12:25:00                                                                 |
| , as well as sedin | entary processes   | such as winnowing and dilution                                                      |
|                    |                    | -                                                                                   |
| Page 2: Moved to   | page 2 (Move #5    | <ol> <li>Anna Joy Drury 24/05/2021 16:42:00</li> </ol>                              |

Understanding past changes in surface water productivity and deep-sea dissolution can inform about past climate development, and vice versa, how global processes affected regional production and deposition of biogenic carbonates.

| Page 2: Deleted Anna Joy Drury      | 24/05/2021 16:42:00                                                                 |
|-------------------------------------|-------------------------------------------------------------------------------------|
| is a                                |                                                                                     |
| Page 2: Inserted Anna Joy Drury     | 24/05/2021 16:42:00                                                                 |
| are                                 |                                                                                     |
| Page 2: Inserted Anna Joy Drury     | 24/05/2021 16:42:00                                                                 |
| s                                   |                                                                                     |
| Page 2: Inserted Anna Joy Drury     | 28/05/2021 16:24:00                                                                 |
| lysocline and                       |                                                                                     |
| Page 2: Inserted Anna Joy Drury     | 28/05/2021 16:28:00                                                                 |
| at greater depths and /or           | · · · ·                                                                             |
| Page 2: Inserted Anna Joy Drury     | 24/05/2021 16:43:00                                                                 |
| Changes in deep-sea currents can a  | lter the composition of the sediment through processes like winnowing or dilution,  |
| which respectively remove fine-gr   | ained material or increase certain sedimentary components relative to others (e.g., |
| increased dilution with terrigenous | material)                                                                           |
| mercused unation what terrigenous   | initerial).                                                                         |
| Page 2: Moved from page 2 (Move     | #5) Anna Joy Drury 24/05/2021 16:42:00                                              |
| Understanding past changes in car   | bonate deposition surface water productivity and deep-sea dissolution can inform    |
| about past climate development b    | y helping to disentangle , and vice versa, how global processes affected regional   |
| production and deposition of bioge  | enic carbonates.                                                                    |
| Page 2: Inserted Anna Joy Drury     | 24/05/2021 16:45:00                                                                 |
| carbonate deposition                |                                                                                     |
| Page 2: Deleted Anna Joy Drury      | 24/05/2021 16:48:00                                                                 |
| surface water productivity and dee  | p-sea dissolution                                                                   |
|                                     | ·                                                                                   |
| Page 2: Inserted Anna Joy Drury     | 24/05/2021 16:48:00                                                                 |
| by helping to disentangle           |                                                                                     |
| Page 2: Deleted Anna Joy Drury      | 24/05/2021 16:48:00                                                                 |
| , and vice versa,                   |                                                                                     |
| Page 2: Inserted Anna Joy Drury     | 24/05/2021 15:03:00                                                                 |
| ,                                   |                                                                                     |
| Page 2: Deleted Anna Joy Drury      | 24/05/2021 15:03:00                                                                 |
| both                                |                                                                                     |
| Page 2: Deleted Anna Joy Drury      | 24/05/2021 15:04:00                                                                 |
|                                     |                                                                                     |

| strategic |
|-----------|
|-----------|

| Page 2: Inserted  | Anna Joy Drury   | 24/05/2021 15:03:00 |  |
|-------------------|------------------|---------------------|--|
| was affected by   |                  |                     |  |
|                   |                  |                     |  |
| Page 2: Deleted   | Anna Joy Drury   | 24/05/2021 15:04:00 |  |
| has the potential | to record        |                     |  |
|                   |                  |                     |  |
| Page 2: Deleted   | Anna Joy Drury   | 24/05/2021 15:04:00 |  |
| changes           |                  |                     |  |
|                   |                  |                     |  |
| Page 2: Inserted  | Anna Joy Drury   | 24/05/2021 15:04:00 |  |
| conditions        |                  |                     |  |
| Page 3: Incerted  | Anna lov Drury   | 11/05/2021 15:57:00 |  |
| the Angola Basi   | n side of        |                     |  |
|                   |                  |                     |  |
| Page 3: Deleted   | Anna Joy Drury   | 28/05/2021 16:45:00 |  |
| se                |                  |                     |  |
|                   |                  |                     |  |
| Page 3: Inserted  | Anna Joy Drury   | 17/03/2021 12:34:00 |  |
| resulting         |                  |                     |  |
|                   |                  |                     |  |
| Page 3: Deleted   | Anna Joy Drury   | 28/05/2021 16:53:00 |  |
| , astronomically  | tuned carbonate  |                     |  |
| Page 3: Deleted   | Anna Joy Drury   | 28/05/2021 16:49:00 |  |
| allow us to       |                  |                     |  |
|                   |                  |                     |  |
| Page 3: Inserted  | Anna Joy Drury   | 28/05/2021 16:49:00 |  |
| help determine sl | hifts in         |                     |  |
|                   |                  |                     |  |
| Page 3: Deleted   | Anna Joy Drury   | 28/05/2021 16:42:00 |  |
| investigate how   |                  |                     |  |
| Page 3: Deleted   | Anna Joy Drury   | 28/05/2021 16:43:00 |  |
| se                | Anna Joy Drury   | 20,00/2021 10.43:00 |  |
|                   |                  |                     |  |
| Page 3: Deleted   | Anna Joy Drury   | 28/05/2021 16:48:00 |  |
| regimes           |                  |                     |  |
|                   |                  |                     |  |
| Page 3: Inserted  | Anna Joy Drury   | 28/05/2021 16:43:00 |  |
| of Southeast Atla | intic CaCO3 depo | sition in           |  |
|                   |                  |                     |  |
| Page 3: Inserted  | Anna Joy Drury   | 28/05/2021 16:43:00 |  |
| ion               |                  |                     |  |
| Page 3: Deleted   | Anna Joy Druss   | 28/05/2021 16:43:00 |  |
| I AND DI DEIELEU  |                  |                     |  |

| Page 3: Inserted   | Anna Joy Drury 28/05/2021 16:46:00                                                                   |
|--------------------|------------------------------------------------------------------------------------------------------|
| We investigate h   | iow widespread Miocene warmth followed by Antarctic glaciation influenced the pacing an              |
| preservation of So | utheast Atlantic carbonate deposition. Finally, we establish the relative timing of the late Miocene |
| early Pliocene Bio | ogenic Bloom (LMBB; acronym from Lyle et al., 2019) in the Southeast Atlantic versus Pacifi          |
| Oceans and explo   | ore what this reveals about the global and regional driving forces of this multi-million-year        |
| productivity event |                                                                                                      |
|                    |                                                                                                      |
| Page 3: Deleted    | Anna Joy Drury 17/03/2021 15:38:00                                                                   |

| composite section   | n                |                     |  |
|---------------------|------------------|---------------------|--|
| Page 3: Inserted    | Anna Joy Drury   | 17/03/2021 15:38:00 |  |
| stratigraphic splic | ce was developed |                     |  |
| Page 3: Deleted     | Anna Joy Drury   | 17/03/2021 15:38:00 |  |
| was developed       |                  |                     |  |
| Page 3: Deleted     | Anna Joy Drury   | 29/05/2021 17:32:00 |  |
| was                 |                  |                     |  |
| Page 3: Inserted    | Anna Joy Drury   | 29/05/2021 17:32:00 |  |
| were                |                  |                     |  |
| Page 3: Inserted    | Anna Joy Drury   | 18/03/2021 13:37:00 |  |
| an                  |                  |                     |  |
| Page 3: Deleted     | Anna Joy Drury   | 29/05/2021 17:32:00 |  |
| are                 |                  |                     |  |
| Page 3: Inserted    | Anna Joy Drury   | 29/05/2021 17:32:00 |  |
| were exceptional    | ly               |                     |  |
| Page 3: Deleted     | Anna Joy Drury   | 24/05/2021 15:45:00 |  |
| 21                  |                  |                     |  |
| Page 3: Inserted    | Anna Joy Drury   | 24/05/2021 15:45:00 |  |
| 19                  |                  |                     |  |
| Page 3: Inserted    | Anna Joy Drury   | 24/05/2021 15:50:00 |  |
| ; LSR =             |                  |                     |  |
| Page 3: Deleted     | Anna Joy Drury   | 24/05/2021 15:50:00 |  |
| ,                   |                  |                     |  |
| Page 3: Deleted     | Anna Joy Drury   | 24/05/2021 15:42:00 |  |
| 1                   |                  |                     |  |
| Page 3: Inserted    | Anna Joy Drury   | 24/05/2021 15:43:00 |  |
| 3.9                 |                  |                     |  |

| Page 3: Deleted    | Anna Joy Drury      | 24/05/2021 15:42:00                                   |
|--------------------|---------------------|-------------------------------------------------------|
| 8                  |                     |                                                       |
| Page 3: Inserted   | Anna Joy Drury      | 24/05/2021 15:48:00                                   |
| 5.4                |                     |                                                       |
| Page 3: Inserted   | Anna Joy Drury      | 24/05/2021 15:51:00                                   |
| , average of 4.7 1 | n/Myr               |                                                       |
| Page 3: Deleted    | Anna Joy Drury      | 24/05/2021 15:50:00                                   |
| and early Plio-P   | leistocene interval | ls (3-0 Ma, 4-8 m/Myr)                                |
| Page 3: Inserted   | Anna Joy Drury      | 17/03/2021 12:38:00                                   |
| Higher shipboard   | 1                   |                                                       |
| Page 3: Deleted    | Anna Joy Drury      | 17/03/2021 12:38:00                                   |
| are higher         |                     |                                                       |
| Page 3: Inserted   | Anna Joy Drury      | 17/03/2021 12:38:00                                   |
| occurred           |                     |                                                       |
| Page 3: Inserted   | Anna Joy Drury      | 24/05/2021 15:49:00                                   |
| 19                 |                     |                                                       |
| Page 3: Deleted    | Anna Joy Drury      | 24/05/2021 15:49:00                                   |
| 21                 |                     |                                                       |
| Page 3: Inserted   | Anna Joy Drury      | 24/05/2021 15:51:00                                   |
| ; LSR = 5.3-9.3 i  | n/Myr, average of   | 7.1m/Myr                                              |
| Page 3: Deleted    | Anna Joy Drury      | 17/03/2021 12:38:00                                   |
| , 6-22 m/Myr       |                     |                                                       |
| Page 3: Inserted   | Anna Joy Drury      | 24/05/2021 15:51:00                                   |
| and early Plio-Pl  | eistocene (3-0 Ma   | ; LSR = $4.5-7.4$ m/Myr, average of $6.0$ m/Myr). The |
| Page 3: Deleted    | Anna Joy Drury      | 24/05/2021 15:52:00                                   |
| and are            |                     |                                                       |
| Page 3: Inserted   | Anna Joy Drury      | 24/05/2021 15:52:00                                   |
| shipboard LSR o    | ccurred             |                                                       |
| Page 3: Inserted   | Anna Joy Drury      | 24/05/2021 15:52:00                                   |
| LSR average 15.    | 9 m/Myr             |                                                       |
| Page 3: Deleted    | Anna Joy Drury      | 24/05/2021 15:52:00                                   |
| they range betwee  | en                  |                                                       |
| Page 3: Inserted   | Anna Joy Drury      | 24/05/2021 15:52:00                                   |
| (7.7-30.5          |                     |                                                       |

| age 3: Deleted Anna Joy Drury         | 24/05/2021 15:53:00         |                     |  |
|---------------------------------------|-----------------------------|---------------------|--|
| 3 and 31                              |                             |                     |  |
| Page 3: Inserted Anna Joy Drury       | 17/03/2021 12:38:00         |                     |  |
| shipboard LSR for the                 |                             |                     |  |
| 4                                     |                             |                     |  |
| Page 3: Deleted Anna Joy Drury        | 17/03/2021 12:38:00         |                     |  |
| shipboard LSR                         |                             |                     |  |
| Page 3: Moved to page 4 (Move #4      | ) Anna Joy Drury            | 24/05/2021 15:57:00 |  |
| (Liebrand et al., 2011, 2016, 2017,   | 2018; Bell et al., 2014, 20 | 15)                 |  |
|                                       |                             |                     |  |
| Page 4: Moved from page 3 (Move       | #4) Anna Joy Drury          | 24/05/2021 15:57:00 |  |
| (Liebrand et al., 2011, 2016, 2017,   | 2018; Bell et al., 2014, 20 | 15)                 |  |
| Page 4: Inserted Anna Jov Drurv       | 17/03/2021 15:39:00         |                     |  |
| composite                             |                             |                     |  |
|                                       |                             |                     |  |
| Page 4: Deleted Anna Joy Drury        | 17/03/2021 15:39:00         |                     |  |
| composite                             |                             |                     |  |
| Page 4: Deleted Anna Joy Drury        | 17/03/2021 13:52:00         |                     |  |
| each                                  | ,,                          |                     |  |
|                                       |                             |                     |  |
| Page 4: Inserted Anna Joy Drury       | 17/03/2021 13:52:00         |                     |  |
| s                                     |                             |                     |  |
| Page 4: Deleted Anna Joy Drury        | 17/03/2021 13:52:00         |                     |  |
| was                                   | 17,05,2022 15:52:00         |                     |  |
|                                       |                             |                     |  |
| Page 4: Inserted Anna Joy Drury       | 17/03/2021 13:52:00         |                     |  |
| were                                  |                             |                     |  |
| Page 4 Deleted Area las Deser         | 17/02/2021 12:20:00         |                     |  |
| composite                             | 17/03/2021 12:39:00         |                     |  |
| 1                                     |                             |                     |  |
| Page 4: Inserted Anna Joy Drury       | 17/03/2021 12:39:00         |                     |  |
| single                                |                             |                     |  |
| Page 4: Deleted Anna Joy Drugs        | 17/03/2021 13:53:00         |                     |  |
| core                                  | 1,00,2021 13.33.00          |                     |  |
|                                       |                             |                     |  |
| Page 4: Inserted Anna Joy Drury       | 17/03/2021 13:53:00         |                     |  |
| for each core                         |                             |                     |  |
| Dama 4. Jacouted Anna In. C           | 17/03/3031 14:05 00         |                     |  |
| " " " " " " " " " " " " " " " " " " " | 17/03/2021 14:05:00         |                     |  |
|                                       |                             |                     |  |
| Page 4: Inserted Anna Joy Drury       | 17/03/2021 14:05:00         |                     |  |
|                                       |                             |                     |  |

| Page 4: Inserted                                | Anna Joy Drury                   | 17/03/2021 13:53:00                        |
|-------------------------------------------------|----------------------------------|--------------------------------------------|
| -                                               |                                  |                                            |
|                                                 |                                  |                                            |
|                                                 |                                  |                                            |
| Page 4: Deleted                                 | Anna Joy Drury                   | 17/03/2021 13:53:00                        |
|                                                 |                                  |                                            |
|                                                 |                                  |                                            |
| Dama A: Dalahad                                 | Arres Jaw Denne                  | 17/02/2021 12-56-00                        |
| Page 4: Deleted                                 | Anna Joy Drury                   | 17/03/2021 13:56:00                        |
| composite                                       |                                  |                                            |
|                                                 |                                  |                                            |
| Page 4: Incerted                                | Anna Joy Drury                   | 17/03/2021 13:56:00                        |
| individual ages                                 |                                  |                                            |
| individual core                                 |                                  |                                            |
|                                                 |                                  |                                            |
| Page 4: Deleted                                 | Anna Joy Drury                   | 17/03/2021 13:57:00                        |
| core                                            |                                  |                                            |
|                                                 |                                  |                                            |
|                                                 |                                  |                                            |
| Page 4: Inserted                                | Anna Joy Drury                   | 17/03/2021 13:56:00                        |
| -                                               |                                  |                                            |
|                                                 |                                  |                                            |
| Dama de Tanas 1. 1                              | Anna Jaw Da                      | 17/02/2021 12-56-00                        |
| Page 4: Inserted                                | Anna Joy Drury                   | 1//03/2021 13:56:00                        |
| core-box                                        |                                  |                                            |
|                                                 |                                  |                                            |
| Page 4: Deleted                                 | Anna Joy Druny                   | 17/02/2021 12:57:00                        |
| Fage 4. Deleteu                                 | Anna Joy Drury                   | 17/05/2021 15:57:00                        |
| CODD (                                          |                                  |                                            |
|                                                 |                                  |                                            |
| Page 4: Inserted                                | Anna Joy Drury                   | 17/03/2021 13:57:00                        |
| the "                                           |                                  |                                            |
| uic                                             |                                  |                                            |
|                                                 |                                  |                                            |
| Page 4: Inserted                                | Anna Joy Drury                   | 17/03/2021 14:05:00                        |
| "                                               |                                  |                                            |
|                                                 |                                  |                                            |
|                                                 |                                  |                                            |
| Page 4: Deleted                                 | Anna Joy Drury                   | 1//03/2021 13:57:00                        |
| )                                               |                                  |                                            |
|                                                 |                                  |                                            |
| Page 4: Deleted                                 | Anna lov Dress                   | 17/03/2021 13:57:00                        |
| · uge - Deleteu                                 | Annu Soy Drury                   | 1/05/2022 25:57:00                         |
| core                                            |                                  |                                            |
|                                                 |                                  |                                            |
| Page 4: Inserted                                | Anna Joy Drury                   | 17/03/2021 13:57:00                        |
|                                                 | ,,                               |                                            |
| -                                               |                                  |                                            |
|                                                 |                                  |                                            |
| Page 4: Deleted                                 | Anna Joy Drury                   | 17/03/2021 13:57:00                        |
| composite                                       |                                  |                                            |
| mposito                                         |                                  |                                            |
|                                                 |                                  |                                            |
|                                                 |                                  |                                            |
| Page 4: Inserted                                | Anna Joy Drury                   | 17/03/2021 13:57:00                        |
| Page 4: Inserted                                | Anna Joy Drury                   | 17/03/2021 13:57:00                        |
| Page 4: Inserted                                | Anna Joy Drury                   | 17/03/2021 13:57:00                        |
| Page 4: Inserted                                | Anna Joy Drury                   | 17/03/2021 13:57:00                        |
| Page 4: Inserted
core-box
Page 4: Deleted | Anna Joy Drury
Anna Joy Drury | 17/03/2021 13:57:00
17/03/2021 13:57:00 |

**33**

| Page 4: Inserted     | Anna Joy Drury     | 17/03/2021 13:57:00                                                            |
|----------------------|--------------------|--------------------------------------------------------------------------------|
| represent            |                    |                                                                                |
| Page 4: Incorted     | Anno Joy Druny     | 17/02/2021 12-58-00                                                            |
| -                    | Anna Joy Drury     | 17/05/2021 15:59:00                                                            |
|                      |                    |                                                                                |
| Page 4: Deleted      | Anna Joy Drury     | 17/03/2021 13:59:00                                                            |
|                      |                    |                                                                                |
| Page 4: Deleted      | Anna Joy Drury     | 17/03/2021 13:59:00                                                            |
| -                    |                    |                                                                                |
| Page 4: Inserted     | Anna Joy Drury     | 17/03/2021 13:59:00                                                            |
| -                    |                    |                                                                                |
| Page 4: Deleted      | Anna Joy Drury     | 17/03/2021 13:59:00                                                            |
| composite            | Anna soy brary     | 17/05/2021 15:55:00                                                            |
|                      |                    |                                                                                |
| Page 4: Inserted     | Anna Joy Drury     | 17/03/2021 14:02:00                                                            |
| preferentially       |                    |                                                                                |
| Page 4: Deleted      | Anna Joy Drury     | 17/03/2021 13:59:00                                                            |
| composite            |                    |                                                                                |
| Page 4: Inserted     | Anna Joy Drury     | 17/03/2021 14:00:00                                                            |
| ,                    |                    |                                                                                |
| Page 4: Deleted      | Anna Joy Drury     | 17/03/2021 14:00:00                                                            |
| they visually hig    | hlight             |                                                                                |
|                      |                    |                                                                                |
| Page 4: Deleted      | Anna Joy Drury     | 17/03/2021 14:00:00                                                            |
| better               |                    |                                                                                |
| Page 4: Inserted     | Anna Joy Drury     | 17/03/2021 14:00:00                                                            |
| is better visible in | n these images     |                                                                                |
| Page 4: Deleted      | Anna Joy Drury     | 17/03/2021 15:39:00                                                            |
| composite            | ,                  |                                                                                |
| Page 4: Incorted     | Anna Joy Drees     | 17/03/2021 15:39:00                                                            |
| stratigraphic        | Anna Joy Diury     | 1, 00, LOLL 10,05,00                                                           |
| Subrapine            |                    |                                                                                |
| Page 4: Inserted     | Anna Joy Drury     | 17/03/2021 14:03:00                                                            |
| The individual c     | ore-box/line-scan  | core images were then combined into a single composite image along the revised |
| Site 1264 splice     | using the "SpliceI | mages" function.                                                               |
| Page 4: Deleted      | Anna Joy Drury     | 17/03/2021 14:03:00                                                            |
| composite            |                    |                                                                                |

| Page 4: Inserted Anna Joy Dr                                                                                                                                                                                                                                                                                                                                                                             | ury 17/03/2021 14:03:00                                                                                                                                                                                                                                                                                                                                                                                                                                                    |
|----------------------------------------------------------------------------------------------------------------------------------------------------------------------------------------------------------------------------------------------------------------------------------------------------------------------------------------------------------------------------------------------------------|----------------------------------------------------------------------------------------------------------------------------------------------------------------------------------------------------------------------------------------------------------------------------------------------------------------------------------------------------------------------------------------------------------------------------------------------------------------------------|
| individual                                                                                                                                                                                                                                                                                                                                                                                               |                                                                                                                                                                                                                                                                                                                                                                                                                                                                            |
| Page 4: Inserted Anna Joy Dr                                                                                                                                                                                                                                                                                                                                                                             | ury 17/03/2021 14:02:00                                                                                                                                                                                                                                                                                                                                                                                                                                                    |
| -                                                                                                                                                                                                                                                                                                                                                                                                        |                                                                                                                                                                                                                                                                                                                                                                                                                                                                            |
| Page 4: Deleted Anna Joy Dr                                                                                                                                                                                                                                                                                                                                                                              | ury 17/03/2021 14:02:00                                                                                                                                                                                                                                                                                                                                                                                                                                                    |
| •                                                                                                                                                                                                                                                                                                                                                                                                        |                                                                                                                                                                                                                                                                                                                                                                                                                                                                            |
| Page 4: Inserted Anna Joy Dr                                                                                                                                                                                                                                                                                                                                                                             | ury 17/03/2021 14:02:00                                                                                                                                                                                                                                                                                                                                                                                                                                                    |
| -                                                                                                                                                                                                                                                                                                                                                                                                        |                                                                                                                                                                                                                                                                                                                                                                                                                                                                            |
| Page 4: Deleted Anna Joy Dr                                                                                                                                                                                                                                                                                                                                                                              | ury 17/03/2021 14:02:00                                                                                                                                                                                                                                                                                                                                                                                                                                                    |
|                                                                                                                                                                                                                                                                                                                                                                                                          |                                                                                                                                                                                                                                                                                                                                                                                                                                                                            |
| Page 4: Inserted Anna Joy Dr                                                                                                                                                                                                                                                                                                                                                                             | ury 17/03/2021 14:09:00                                                                                                                                                                                                                                                                                                                                                                                                                                                    |
| in the 1264 splice image                                                                                                                                                                                                                                                                                                                                                                                 |                                                                                                                                                                                                                                                                                                                                                                                                                                                                            |
| Page 4: Deleted Anna Joy Dr                                                                                                                                                                                                                                                                                                                                                                              | ury 17/03/2021 14:10:00                                                                                                                                                                                                                                                                                                                                                                                                                                                    |
| to form a continuous composi                                                                                                                                                                                                                                                                                                                                                                             | te core image spanning the early Oligocene to present day                                                                                                                                                                                                                                                                                                                                                                                                                  |
| Page 4: Deleted Anna Joy Dr                                                                                                                                                                                                                                                                                                                                                                              | ury 17/03/2021 14:10:00                                                                                                                                                                                                                                                                                                                                                                                                                                                    |
| ,                                                                                                                                                                                                                                                                                                                                                                                                        |                                                                                                                                                                                                                                                                                                                                                                                                                                                                            |
|                                                                                                                                                                                                                                                                                                                                                                                                          |                                                                                                                                                                                                                                                                                                                                                                                                                                                                            |
| Page 4: Inserted Anna Joy Dr                                                                                                                                                                                                                                                                                                                                                                             | ury 17/03/2021 14:10:00                                                                                                                                                                                                                                                                                                                                                                                                                                                    |
| Page 4: Inserted Anna Joy Dr
This resulted in a continuou                                                                                                                                                                                                                                                                                                                                             | ury 17/03/2021 14:10:00
spliced image of the sedimentary succession at Site 1264-1265 spanning the early                                                                                                                                                                                                                                                                                                                                                                |
| Page 4: Inserted Anna Joy Dr
This resulted in a continuou
Oligocene to present day.                                                                                                                                                                                                                                                                                                                | ury 17/03/2021 14:10:00
spliced image of the sedimentary succession at Site 1264-1265 spanning the early                                                                                                                                                                                                                                                                                                                                                                |
| Page 4: Inserted Anna Joy Dr
This resulted in a continuou
Oligocene to present day.
Page 4: Deleted Anna Joy Dr                                                                                                                                                                                                                                                                                 | ury 17/03/2021 14:10:00
s spliced image of the sedimentary succession at Site 1264-1265 spanning the early
ury 17/03/2021 14:28:00                                                                                                                                                                                                                                                                                                                                   |
| Page 4: Inserted Anna Joy Dr           This resulted in a continuou         Oligocene to present day.           Page 4: Deleted Anna Joy Dr         195                                                                                                                                                                                                                                                  | ury 17/03/2021 14:10:00
spliced image of the sedimentary succession at Site 1264-1265 spanning the early
ury 17/03/2021 14:28:00                                                                                                                                                                                                                                                                                                                                     |
| Page 4: Inserted Anna Joy Dr         This resulted in a continuou         Oligocene to present day.         Page 4: Deleted Anna Joy Dr         195         Page 4: Inserted Anna Joy Dr                                                                                                                                                                                                                 | rry         17/03/2021 14:10:00           spliced image of the sedimentary succession at Site 1264-1265 spanning the early           ury         17/03/2021 14:28:00           ury         17/03/2021 14:28:00                                                                                                                                                                                                                                                             |
| Page 4: Inserted Anna Joy Dr
This resulted in a continuou
Oligocene to present day.
Page 4: Deleted Anna Joy Dr
195
Page 4: Inserted Anna Joy Dr
205                                                                                                                                                                                                                                   | ury         17/03/2021 14:10:00           spliced image of the sedimentary succession at Site 1264-1265 spanning the early           ury         17/03/2021 14:28:00                                                                                                                                                                                                                                                                                                       |
| Page 4: Inserted Anna Joy Dr         This resulted in a continuou         Oligocene to present day.         Page 4: Deleted Anna Joy Dr         195         Page 4: Inserted Anna Joy Dr         205         Page 4: Deleted Anna Joy Dr         205                                                                                                                                                     | ury         17/03/2021 14:10:00           spliced image of the sedimentary succession at Site 1264-1265 spanning the early           ury         17/03/2021 14:28:00           ury         17/03/2021 14:28:00           ury         17/03/2021 14:28:00                                                                                                                                                                                                                   |
| Page 4: Inserted Anna Joy Dr
This resulted in a continuou
Oligocene to present day.
Page 4: Deleted Anna Joy Dr
195
Page 4: Inserted Anna Joy Dr
205
Page 4: Deleted Anna Joy Dr
195                                                                                                                                                                                             | ury         17/03/2021 14:10:00           spliced image of the sedimentary succession at Site 1264-1265 spanning the early           ury         17/03/2021 14:28:00           ury         17/03/2021 14:28:00           ury         17/03/2021 14:28:00                                                                                                                                                                                                                   |
| Page 4: Inserted Anna Joy Dr
This resulted in a continuou
Oligocene to present day.
Page 4: Deleted Anna Joy Dr
195
Page 4: Deleted Anna Joy Dr
205
Page 4: Deleted Anna Joy Dr
195
Page 4: Deleted Anna Joy Dr
195                                                                                                                                                        | ury         17/03/2021 14:10:00           spliced image of the sedimentary succession at Site 1264-1265 spanning the early           ury         17/03/2021 14:28:00                                                                                                                               |
| Page 4: Inserted Anna Joy Dr
This resulted in a continuou
Oligocene to present day.
Page 4: Deleted Anna Joy Dr
195
Page 4: Deleted Anna Joy Dr
205
Page 4: Deleted Anna Joy Dr
195
Page 4: Inserted Anna Joy Dr
205                                                                                                                                                       | ury         17/03/2021 14:10:00           spliced image of the sedimentary succession at Site 1264-1265 spanning the early           ury         17/03/2021 14:28:00           ury         17/03/2021 14:28:00           ury         17/03/2021 14:29:00           ury         17/03/2021 14:29:00                                                                                                                                                                         |
| Page 4: Inserted Anna Joy Dr
This resulted in a continuou
Oligocene to present day.
Page 4: Deleted Anna Joy Dr
105
Page 4: Inserted Anna Joy Dr
205
Page 4: Deleted Anna Joy Dr
195
Page 4: Inserted Anna Joy Dr
205
Page 4: Inserted Anna Joy Dr
205                                                                                                               | ury         17/03/2021 14:10:00           spliced image of the sedimentary succession at Site 1264-1265 spanning the early           ury         17/03/2021 14:28:00           ury         17/03/2021 14:29:00           ury         17/03/2021 14:29:00           ury         17/03/2021 14:29:00           ury         17/03/2021 14:29:00                                                                                                                               |
| Page 4: Inserted Anna Joy Dr
This resulted in a continuou
Oligocene to present day.
Page 4: Deleted Anna Joy Dr
205
Page 4: Inserted Anna Joy Dr
205
Page 4: Deleted Anna Joy Dr
Was                                                                        | ury         17/03/2021 14:10:00           spliced image of the sedimentary succession at Site 1264-1265 spanning the early           ury         17/03/2021 14:28:00           ury         17/03/2021 14:29:00           ury         17/03/2021 14:29:00           ury         17/03/2021 14:29:00           ury         17/03/2021 14:29:00           ury         29/05/2021 14:29:00                                                                                     |
| Page 4: Inserted Anna Joy Dr         This resulted in a continuou         Oligocene to present day.         Page 4: Deleted Anna Joy Dr         195         Page 4: Deleted Anna Joy Dr         205         Page 4: Deleted Anna Joy Dr         205         Page 4: Deleted Anna Joy Dr         205         Page 4: Deleted Anna Joy Dr         Was         Page 4: Inserted Anna Joy Dr                 | ury       17/03/2021 14:10:00         spliced image of the sedimentary succession at Site 1264-1265 spanning the early         ury       17/03/2021 14:28:00         ury       17/03/2021 14:28:00         ury       17/03/2021 14:29:00         ury       17/03/2021 14:29:00         ury       17/03/2021 14:29:00         ury       29/05/2021 17:33:00                                                                                                                 |
| Page 4: Inserted Anna Joy Dr         This resulted in a continuou         Oligocene to present day.         Page 4: Deleted Anna Joy Dr         195         Page 4: Deleted Anna Joy Dr         195         Page 4: Deleted Anna Joy Dr         205         Page 4: Deleted Anna Joy Dr         205         Page 4: Deleted Anna Joy Dr         Was         Page 4: Inserted Anna Joy Dr         was     | ury       17/03/2021 14:10:00         spliced image of the sedimentary succession at Site 1264-1265 spanning the early         ury       17/03/2021 14:28:00         ury       17/03/2021 14:28:00         ury       17/03/2021 14:28:00         ury       17/03/2021 14:29:00         ury       17/03/2021 14:29:00         ury       29/05/2021 17:33:00                                                                                                                 |
| Page 4: Inserted Anna Joy Dr
This resulted in a continuou
Oligocene to present day.
Page 4: Deleted Anna Joy Dr
205
Page 4: Deleted Anna Joy Dr
205
Page 4: Deleted Anna Joy Dr
205
Page 4: Inserted Anna Joy Dr
205
Page 4: Inserted Anna Joy Dr
Was
Page 4: Inserted Anna Joy Dr
Was
Page 4: Inserted Anna Joy Dr
Ware
Page 4: Inserted Anna Joy Dr | ury         17/03/2021 14:10:00           spliced image of the sedimentary succession at Site 1264-1265 spanning the early           ury         17/03/2021 14:28:00           ury         17/03/2021 14:28:00           ury         17/03/2021 14:28:00           ury         17/03/2021 14:29:00           ury         12/03/2021 14:29:00           ury         29/05/2021 17:33:00           ury         29/05/2021 17:33:00           ury         17/03/2021 14:31:00 |

| Page 4: Inserted Anna Joy Drury      | 17/03/2021 14:31:00 |
|--------------------------------------|---------------------|
| four                                 |                     |
| Page 4: Inserted Anna Joy Drury      | 17/03/2021 14:31:00 |
| 2011 (195-205 rmcd),                 |                     |
| Page 4: Inserted Anna Joy Drury      | 17/03/2021 15:02:00 |
| directly at the core surface of Site | 1264 archive halves |
| Page 4: Deleted Anna Joy Drury       | 17/03/2021 15:02:00 |
| during                               |                     |
| Page 4: Inserted Anna Joy Drury      | 17/03/2021 15:02:00 |
| using                                |                     |
| Page 4: Inserted Anna Joy Drury      | 17/03/2021 15:01:00 |
| .2                                   |                     |
| Page 4: Deleted Anna Joy Drury       | 17/03/2021 15:02:00 |
| down-core                            |                     |
| Page 4: Inserted Anna Joy Drury      | 17/03/2021 15:02:00 |
| a                                    |                     |
| Page 4: Inserted Anna Joy Drury      | 17/03/2021 15:01:00 |
| mm down-core and                     |                     |
| Page 4: Deleted Anna Joy Drury       | 17/03/2021 15:01:00 |
| -                                    |                     |
| Page 4: Inserted Anna Joy Drury      | 17/03/2021 15:02:00 |
| cross-core                           |                     |
| Page 4: Deleted Anna Joy Drury       | 17/03/2021 15:02:00 |
| directly at the core surface of Site | 1264 archive halves |
| Page 4: Inserted Anna Joy Drury      | 17/03/2021 15:03:00 |
| with the same slit conditions        |                     |
| Page 4: Deleted Anna Joy Drury       | 17/03/2021 15:12:00 |
| during                               |                     |
| Page 4: Inserted Anna Joy Drury      | 17/03/2021 15:12:00 |
| using                                |                     |
| Page 4: Deleted Anna Joy Drury       | 17/03/2021 15:12:00 |
| relatively                           |                     |
| Page 4: Deleted Anna Joy Drury       | 17/03/2021 15:13:00 |
| splice                               |                     |

|                                                                                                                                                                                            | Anna Joy Drury                                                                                                                               | 17/03/2021 15:13:00                                                                                                                                                                                                                                                      |
|--------------------------------------------------------------------------------------------------------------------------------------------------------------------------------------------|----------------------------------------------------------------------------------------------------------------------------------------------|--------------------------------------------------------------------------------------------------------------------------------------------------------------------------------------------------------------------------------------------------------------------------|
| accurately correla                                                                                                                                                                         | ate between holes                                                                                                                            |                                                                                                                                                                                                                                                                          |
| Page 4: Inserted                                                                                                                                                                           | Anna Joy Drury                                                                                                                               | 17/03/2021 14:32:00                                                                                                                                                                                                                                                      |
| 2011) MARUM                                                                                                                                                                                | XRF III, 10 kV/0.                                                                                                                            | 15 mA/10 s count time/Cl-Rh filter (see also                                                                                                                                                                                                                             |
| Page 4: Deleted                                                                                                                                                                            | Anna Joy Drury                                                                                                                               | 17/03/2021 14:41:00                                                                                                                                                                                                                                                      |
| (                                                                                                                                                                                          |                                                                                                                                              |                                                                                                                                                                                                                                                                          |
| Page 4: Inserted                                                                                                                                                                           | Anna Joy Drury                                                                                                                               | 17/03/2021 14:40:00                                                                                                                                                                                                                                                      |
| (Liebrand et al., 2                                                                                                                                                                        | 2016);                                                                                                                                       |                                                                                                                                                                                                                                                                          |
| Page 4: Inserted                                                                                                                                                                           | Anna Joy Drury                                                                                                                               | 17/03/2021 15:06:00                                                                                                                                                                                                                                                      |
| /Cl-Rh filter                                                                                                                                                                              |                                                                                                                                              |                                                                                                                                                                                                                                                                          |
| Page 4: Inserted                                                                                                                                                                           | Anna Joy Drury                                                                                                                               | 17/03/2021 15:08:00                                                                                                                                                                                                                                                      |
| /no filter                                                                                                                                                                                 |                                                                                                                                              |                                                                                                                                                                                                                                                                          |
| Page 4: Deleted                                                                                                                                                                            | Anna Joy Drury                                                                                                                               | 29/05/2021 17:33:00                                                                                                                                                                                                                                                      |
|                                                                                                                                                                                            |                                                                                                                                              |                                                                                                                                                                                                                                                                          |
| Page 4: Inserted                                                                                                                                                                           | Anna Joy Drury                                                                                                                               | 17/03/2021 15:08:00                                                                                                                                                                                                                                                      |
| /no filter                                                                                                                                                                                 |                                                                                                                                              |                                                                                                                                                                                                                                                                          |
| Page 5: Inserted                                                                                                                                                                           | Anna Joy Drury                                                                                                                               | 11/05/2021 16:43:00                                                                                                                                                                                                                                                      |
| All data were ins                                                                                                                                                                          | pected directly fo                                                                                                                           | llowing collection and outliers were removed if they were clearly associated with                                                                                                                                                                                        |
|                                                                                                                                                                                            |                                                                                                                                              | as a Fallowing this the ln(Co/Fa) data ware additionally described using the CODD                                                                                                                                                                                        |
| cracks and/or une                                                                                                                                                                          | even sediment surf                                                                                                                           | ace. Following this, the in(Carre) data were additionally despiked using the CODD                                                                                                                                                                                        |
| cracks and/or une
editing functions                                                                                                                                                     | even sediment surf                                                                                                                           | ace: ronowing this, the infCarre) data were additionally despiked using the CODD                                                                                                                                                                                         |
| cracks and/or une
editing functions
Page 5: Deleted                                                                                                                                  | Anna Joy Drury                                                                                                                               | 29/05/2021 17:33:00                                                                                                                                                                                                                                                      |
| cracks and/or une
editing functions
Page 5: Deleted
was                                                                                                                           | even sediment surf
Anna Joy Drury                                                                                                         | ace. Following inits, me init a rej uala were additionally despiked using me CODD
29/05/2021 17:33:00                                                                                                                                                                 |
| cracks and/or une
editing functions
Page 5: Deleted
was
Page 5: Inserted                                                                                                       | Anna Joy Drury                                                                                                                               | ace ronowing inits, me init a rej uana were additionally despited using ine CODD
29/05/2021 17:33:00
29/05/2021 17:33:00                                                                                                                                           |
| cracks and/or une
editing functions
Page 5: Deleted
Was
Page 5: Inserted
Were                                                                                               | Anna Joy Drury                                                                                                                               | ace ronowing inits, me in Carrey unit were additionally despited using ine CODD
29/05/2021 17:33:00
29/05/2021 17:33:00                                                                                                                                            |
| cracks and/or une
editing functions
Page 5: Deleted
Was
Page 5: Inserted
were
Page 5: Inserted                                                                           | Anna Joy Drury Anna Joy Drury Anna Joy Drury                                                                                                 | ace rolowing inits, me in Carrey data were additionally despited using the CODD
29/05/2021 17:33:00
29/05/2021 17:33:00                                                                                                                                            |
| cracks and/or unc
editing functions
Page 5: Deleted
was
Page 5: Inserted
wcre
Page 5: Inserted
including                                                              | Anna Joy Drury Anna Joy Drury Anna Joy Drury                                                                                                 | ace Poliswing inits, me in Carrey uan were additionally despited using ine CODD
29/05/2021 17:33:00
29/05/2021 17:33:00
17/03/2021 15:08:00                                                                                                                     |
| eracks and/or une
editing functions
Page 5: Deleted
was
Page 5: Inserted
were
Page 5: Inserted
including
Page 5: Inserted                                          | Anna Joy Drury Anna Joy Drury Anna Joy Drury Anna Joy Drury                                                                                  | 29/05/2021 17:33:00
29/05/2021 17:33:00
17/03/2021 15:08:00
29/05/2021 14:26:00                                                                                                                                                                                 |
| cracks and/or une
editing functions
Page 5: Deleted
was
Page 5: Inserted
were
Page 5: Inserted
including
Page 5: Inserted
and 2                                 | Anna Joy Drury Anna Joy Drury Anna Joy Drury Anna Joy Drury                                                                                  | 29/05/2021 17:33:00
29/05/2021 17:33:00
17/03/2021 15:08:00
29/05/2021 14:26:00                                                                                                                                                                                 |
| cracks and/or une
editing functions
Page 5: Deleted
was
Page 5: Inserted
were
Page 5: Inserted
including
Page 5: Inserted
and 2
Page 5: Inserted             | Anna Joy Drury                                                                   | acc         Following inits, me in Carrey uain were additionality despited using ine CODD           29/05/2021 17:33:00         29/05/2021 17:33:00           17/03/2021 15:08:00         29/05/2021 14:26:00           17/03/2021 15:52:00         29/05/2021 15:52:00  |
| cracks and/or une
editing functions
Page 5: Deleted
was
Page 5: Inserted
were
Page 5: Inserted
including
Page 5: Inserted
and 2
Page 5: Inserted             | Anna Joy Drury
Anna Joy Drury
Anna Joy Drury
Anna Joy Drury
Anna Joy Drury                                                       | acc         Following inits, me in Car Fej uani were administrative spinced using ine CODD           29/05/2021 17:33:00         29/05/2021 17:33:00           17/03/2021 15:08:00         29/05/2021 14:26:00           17/03/2021 15:52:00         17/03/2021 15:52:00 |
| cracks and/or une
editing functions
Page 5: Deleted
was
Page 5: Inserted
were
Page 5: Inserted
and 2
Page 5: Inserted
Page 5: Inserted                          | Anna Joy Drury
Anna Joy Drury
Anna Joy Drury
Anna Joy Drury
Anna Joy Drury
Anna Joy Drury                                     | 29/05/2021 17:33:00
29/05/2021 17:33:00
17/03/2021 15:08:00
20/05/2021 15:52:00
11/05/2021 16:52:00                                                                                                                                                          |
| cracks and/or une
editing functions
Page 5: Deleted
was
Page 5: Inserted
were
Page 5: Inserted
and 2
Page 5: Inserted
Page 5: Inserted                          | Anna Joy Drury
Anna Joy Drury
Anna Joy Drury
Anna Joy Drury
Anna Joy Drury
Anna Joy Drury                                     | 29/05/2021 17:33:00           29/05/2021 17:33:00           17/03/2021 15:08:00           29/05/2021 15:52:00           11/05/2021 16:52:00                                                                                                                              |
| cracks and/or une
editing functions
Page 5: Deleted
was
Page 5: Inserted
including
Page 5: Inserted
and 2
Page 5: Inserted
Page 5: Inserted
Page 5: Inserted | Anna Joy Drury
Anna Joy Drury | acc Poliswing inits, the init car Pej data were additionally despited using the CODD           29/05/2021 17:33:00           29/05/2021 17:33:00           17/03/2021 15:08:00           29/05/2021 14:26:00           11/05/2021 16:29:00           11/05/2021 16:29:00 |

| Page 5: Deleted Anna Joy Drury 11/05/2021 12:38:00                                                                    |
|-----------------------------------------------------------------------------------------------------------------------|
| 79.642                                                                                                                |
| Page 5: Inserted Anna Joy Drury 10/05/2021 18:10:00                                                                   |
| ±1.069                                                                                                                |
| Page 5: Deleted Anna Joy Drury 11/05/2021 12:38:00                                                                    |
|                                                                                                                       |
| Page 5: Inserted Anna Joy Drury 11/05/2021 12:40:00                                                                   |
| $2.526 \pm 0.188$                                                                                                     |
| Page 5: Deleted Anna Joy Drury 11/05/2021 12:40:00                                                                    |
| 2.6441                                                                                                                |
| Page 5: Inserted Anna Joy Drury 11/05/2021 12:47:00                                                                   |
| 0.622                                                                                                                 |
| Page 5: Deleted Anna Joy Drury 11/05/2021 12:47:00                                                                    |
| 0.7572                                                                                                                |
| Page 5: Deleted Anna Jov Drury 29/05/2021 14:27:00                                                                    |
| 5                                                                                                                     |
| Page 5: Inserted Anna Joy Drury 29/05/2021 14:27:00                                                                   |
| 6                                                                                                                     |
| Page 5: Deleted Anna Joy Drury 11/05/2021 12:52:00                                                                    |
| This calibration is within the $2\sigma$ uncertainty of the                                                           |
| Page 5: Inserted Anna Joy Drury 11/05/2021 12:51:00                                                                   |
| The                                                                                                                   |
| Page 5: Inserted Anna Joy Drury 11/05/2021 12:52:00                                                                   |
| is within the 2σ uncertainty of the new %CaCO3 calibration, which equates to ±2.2% in the calibrated %CaCO            |
| dataset.                                                                                                              |
|                                                                                                                       |
| Page 5: Inserted Anna Joy Drury 11/05/2021 12:55:00                                                                   |
| I ne uncertainty in the calibration likely originates from the scatter of the snipboard coulometry-derived %c.ac.     |
| data that were used in the calibration. This uncertainty only pertains to the absolute %CaCO3 values. The trends a    |
| cyclicity observed in the calibrated CaCO3 data are independent of this uncertainty, as these patterns are present in |
| raw ln(Ca/Fe) timeseries.                                                                                             |
|                                                                                                                       |

Page 5: Inserted Anna Joy Drury 11/05/2021 12:53:00 e new and recalibrated %CaCO3

Page 5: Deleted Anna Joy Drury 11/05/2021 12:53:00

| Page 5: Inserted Anna Joy Drury | 11/05/2021 12:54:00                               |
|---------------------------------|---------------------------------------------------|
| from Site 1264                  |                                                   |
| Page 5: Deleted Anna Joy Drury  | 29/05/2021 17:33:00                               |
| was                             |                                                   |
| Page 5: Inserted Anna Joy Drury | 29/05/2021 17:33:00                               |
| were                            |                                                   |
| Page 5: Inserted Anna Joy Drury | 11/05/2021 13:01:00                               |
|                                 | $MAR_{detrital} = MAR_{Bulk} - MAR_{CaCO_3} $ (3) |
| Page 5: Deleted Anna Joy Drury  | 29/05/2021 17:34:00                               |
| using t                         |                                                   |
| Page 5: Inserted Anna Joy Drury | 29/05/2021 17:34:00                               |
| Т                               |                                                   |
| Page 5: Inserted Anna Joy Drury | 29/05/2021 17:34:00                               |
| was                             |                                                   |
| Page 5: Inserted Anna Joy Drury | 29/05/2021 17:34:00                               |
|                                 |                                                   |
| Page 5: Deleted Anna Joy Drury  | 29/05/2021 17:34:00                               |
| and d                           |                                                   |
| Page 5: Inserted Anna Joy Drury | 29/05/2021 17:34:00                               |
| D                               |                                                   |
| Page 5: Inserted Anna Joy Drury | 29/05/2021 17:34:00                               |
| was                             |                                                   |
| Page 5: Deleted Anna Joy Drury  | 29/05/2021 14:27:00                               |
| 6                               |                                                   |
| Page 5: Inserted Anna Joy Drury | 29/05/2021 14:27:00                               |
| 7                               |                                                   |

Page 5: Inserted Anna Joy Drury 17/03/2021 15:31:00 The uncertainty in the MARs is difficult to quantify. The largest uncertainties affecting bulk, CaCO3 and detrital MARs arise from uncertainties in the  $\rho_{aya}$ , which was calculated using shipboard GRA and discrete dry density data, and the LSR, both of which are difficult to estimate. CaCO3 MARs additionally have ±2.2% 20 calibration uncertainty. However, as %CaCO3 is on bight a Site 1264, the %CaCO4 calibration uncertainty will have a smaller affect compared with the changes in LSR. Because detrital MARs are low and calculated using the difference between bulk and CaCO3 MARs, changes in detrital MARs should be treated cautiously.

| Page 6: Deleted    | Anna Joy Drury    | 17/03/2021 15:33:00            |
|--------------------|-------------------|--------------------------------|
| composite          |                   |                                |
|                    |                   |                                |
| Page 6: Inserted   | Anna Joy Drury    | 17/03/2021 15:33:00            |
| core               |                   |                                |
| Page 6: Inserted   | Anna Joy Drury    | 17/03/2021 15:33:00            |
| , leading to dupli | cated and/or miss | ing intervals in the shipboard |
|                    |                   |                                |
| Page 6: Deleted    | Anna Joy Drury    | 17/03/2021 15:33:00            |
| and                |                   |                                |
| Page 6: Incorted   | Anna Joy Drumy    | 17/02/2021 15:24:00            |
| ese misalignmen    | Is state          | 17/03/2021 13:34:00            |
| ese misuigninen    |                   |                                |
| Page 6: Deleted    | Anna Joy Drury    | 17/03/2021 15:34:00            |
| is                 |                   |                                |
|                    |                   |                                |
| Page 6: Inserted   | Anna Joy Drury    | 17/03/2021 15:34:00            |
| are                |                   |                                |
| Page 6: Deleted    | Anna Joy Drury    | 17/03/2021 15:34:00            |
| is                 |                   |                                |
|                    |                   |                                |
| Page 6: Deleted    | Anna Joy Drury    | 17/03/2021 15:35:00            |
| Using              |                   |                                |
| Dama C. Tanantad   | Anna Jaw Davan    | 17/07/2021 15:26:00            |
| Predominantly      | Anna Joy Drury    | 17/05/2021 15:56:00            |
| Treatminiantry     |                   |                                |
| Page 6: Inserted   | Anna Joy Drury    | 17/03/2021 15:36:00            |
| using              |                   |                                |
|                    |                   |                                |
| Page 6: Deleted    | Anna Joy Drury    | 18/03/2021 13:59:00            |
| 190.13             |                   |                                |
| Page 6: Inserted   | Anna Joy Drury    | 18/03/2021 13:59:00            |
| 205                | 11                |                                |
|                    |                   |                                |
| Page 6: Deleted    | Anna Joy Drury    | 29/05/2021 14:27:00            |
| 2                  |                   |                                |
|                    |                   | 20/07/2024 4 4 22 00           |
| Page 6: Inserted   | Anna Joy Drury    | 29/05/2021 14:27:00            |
| 5                  |                   |                                |
| Page 6: Deleted    | Anna Joy Drury    | 29/05/2021 17:34:00            |
| was                |                   |                                |
|                    |                   |                                |
| Page 6: Inserted   | Anna Joy Drury    | 29/05/2021 17:34:00            |

were

| Page 6: Deleted    | Anna Joy Druny   | 28/0E/2021 14:27:00   |
|--------------------|------------------|-----------------------|
| 2                  | Allia Joy Dialy  | 29/03/2021 14:27:00   |
| 2                  |                  |                       |
|                    |                  |                       |
| Page 6: Inserted   | Anna Joy Drury   | 29/05/2021 14:27:00   |
| 3                  |                  |                       |
|                    |                  |                       |
| Page 6: Deleted    | Anna Joy Drury   | 17/03/2021 15:36:00   |
| and splice         |                  |                       |
|                    |                  |                       |
| Page 6: Deleted    | Anna Joy Drury   | 24/05/2021 16:16:00   |
| Hole               |                  |                       |
|                    |                  |                       |
| Page 6: Inserted   | Anna Joy Drury   | 24/05/2021 16:16:00   |
| Core               |                  |                       |
|                    |                  |                       |
| Page 6: Inserted   | Anna Joy Drury   | 24/05/2021 16:16:00   |
| Н                  |                  | -,,                   |
|                    |                  |                       |
| Page 6: Deleted    | Anno Joy Druny   | 24/05/2021 16:16:00   |
| Fage 0. Deleteu    | Anna Joy Drury   | 24/05/2021 10:10:00   |
| Hole               |                  |                       |
|                    |                  |                       |
| Page 6: Inserted   | Anna Joy Drury   | 24/05/2021 16:16:00   |
| Core               |                  |                       |
|                    |                  |                       |
| Page 6: Inserted   | Anna Joy Drury   | 24/05/2021 16:16:00   |
| Н                  |                  |                       |
|                    |                  |                       |
| Page 6: Deleted    | Anna Joy Drury   | 17/03/2021 15:37:00   |
| in order           |                  |                       |
|                    |                  |                       |
| Page 6: Deleted    | Anna Joy Drury   | 17/03/2021 15:37:00   |
| composite          |                  |                       |
|                    |                  |                       |
| Page 6: Deleted    | Anna Joy Drury   | 17/03/2021 15:37:00   |
| and                |                  |                       |
| •                  |                  |                       |
| Page 6: Inserted   | Anna Joy Drugy   | 17/03/2021 15:37:00   |
| new                | the set of brand |                       |
|                    |                  |                       |
| Page 6: Incorted   | Anno Joy Druny   | 17/02/2021 15:27:00   |
| . age o. msetted   | Anna Joy Didry   | 1, 100/ LOLL 10:01:00 |
| and                |                  |                       |
|                    |                  |                       |
| Page 6: Deleted    | Anna Joy Drury   | 1//03/2021 15:37:00   |
| , together with th | e                |                       |
|                    |                  |                       |
| Page 6: Deleted    | Anna Joy Drury   | 17/03/2021 15:37:00   |
| composite          |                  |                       |

| Page 6: Inserted Anna Joy Drury
ary
Page 6: Deleted Anna Joy Drury
column
Page 6: Inserted Anna Joy Drury | 17/03/2021 15:40:00
17/03/2021 15:40:00 |
|-----------------------------------------------------------------------------------------------------------------------|--------------------------------------------|
| Page 6: Inserted Anna Joy Drury
ary
Page 6: Deleted Anna Joy Drury
column
Page 6: Inserted Anna Joy Drury | 17/03/2021 15:40:00
17/03/2021 15:40:00 |
| ary Page 6: Deleted Anna Joy Drury column Page 6: Inserted Anna Joy Drury                                             | 17/03/2021 15:40:00                        |
| Page 6: Deleted Anna Joy Drury
column
Page 6: Inserted Anna Joy Drury                                           | 17/03/2021 15:40:00                        |
| column Page 6: Inserted Anna Joy Drury                                                                                |                                            |
| Page 6: Inserted Anna Joy Drury                                                                                       |                                            |
|                                                                                                                       | 17/03/2021 15:40:00                        |
| succession (0-205 rmcd)                                                                                               |                                            |
| Page 6: Inserted Anna Joy Drury                                                                                       | 17/03/2021 15:41:00                        |
| Site 1264                                                                                                             |                                            |
| Page 6: Inserted Anna Joy Drury                                                                                       | 17/03/2021 15:41:00                        |
| , stratigraphic                                                                                                       |                                            |
| Page 6: Deleted Anna Joy Drury                                                                                        | 17/03/2021 15:41:00                        |
| /                                                                                                                     |                                            |
| Page 6: Deleted Anna Joy Drury                                                                                        | 29/05/2021 14:27:00                        |
| 3                                                                                                                     |                                            |
| Page 6: Inserted Anna Joy Drury                                                                                       | 29/05/2021 14:27:00                        |
| 4                                                                                                                     |                                            |
| Page 6: Inserted Anna Joy Drury                                                                                       | 11/05/2021 14:39:00                        |
| , which were                                                                                                          |                                            |
| Page 6: Deleted Anna Joy Drury                                                                                        | 11/05/2021 14:39:00                        |
|                                                                                                                       |                                            |
| Page 6: Inserted Anna Joy Drury                                                                                       | 17/03/2021 15:44:00                        |
| filled with new isotope data (Weste                                                                                   | rhold et al., 2020).                       |
| Page 6: Inserted Anna Joy Drury                                                                                       | 12/05/2021 11:51:00                        |
| XRF intensities,                                                                                                      |                                            |
| Page 6: Deleted Anna Joy Drury                                                                                        | 11/05/2021 13:15:00                        |
|                                                                                                                       |                                            |

Page 6: Inserted Anna Joy Drury 18/03/2021 13:22:00 The range of observed %CaCO, variability is close to the 2.2% uncertainty associated with the calibration. However, we are confident that both the long-term trends and short-term variability discussed below represent true changes in carbonate content, as these patterns originate in the original ln(Ca/Fe) ratio. The calibration uncertainty is most relevant to the absolute carbonate content.

| Page 7: Inserted   | Anna Joy Drury | 18/03/2021 13:39:00 |
|--------------------|----------------|---------------------|
| span 93-96%        |                |                     |
|                    |                |                     |
| Page 7: Deleted    | Anna Joy Drury | 18/03/2021 13:39:00 |
| span 93-96%        |                |                     |
| Page 7: Deleted    | Anna Joy Drury | 18/03/2021 13:39:00 |
| is                 | and soy brary  | 10/03/2021 10:05:00 |
|                    |                |                     |
| Page 7: Inserted   | Anna Joy Drury | 18/03/2021 13:39:00 |
| agrees             |                |                     |
|                    |                |                     |
| Page 7: Deleted    | Anna Joy Drury | 18/03/2021 13:39:00 |
| or                 |                |                     |
| Page 7: Inserted   | Anna Jov Drurv | 18/03/2021 13:39:00 |
| with               | ,,             |                     |
|                    |                |                     |
| Page 7: Inserted   | Anna Joy Drury | 18/03/2021 13:39:00 |
| s                  |                |                     |
|                    |                | 10/02/2021 12 10 00 |
| Page 7: Deleted 7  | Anna Joy Drury | 18/03/2021 13:40:00 |
| especially         |                |                     |
| Page 7: Inserted   | Anna Joy Drury | 18/03/2021 13:40:00 |
| especially         |                |                     |
|                    |                |                     |
| Page 7: Inserted A | Anna Joy Drury | 18/03/2021 13:40:00 |
| s                  |                |                     |
| Page 7: Incerted   | Anna Joy Drury | 18/03/2021 13:41:00 |
| s                  | ania soy stary | 20/05/2022 25:42:00 |
|                    |                |                     |
| Page 7: Inserted   | Anna Joy Drury | 18/03/2021 13:41:00 |
| s                  |                |                     |
| Dana 7. Innard     | and low Dec    | 24/05/2021 17:52:00 |
| (mid Miocana)      | anna Joy Drury | 24/05/2021 17:55:00 |
| (min mocene)       |                |                     |
| Page 7: Inserted   | Anna Joy Drury | 18/03/2021 13:42:00 |
| then               |                |                     |
|                    |                |                     |
| Page 7: Inserted A | Anna Joy Drury | 18/03/2021 13:42:00 |
| es                 |                |                     |
| Page 7: Inserted   | Anna Joy Drury | 24/05/2021 17:54:00 |
| 0                  |                | -,-,                |
| -                  |                |                     |
| Page 7: Deleted    | Anna Joy Drury | 18/03/2021 13:42:00 |
| %                  |                |                     |

| Page 7: Inserted   | Anna Joy Drury      | 18/03/2021 13:42:00                                                                  |
|--------------------|---------------------|--------------------------------------------------------------------------------------|
| the                |                     |                                                                                      |
| Page 7: Inserted   | Anna Joy Drury      | 18/03/2021 13:42:00                                                                  |
| content            |                     |                                                                                      |
| Page 7: Deleted    | Anna Joy Drury      | 18/03/2021 13:42:00                                                                  |
| remains            |                     |                                                                                      |
| Page 7: Inserted   | Anna Joy Drury      | 18/03/2021 13:42:00                                                                  |
| decreases slightly | y to                |                                                                                      |
| Page 7: Inserted   | Anna Joy Drury      | 24/05/2021 17:54:00                                                                  |
| (early Pliocene)   |                     |                                                                                      |
| Page 7: Inserted   | Anna Joy Drury      | 24/05/2021 17:54:00                                                                  |
| -early Pliocene    |                     |                                                                                      |
| Page 7: Deleted    | Anna Joy Drury      | 29/05/2021 17:35:00                                                                  |
| ,                  |                     |                                                                                      |
| Page 7: Inserted   | Anna Joy Drury      | 24/05/2021 17:58:00                                                                  |
| s                  |                     |                                                                                      |
| Page 7: Inserted   | Anna Joy Drury      | 24/05/2021 17:54:00                                                                  |
| (Pleistocene)      |                     |                                                                                      |
| Page 7: Inserted   | Anna Joy Drury      | 12/05/2021 11:52:00                                                                  |
| The Si and K in    | tensities are comp  | arable throughout the record, although Si is generally slightly higher than K (Fig   |
| 3). Both element   | s, together with F  | e and Ti intensities, display the same short-term variability and long-term trends   |
| (Fig 3 and Suppl   | ementary Figure 2   | ), indicating that these elements reflect changes in aluminosilicates. As the trends |
| of Si and K are in | iverse to those see | n in the CaCO3 content, this supports that Site 1264 is predominantly composed of    |
| carbonate and cla  | ay, with minimal i  | influence of biogenic silica. The amplitude of changes in Si and K becomes much      |
| smaller relative t | o CaCO3 content o   | changes between ~115-0 rmcd compared to ~315-115 rmcd.                               |
| Page 7: Inserted   | Anna Joy Drury      | 18/03/2021 13:50:00                                                                  |
| Because            |                     |                                                                                      |
| Page 7: Deleted    | Anna Joy Drury      | 18/03/2021 13:50:00                                                                  |
| . As a result      |                     |                                                                                      |
| Page 7: Deleted    | Anna Joy Drury      | 11/05/2021 14:00:00                                                                  |
| sedimentation rat  | les                 |                                                                                      |
| Page 7: Inserted   | Anna Joy Drury      | 11/05/2021 14:00:00                                                                  |
| LSR                |                     |                                                                                      |
|                    |                     |                                                                                      |

| Page 7: Inserted  | i Anna Joy Drury      | 11/05/2021 14:00:00                                                                     |
|-------------------|-----------------------|-----------------------------------------------------------------------------------------|
| LSR also strong   | ly affect detrital M  | IARs; however, these remain low throughout at Site 1264 (0.01-0.2 g/cm 2 /ky |
| Page 7: Inserted  | d Anna Joy Drury      | 18/03/2021 13:52:00                                                                     |
| CaCO 3 |                       |                                                                                         |
| Page 7: Deleted   | Anna Joy Drury        | 11/05/2021 14:07:00                                                                     |
| variability       |                       |                                                                                         |
| Page 7: Inserted  | i Anna Joy Drury      | 11/05/2021 14:07:00                                                                     |
| changes           |                       |                                                                                         |
| Page 7: Inserted  | i Anna Joy Drury      | 11/05/2021 14:06:00                                                                     |
| ; however, this v | /ariability is smalle | er than that variability                                                                |
| Page 7: Deleted   | Anna Joy Drury        | 11/05/2021 14:07:00                                                                     |
| superimposed u    | ipon variability      |                                                                                         |
| Page 7: Inserted  | d Anna Joy Drury      | 11/05/2021 14:07:00                                                                     |
| (see section 4.2  | )                     |                                                                                         |
| Page 7: Inserted  | d Anna Joy Drury      | 29/05/2021 14:36:00                                                                     |
| (Fig 3)           |                       |                                                                                         |
| Page 7: Deleted   | Anna Joy Drury        | 18/03/2021 14:00:00                                                                     |
| record            |                       |                                                                                         |
| Page 7: Deleted   | Anna Joy Drury        | 18/03/2021 14:00:00                                                                     |
| sequence recove   | red                   |                                                                                         |
| Page 7: Inserted  | d Anna Joy Drury      | 18/03/2021 14:00:00                                                                     |
| succession        |                       |                                                                                         |
| Page 7: Inserted  | d Anna Joy Drury      | 18/03/2021 14:00:00                                                                     |
| 1                 |                       |                                                                                         |
| Page 7: Deleted   | Anna Joy Drury        | 18/03/2021 14:00:00                                                                     |
| 3                 |                       |                                                                                         |
| Page 8: Deleted   | Anna Joy Drury        | 29/05/2021 17:35:00                                                                     |
| •                 |                       |                                                                                         |
| Page 8: Inserted  | i Anna Joy Drury      | 18/03/2021 14:01:00                                                                     |
| u                 |                       |                                                                                         |
|                   |                       |                                                                                         |
| Page 8: Deleted   | Anna Joy Drury        | 29/05/2021 17:36:00                                                                     |
| Page 8: Deleted   | Anna Joy Drury        | 29/05/2021 17:36:00                                                                     |

|                                                                                                                                                                                            | Anna Joy Drury                                                                                                    | 29/05/2021 14:28:00                                                                                                                    |
|--------------------------------------------------------------------------------------------------------------------------------------------------------------------------------------------|-------------------------------------------------------------------------------------------------------------------|----------------------------------------------------------------------------------------------------------------------------------------|
| 7                                                                                                                                                                                          |                                                                                                                   |                                                                                                                                        |
| Page 8: Inserted                                                                                                                                                                           | Anna Joy Drury                                                                                                    | 29/05/2021 14:28:00                                                                                                                    |
| 8                                                                                                                                                                                          |                                                                                                                   | ,,                                                                                                                                     |
|                                                                                                                                                                                            |                                                                                                                   |                                                                                                                                        |
| Page 8: Deleted                                                                                                                                                                            | Anna Joy Drury                                                                                                    | 29/05/2021 14:28:00                                                                                                                    |
| 0                                                                                                                                                                                          |                                                                                                                   |                                                                                                                                        |
| Page 8: Inserted                                                                                                                                                                           | Anna Joy Drury                                                                                                    | 29/05/2021 14:28:00                                                                                                                    |
| 9                                                                                                                                                                                          |                                                                                                                   |                                                                                                                                        |
| Page 8: Inserted                                                                                                                                                                           | Anna Joy Drury                                                                                                    | 29/05/2021 14:37:00                                                                                                                    |
| ; Fig 4 and Supp                                                                                                                                                                           | lementary Figure                                                                                                  | 10                                                                                                                                     |
| Page 8: Deleted                                                                                                                                                                            | Anna Joy Drury                                                                                                    | 29/05/2021 17:36:00                                                                                                                    |
|                                                                                                                                                                                            | Annu Soy Drury                                                                                                    | 23/03/2022 27/30/30                                                                                                                    |
|                                                                                                                                                                                            |                                                                                                                   |                                                                                                                                        |
| Page 8: Deleted                                                                                                                                                                            | Anna Joy Drury                                                                                                    | 18/03/2021 14:02:00                                                                                                                    |
| Lunological cycl                                                                                                                                                                           | ies broadly varyin                                                                                                | g around 2 and 0.5 m length are present in the                                                                                         |
| Page 8: Inserted                                                                                                                                                                           | Anna Joy Drury                                                                                                    | 18/03/2021 14:02:00                                                                                                                    |
| The depth-doma                                                                                                                                                                             | in                                                                                                                |                                                                                                                                        |
| Page 8: Inserted                                                                                                                                                                           | Anna Joy Drury                                                                                                    | 18/03/2021 14:25:00                                                                                                                    |
| between 205 and                                                                                                                                                                            | 190 rmcd highlig                                                                                                  | hts the lithological cycles in %CaCO3, which broadly varies around 2 and 0.5 m                                                         |
| length                                                                                                                                                                                     |                                                                                                                   |                                                                                                                                        |
|                                                                                                                                                                                            |                                                                                                                   |                                                                                                                                        |
| Page 8: Deleted                                                                                                                                                                            | Anna Joy Drury                                                                                                    | 18/03/2021 14:03:00                                                                                                                    |
| Page 8: Deleted
for the interval                                                                                                                                                        | Anna Joy Drury                                                                                                    | 18/03/2021 14:03:00                                                                                                                    |
| Page 8: Deleted
for the interval                                                                                                                                                        | Anna Joy Drury
Anna Joy Drury                                                                                  | 18/03/2021 14:03:00
18/03/2021 14:25:00                                                                                             |
| Page 8: Deleted
for the interval
Page 8: Deleted
between 205 and                                                                                                                  | Anna Joy Drury Anna Joy Drury 190 rmcd                                                                            | 18/03/2021 14:03:00
18/03/2021 14:25:00                                                                                             |
| Page 8: Deleted
for the interval
Page 8: Deleted
between 205 and                                                                                                                  | Anna Joy Drury
Anna Joy Drury
190 rmcd                                                                      | 18/03/2021 14:03:00
18/03/2021 14:25:00                                                                                             |
| Page 8: Deleted
for the interval
Page 8: Deleted
between 205 and
Page 8: Deleted
4                                                                                          | Anna Joy Drury
Anna Joy Drury
190 rmcd
Anna Joy Drury                                                    | 18/03/2021 14:03:00
18/03/2021 14:25:00
24/05/2021 20:44:00                                                                      |
| Page 8: Deleted
for the interval
Page 8: Deleted
between 205 and
Page 8: Deleted
4                                                                                          | Anna Joy Drury
Anna Joy Drury
190 rmcd
Anna Joy Drury                                                    | 18/03/2021 14:03:00
18/03/2021 14:25:00
24/05/2021 20:44:00                                                                      |
| Page 8: Deleted
for the interval
Page 8: Deleted
between 205 and
Page 8: Deleted
4
Page 8: Inserted                                                                      | Anna Joy Drury
Anna Joy Drury
190 rmcd
Anna Joy Drury
Anna Joy Drury                                  | 18/03/2021 14:03:00
18/03/2021 14:25:00
24/05/2021 20:44:00
24/05/2021 20:44:00                                               |
| Page 8: Deleted
for the interval
Page 8: Deleted
between 205 and
Page 8: Deleted
4
Page 8: Inserted
3                                                                 | Anna Joy Drury
Anna Joy Drury
190 rmed
Anna Joy Drury
Anna Joy Drury                                  | 18/03/2021 14:03:00
18/03/2021 14:25:00
24/05/2021 20:44:00
24/05/2021 20:44:00                                               |
| Page 8: Deleted
for the interval
Page 8: Deleted
between 205 and
Page 8: Deleted
4
Page 8: Inserted
3
Page 8: Deleted                                              | Anna Joy Drury
Anna Joy Drury
1190 rmcd
Anna Joy Drury
Anna Joy Drury
Anna Joy Drury               | 18/03/2021 14:03:00
18/03/2021 14:25:00
24/05/2021 20:44:00
24/05/2021 20:44:00
18/03/2021 14:04:00                        |
| Page 8: Deleted
for the interval
Page 8: Deleted
between 205 and
Page 8: Deleted
4
Page 8: Inserted
3
Page 8: Deleted
decreases to low                          | Anna Joy Drury
Anna Joy Drury
190 rmed
Anna Joy Drury
Anna Joy Drury
Anna Joy Drury
values that | 18/03/2021 14:03:00
18/03/2021 14:25:00
24/05/2021 20:44:00
24/05/2021 20:44:00
18/03/2021 14:04:00                        |
| Page 8: Deleted
for the interval
Page 8: Deleted
between 205 and
Page 8: Deleted
4
Page 8: Inserted
3
Page 8: Deleted
decreases to low
Page 8: Deleted       | Anna Joy Drury
Anna Joy Drury
190 med
Anna Joy Drury
Anna Joy Drury
values that
Anna Joy Drury  | 18/03/2021 14:03:00
18/03/2021 14:25:00
24/05/2021 20:44:00
24/05/2021 20:44:00
18/03/2021 14:04:00
24/05/2021 18:17:00 |
| Page 8: Deleted
for the interval
Page 8: Deleted
between 205 and
Page 8: Deleted
4
Page 8: Inserted
3
Page 8: Deleted
decreases to low
Page 8: Deleted
0. | Anna Joy Drury Anna Joy Drury 190 rmcd Anna Joy Drury Anna Joy Drury Anna Joy Drury values that Anna Joy Drury    | 18/03/2021 14:03:00
18/03/2021 14:25:00
24/05/2021 20:44:00
24/05/2021 20:44:00
18/03/2021 14:04:00
24/05/2021 14:17:00 |
| Page 8: Deleted
for the interval
Page 8: Deleted
between 205 and
Page 8: Deleted
4
Page 8: Inserted
3
Page 8: Deleted
decreases to low
Page 8: Deleted
0. | Anna Joy Drury
Anna Joy Drury
190 rmcd
Anna Joy Drury
Anna Joy Drury
values that
Anna Joy Drury | 18/03/2021 14:03:00
18/03/2021 14:25:00
24/05/2021 20:44:00
24/05/2021 20:44:00
18/03/2021 14:04:00
24/05/2021 18:17:00 |

| Page 8: Deleted   | Anna Joy Drury   | 24/05/2021 18:18:00 |  |
|-------------------|------------------|---------------------|--|
| kyr               |                  |                     |  |
| Page 8: Inserted  | Anna Joy Drury   | 24/05/2021 18:18:00 |  |
| Myr               |                  |                     |  |
| Page 8: Deleted   | Anna Joy Drury   | 29/05/2021 17:36:00 |  |
| 125               |                  |                     |  |
| Page 8: Inserted  | Anna Joy Drury   | 29/05/2021 17:36:00 |  |
| 110               |                  |                     |  |
| Page 8: Deleted   | Anna Joy Drury   | 12/05/2021 17:18:00 |  |
| cyclicity         |                  |                     |  |
| Page 8: Inserted  | Anna Joy Drury   | 12/05/2021 17:18:00 |  |
| variability       |                  |                     |  |
| Page 8: Deleted   | Anna Joy Drury   | 18/03/2021 14:05:00 |  |
| e.g.,             |                  |                     |  |
| Page 8: Inserted  | Anna Joy Drury   | 18/03/2021 14:05:00 |  |
| which shows       |                  |                     |  |
| Page 8: Inserted  | Anna Joy Drury   | 21/03/2021 14:36:00 |  |
| ~                 |                  |                     |  |
| Page 8: Inserted  | Anna Joy Drury   | 21/03/2021 14:36:00 |  |
| (e.g. the ~95 and | ~125 kyr cycles) | with                |  |
| Page 8: Inserted  | Anna Joy Drury   | 18/03/2021 14:06:00 |  |
| longer            |                  |                     |  |
| Page 8: Deleted   | Anna Joy Drury   | 24/05/2021 20:45:00 |  |
| ,4                |                  |                     |  |
| Page 8: Deleted   | Anna Joy Drury   | 24/05/2021 20:46:00 |  |
| 5                 |                  |                     |  |
| Page 8: Inserted  | Anna Joy Drury   | 24/05/2021 20:46:00 |  |
| 4                 |                  |                     |  |
| Page 8: Deleted   | Anna Joy Drury   | 29/05/2021 17:36:00 |  |
|                   |                  |                     |  |
| Page 8: Inserted  | Anna Joy Drury   | 18/03/2021 14:26:00 |  |
| respectively      |                  |                     |  |
| Page 8: Inserted  | Anna Joy Drury   | 18/03/2021 14:06:00 |  |

| Page 8: Deleted Anna Joy Drury    | 18/03/2021 14:06:00    |
|-----------------------------------|------------------------|
| about                             |                        |
|                                   |                        |
| Page 8: Deleted Anna Joy Drury    | 18/03/2021 14:06:00    |
| approximately                     |                        |
|                                   |                        |
| Page 8: Inserted Anna Joy Drury   | 18/03/2021 14:06:00    |
| ~                                 |                        |
|                                   |                        |
| Page 8: Deleted Anna Joy Drury    | 18/03/2021 14:07:00    |
| in the range of                   |                        |
| -                                 |                        |
| Page 8: Inserted Anna Joy Drury   | 18/03/2021 14:07:00    |
| from ~                            |                        |
|                                   |                        |
| Page 8: Inserted Anna Joy Drury   | 18/03/2021 14:07:00    |
| ~                                 |                        |
|                                   |                        |
| Page 8: Inserted Anna Joy Drury   | 18/03/2021 14:26:00    |
| in the depth-domain wavelet analy | vsis of the CaCO3 data |
|                                   |                        |
| Page 8: Deleted Anna Joy Drury    | 24/05/2021 20:45:00    |
| 4                                 |                        |
|                                   |                        |
| Page 8: Inserted Anna Joy Drury   | 24/05/2021 20:45:00    |
| 3                                 |                        |
|                                   |                        |
| Page 8: Inserted Anna Joy Drury   | 18/03/2021 14:27:00    |
| gradually shifting,               |                        |
|                                   |                        |
| Page 8: Deleted Anna Joy Drury    | 18/03/2021 14:07:00    |
| are resultant from                |                        |
|                                   |                        |
| Page 8: Inserted Anna Joy Drury   | 18/03/2021 14:07:00    |
| reflect                           |                        |
|                                   |                        |
| Page 8: Deleted Anna Joy Drury    | 18/03/2021 14:07:00    |
| , that vary                       |                        |
|                                   |                        |
| Page 8: Inserted Anna Joy Drury   | 24/05/2021 18:18:00    |
| 0                                 |                        |
|                                   |                        |
| Page 8: Deleted Anna Joy Drury    | 24/05/2021 18:18:00    |
| c                                 |                        |
|                                   |                        |
| Page 8: Inserted Anna Joy Drury   | 24/05/2021 18:18:00    |
| M                                 |                        |
|                                   |                        |
| Page 8: Deleted Anna Joy Drury    | 24/05/2021 18:18:00    |

| Page 8: Deleted   | Anna Joy Drury | 24/05/2021 18:18:00 |
|-------------------|----------------|---------------------|
|                   |                |                     |
|                   |                |                     |
| Page 8: Deleted   | Anna Joy Drury | 24/05/2021 18:18:00 |
| с                 |                |                     |
|                   |                |                     |
| Page 8: Inserted  | Anna Joy Drury | 24/05/2021 18:18:00 |
| М                 |                |                     |
|                   |                |                     |
| Page 8: Deleted   | Anna Joy Drury | 24/05/2021 18:18:00 |
| k                 |                |                     |
| -                 |                |                     |
| Dago 9: Incorted  | Anna Joy Druny | 18/02/2021 14:27:00 |
| Fage 6. Thiserteu | Anna Joy Drury | 16/05/2021 14:27:00 |
| ,                 |                |                     |
| Page 9: Deleted   | Anno Joy Druny | 18/02/2021 14:08:00 |
| Page 6: Deleteu   | Anna Joy Drury | 18/03/2021 14:08:00 |
| can               |                |                     |
| Dama Or Terrented | Anna Inu Doum  | 18/02/2021 14:08:00 |
| Page 8: Inserted  | Anna Joy Drury | 18/03/2021 14:08:00 |
| ~                 |                |                     |
|                   |                |                     |
| Page 8: Inserted  | Anna Joy Drury | 18/03/2021 14:08:00 |
| ~                 |                |                     |
|                   |                |                     |
| Page 8: Inserted  | Anna Joy Drury | 18/03/2021 14:08:00 |
| ~                 |                |                     |
|                   |                |                     |
| Page 8: Inserted  | Anna Joy Drury | 18/03/2021 14:08:00 |
| ~                 |                |                     |
|                   |                |                     |
| Page 8: Deleted   | Anna Joy Drury | 29/05/2021 14:29:00 |
| 8                 |                |                     |
|                   |                |                     |
| Page 8: Inserted  | Anna Joy Drury | 29/05/2021 14:29:00 |
| 9                 |                |                     |
|                   |                |                     |
| Page 8: Deleted   | Anna Joy Drury | 18/03/2021 14:27:00 |
| cycle             |                |                     |
| -                 |                |                     |
| Page 8: Inserted  | Anna Joy Drury | 24/05/2021 18:01:00 |
| of these cycles   |                |                     |
|                   |                |                     |
| Page 8: Inserted  | Anna Joy Drury | 18/03/2021 14:08:00 |
|                   |                |                     |
| ,                 |                |                     |
| Page 8: Deleted   | Anna lov Drury | 18/03/2021 14:08:00 |
| is still          | Anna Joy Drury | 10/03/2022 21:00:00 |
| 10 0000           |                |                     |

| Page 9: Terrented Arms Jan Deven 18/0               | 2/2021 14:00:00                                                             |
|-----------------------------------------------------|-----------------------------------------------------------------------------|
| Page 8: Inserted Anna Joy Drury 18/0                | 5/2021 14:08:00                                                             |
| remains the                                         |                                                                             |
| Page 8: Inserted Anna Joy Drury 18/0                | 3/2021 14:09:00                                                             |
| cycle                                               |                                                                             |
| Page 8: Inserted Anna Joy Drury 18/0                | 3/2021 14:10:00                                                             |
| in line with the strong ~110-kyr eccentric          | zity cycles observed                                                        |
|                                                     |                                                                             |
| Page 8: Deleted Anna Joy Drury 18/0                 | 3/2021 14:10:00                                                             |
| similar to the older interval                       |                                                                             |
| Page 8: Inserted Anna Joy Drury 18/0                | 3/2021 14:10:00                                                             |
| ~110-kyr                                            |                                                                             |
|                                                     |                                                                             |
| Page 9: Deleted Anna Joy Drury 29/0                 | 5/2021 17:36:00                                                             |
| •                                                   |                                                                             |
| Page 9: Inserted Anna Joy Drury 18/0                | 3/2021 14:12:00                                                             |
| Because of several splice revisions in the          | upper 55 rmcd of Site 1264 (see Section 3.1.),                              |
| B                                                   |                                                                             |
| Page 9: Deleted Anna Joy Drury 18/0                 | 3/2021 14:13:00                                                             |
| Annough detailed depin and age models               | are available for upper 55 fined of Sile 1204 (Ben et al., 2014), resulting |
| from several splice revisions (see Section          | 13.1.)                                                                      |
| Page 9: Inserted Anna Joy Drury 18/0                | 3/2021 14:13:00                                                             |
| , even though detailed investigations were          | e previously made (Bell et al., 2014)                                       |
|                                                     |                                                                             |
| Page 9: Inserted Anna Joy Drury 18/0                | 3/2021 14:32:00                                                             |
| Visible inspection of the CaCO 3 content | data and t                                                                  |
| Page 9: Deleted Anna Joy Drury 18/0                 | 3/2021 14:28:00                                                             |
| Т                                                   |                                                                             |
|                                                     |                                                                             |
| Page 9: Inserted Anna Joy Drury 18/0                | 3/2021 14:28:00                                                             |
| associated                                          |                                                                             |
| Page 9: Inserted Anna Joy Drury 18/0                | 3/2021 14:21:00                                                             |
| depth-domain                                        |                                                                             |
| -                                                   |                                                                             |
| Page 9: Inserted Anna Joy Drury 18/0                | 3/2021 14:32:00                                                             |
| both                                                |                                                                             |
| Page 9: Deleted Anna Joy Drurv 18/0                 | 3/2021 14:29:00                                                             |
| of the CaCO3 data in the stratigraphic dep          | pth domain between 115 and 35 rmcd                                          |
| 3                                                   |                                                                             |
| Page 9: Inserted Anna Joy Drury 18/0                | 3/2021 14:30:00                                                             |
| that there is                                       |                                                                             |

| Page 9: Deleted    | Anna Joy Drury    | 18/03/2021 14:30:00 |
|--------------------|-------------------|---------------------|
| clear              |                   |                     |
| Page 9: Inserted   | Anna Joy Drury    | 18/03/2021 14:30:00 |
| short-term         |                   |                     |
| Page 9: Inserted   | Anna Joy Drury    | 18/03/2021 14:30:00 |
| present in the dat | a between 115 an  | d 35 rmcd           |
| Page 9: Deleted    | Anna Joy Drury    | 24/05/2021 20:45:00 |
| 4                  |                   |                     |
| Page 9: Inserted   | Anna Joy Drury    | 24/05/2021 20:45:00 |
| 3                  |                   |                     |
| Page 9: Deleted    | Anna Joy Drury    | 18/03/2021 14:33:00 |
| in comparison to   | the previous dept | h intervals         |
| Page 9: Inserted   | Anna Joy Drury    | 18/03/2021 14:33:00 |
| in comparison to   | the previous dep  | th intervals        |
| Page 9: Deleted    | Anna Joy Drury    | 18/03/2021 14:30:00 |
| and                |                   |                     |
| Page 9: Inserted   | Anna Joy Drury    | 18/03/2021 14:30:00 |
| which means that   | t                 |                     |
| Page 9: Deleted    | Anna Joy Drury    | 18/03/2021 14:33:00 |
| none of            |                   |                     |
| Page 9: Inserted   | Anna Joy Drury    | 18/03/2021 14:33:00 |
| not                |                   |                     |
| Page 9: Inserted   | Anna Joy Drury    | 18/03/2021 14:30:00 |
| above the 95% le   | evel              |                     |
| Page 9: Inserted   | Anna Joy Drury    | 18/03/2021 14:30:00 |
| -                  |                   |                     |
| Page 9: Deleted    | Anna Joy Drury    | 18/03/2021 14:30:00 |
|                    |                   |                     |
| Page 9: Inserted   | Anna Joy Drury    | 18/03/2021 14:30:00 |
| wavelet analyse    | ŝ                 |                     |
| Page 9: Deleted    | Anna Joy Drury    | 18/03/2021 14:30:00 |
| above the 95% l    | evel              |                     |
| Page 9: Inserted   | Anna Joy Drury    | 18/03/2021 14:34:00 |

| Page 9: Inserted Anna Joy Drury       | 18/03/2021 14:34:00 |
|---------------------------------------|---------------------|
| -                                     |                     |
|                                       |                     |
| Page 9: Deleted Anna Joy Drury        | 18/03/2021 14:34:00 |
| to                                    |                     |
|                                       |                     |
| Page 9: Inserted Anna Joy Drury       | 18/03/2021 14:34:00 |
| -                                     |                     |
|                                       |                     |
| Page 9: Deleted Anna Joy Drury        | 18/03/2021 14:34:00 |
| 18                                    |                     |
| Page 9: Deleted Anna Joy Drury        | 18/03/2021 14:34:00 |
| From the bio-/magnetostratigraphi     | 19/99/1021119/00    |
| · · · · · · · · · · · · · · · · · · · |                     |
| Page 9: Inserted Anna Joy Drury       | 18/03/2021 14:34:00 |
| W                                     |                     |
|                                       |                     |
| Page 9: Inserted Anna Joy Drury       | 24/05/2021 18:17:00 |
| 0                                     |                     |
|                                       |                     |
| Page 9: Inserted Anna Joy Drury       | 24/05/2021 18:17:00 |
| 0                                     |                     |
| Page 0. Deleted Arms Inc. Down        | 24/05/2021 10.17.00 |
| Page 9: Deleted Anna Joy Drury        | 24/05/2021 18:17:00 |
| ciii                                  |                     |
| Page 9: Inserted Anna Joy Drury       | 24/05/2021 18:17:00 |
| m                                     |                     |
|                                       |                     |
| Page 9: Deleted Anna Joy Drury        | 24/05/2021 18:17:00 |
| kyr                                   |                     |
|                                       |                     |
| Page 9: Inserted Anna Joy Drury       | 24/05/2021 18:17:00 |
| Myr based on the bio-/magnetostr      | tigraphic ages      |
| Press 0: Deleted Arms 1: D            | 18/03/2021 14:24:00 |
| pariodicities                         | 10/03/2021 14:34:00 |
| periodicities                         |                     |
| Page 9: Inserted Anna Jov Drury       | 18/03/2021 14:34:00 |
| depth cycles                          |                     |
|                                       |                     |
| Page 9: Deleted Anna Joy Drury        | 18/03/2021 14:34:00 |
| in the depth domain                   |                     |
|                                       |                     |
| Page 9: Inserted Anna Joy Drury       | 18/03/2021 14:35:00 |
| respectively                          |                     |
|                                       |                     |
| Page 9: Inserted Anna Joy Drury       | 18/03/2021 14:34:00 |
|                                       |                     |

(~0.5 m)

| Page 9: Inserted Anna Joy Drury       | 18/03/2021 14:35:00                                                                                          |
|---------------------------------------|--------------------------------------------------------------------------------------------------------------|
| (~1 m)                                |                                                                                                              |
|                                       |                                                                                                              |
| Page 9: Inserted Anna Joy Drury       | 18/03/2021 14:36:00                                                                                          |
| (~3-4 and ~10-12 m)                   |                                                                                                              |
|                                       |                                                                                                              |
| Page 9: Deleted Anna Joy Drury        | 18/03/2021 14:36:00                                                                                          |
|                                       |                                                                                                              |
|                                       |                                                                                                              |
| Page 9: Deleted Anna Joy Drury        | 18/03/2021 14:35:00                                                                                          |
| respectively                          |                                                                                                              |
| 1 5                                   |                                                                                                              |
| Page 9: Inserted Anna Joy Drury       | 29/05/2021 14:31:00                                                                                          |
| Fig 3:                                | 25/05/2022 14/52/00                                                                                          |
|                                       |                                                                                                              |
| Page 9: Deleted Anna Joy Drury        | 29/05/2021 14-31-00                                                                                          |
| 6                                     | 23/05/2022 24/52/00                                                                                          |
| 0                                     |                                                                                                              |
| Page 9: Incorted Anna Joy Drugs       | 20/05/2021 14:21:00                                                                                          |
| Page 5. Tilserteu Allia Joy Diury     | 25/05/2021 14.51.00                                                                                          |
| 8                                     |                                                                                                              |
|                                       | 20/05/2024 4 4 24 40                                                                                         |
| Page 9: Deleted Anna Joy Drury        | 29/05/2021 14:31:00                                                                                          |
| 8                                     |                                                                                                              |
|                                       |                                                                                                              |
| Page 9: Inserted Anna Joy Drury       | 29/05/2021 14:31:00                                                                                          |
| 9                                     |                                                                                                              |
|                                       |                                                                                                              |
| Page 9: Inserted Anna Joy Drury       | 18/03/2021 14:40:00                                                                                          |
| В                                     |                                                                                                              |
|                                       |                                                                                                              |
| Page 9: Deleted Anna Joy Drury        | 18/03/2021 14:39:00                                                                                          |
| For part of this depth interval (55-3 | $35$ rmcd), both CaCO 3 estimate data and benthic foraminiferal $\delta^{18}$ O data is available |
|                                       |                                                                                                              |
| Page 9: Inserted Anna Joy Drury       | 18/03/2021 14:39:00                                                                                          |
| etween 55 and 35 rmcd                 |                                                                                                              |
|                                       |                                                                                                              |
| Page 9: Inserted Anna Joy Drury       | 18/03/2021 14:40:00                                                                                          |
| w                                     |                                                                                                              |
|                                       |                                                                                                              |
| Page 9: Deleted Anna Joy Drury        | 18/03/2021 14:40:00                                                                                          |
| and w                                 |                                                                                                              |
|                                       |                                                                                                              |
| Page 9: Inserted Anna Joy Drury       | 18/03/2021 14:39:00                                                                                          |
| CaCO3 content data and benthic for    | raminiferal δ 18 O data                                                                           |
|                                       |                                                                                                              |
| Page 9: Deleted Anna Joy Drury        | 18/03/2021 14:41:00                                                                                          |
| these two proxy records               |                                                                                                              |
|                                       |                                                                                                              |

| Page 9: Deleted Anna Joy Drury         | 29/05/2021 14:29:00 |
|----------------------------------------|---------------------|
| 9                                      |                     |
|                                        |                     |
| Page 9: Inserted Anna Joy Drury        | 29/05/2021 14:29:00 |
| 11                                     |                     |
|                                        |                     |
| Page 9: Deleted Anna Joy Drury         | 29/05/2021 17:36:00 |
|                                        |                     |
|                                        |                     |
| Page 9: Deleted Anna Joy Drury         | 29/05/2021 17:36:00 |
|                                        |                     |
|                                        |                     |
| Page 9: Deleted Anna Joy Drury         | 18/03/2021 14:41:00 |
| In general, clear                      |                     |
|                                        |                     |
| At Site 1264 place                     | 10/03/2021 14:42:00 |
| At Site 1204, cidar                    |                     |
| Page 9: Incorted Appa Ic.: Down        | 18/02/2021 14:41:00 |
| cenerally                              | 10/03/2021 14:41:00 |
| generally                              |                     |
| Page 9: Incerted Anna Joy Drury        | 18/03/2021 14:42:00 |
| depth-domain CaCO 2 content | 10/05/2022 1442.00  |
| depin donam cucoy content              |                     |
| Page 9: Deleted Anna Joy Drury         | 18/03/2021 14:42:00 |
| of the Site 1264 CaCO3 content         |                     |
|                                        |                     |
| Page 9: Deleted Anna Joy Drury         | 18/03/2021 14:42:00 |
| apart from                             |                     |
|                                        |                     |
| Page 9: Inserted Anna Joy Drury        | 18/03/2021 14:42:00 |
| except for                             |                     |
|                                        |                     |
| Page 9: Deleted Anna Joy Drury         | 18/03/2021 14:42:00 |
| somewhat                               |                     |
|                                        |                     |
| Page 9: Inserted Anna Joy Drury        | 18/03/2021 14:42:00 |
| occasional                             |                     |
|                                        | 10/00/2004 11 10 00 |
| Page 9: Inserted Anna Joy Drury        | 18/03/2021 14:43:00 |
| ~1.0-1.5 m                             |                     |
| Rado & Deleted Appa low Drunk          | 18/02/2021 14:42:00 |
| with pariodicities of 1.0 to 1.5 m     | 10/03/2021 14.43.00 |
| with periodicities of 1.0 to 1.5 Ill   |                     |
| Page 9: Deleted Anna Joy Drury         | 18/03/2021 14:43:00 |
| are able to                            |                     |
| are usie to                            |                     |
| Page 9: Inserted Anna Joy Drury        | 18/03/2021 14:43:00 |
| can                                    |                     |
|                                        |                     |

| Page 9: Deleted Anna Joy Drury 18/03/2021 14:43:00                                                               |
|------------------------------------------------------------------------------------------------------------------|
| se                                                                                                               |
|                                                                                                                  |
| Page 9: Inserted Anna Joy Drury 18/03/2021 14:43:00                                                              |
| CaCO 3 content                                                                                        |
|                                                                                                                  |
| Page 9: Deleted Anna Joy Drury 18/03/2021 14:43:00                                                               |
| ir                                                                                                               |
|                                                                                                                  |
| Page 9: Inserted Anna Joy Drury 18/03/2021 14:43:00                                                              |
| of these cycles                                                                                                  |
|                                                                                                                  |
| Page 9: Deleted Anna Joy Drury 18/03/2021 14:44:00                                                               |
| not as pronounced                                                                                                |
| Page 9: Inserted Anna Joy Drury 18/03/2021 14:44:00                                                              |
| muted                                                                                                            |
|                                                                                                                  |
| Page 9: Deleted Anna Joy Drury 18/03/2021 14:44:00                                                               |
| interval                                                                                                         |
|                                                                                                                  |
| Page 9: Inserted Anna Joy Drury 18/03/2021 14:44:00                                                              |
| cycles observed                                                                                                  |
|                                                                                                                  |
| Page 9: Moved to page 9 (Move #1) Anna Joy Drury 18/03/2021 14:51:00                                             |
| We derive averaged LSR of <1 cm/kyr for this interval based on the initial bio-/magnetostratigraphic age model.  |
| Page 9: Inserted Anna Joy Drury 18/03/2021 14:51:00                                                              |
| appear to                                                                                                        |
|                                                                                                                  |
| Page 9: Inserted Anna Joy Drury 18/03/2021 14:51:00                                                              |
| in the upper 35 m                                                                                                |
|                                                                                                                  |
| Page 9: Moved to page 9 (Move #2) Anna Joy Drury 18/03/2021 14:53:00                                             |
| Based on the initial age model we note absence of clear precession and obliquity paced cyclicity in both benthic |
| for
aminiferal $\delta^{18}O$ and CaCO3 content records during the last 2.5 Ma (Supplementary Figure 8).      |
| Page 9: Moved from page 9 (Move #1) Anna Joy Drury 18/03/2021 14:51:00                                           |
| We derive averaged LSR of <10 cm/Mkyr for this0-35 mcd interval based on the initial bio-/magnetostratigraphic   |
| ana madal                                                                                                        |
| age model.                                                                                                       |
| Page 9: Inserted Anna Joy Drury 24/05/2021 18:19:00                                                              |
| 0                                                                                                                |
| Page 9: Deleted Anna Joy Drury 24/05/2021 18:19:00                                                               |
| c                                                                                                                |
|                                                                                                                  |
| Page 9: Inserted Anna Joy Drury 24/05/2021 18:19:00                                                              |
| M                                                                                                                |

| Page 9: Deleted Anna Joy Drury                 | 24/05/2021 18:19:00                                                               |
|------------------------------------------------|-----------------------------------------------------------------------------------|
| k                                              |                                                                                   |
| Page 9: Deleted Anna Joy Drury                 | 18/03/2021 14:52:00                                                               |
| this                                           |                                                                                   |
| Page 9: Inserted Anna Joy Drury                | 18/03/2021 14:52:00                                                               |
| 0-35 rmcd                                      |                                                                                   |
| Page 9: Deleted Anna Joy Drury                 | 18/03/2021 14:52:00                                                               |
| interval                                       |                                                                                   |
| Page 9: Inserted Anna Joy Drury                | 18/03/2021 14:52:00                                                               |
| observed                                       |                                                                                   |
| Page 9: Deleted Anna Joy Drury                 | 18/03/2021 14:52:00                                                               |
| periodicity                                    |                                                                                   |
| Page 9: Inserted Anna Joy Drury                | 18/03/2021 14:52:00                                                               |
| cycles                                         |                                                                                   |
| Page 9: Deleted Anna Joy Drury                 | 18/03/2021 14:52:00                                                               |
| is                                             |                                                                                   |
| Page 9: Inserted Anna Joy Drury                | 18/03/2021 14:52:00                                                               |
| are                                            |                                                                                   |
| Page 9: Deleted Anna Joy Drury                 | 18/03/2021 14:52:00                                                               |
| either                                         |                                                                                   |
| Page 9: Inserted Anna Joy Drury                | 24/05/2021 18:28:00                                                               |
| (Bailey et al., 2013)                          |                                                                                   |
| Page 9: Moved from page 9 (Move                | e #2) Anna Joy Drury 18/03/2021 14:53:00                                          |
| Based on the initial age model v               | ve note absence of clear precession and obliquity paced cyclicity in both benthic |
| for
aminiferal $\delta^{18}O$ and CaCO3 con | ntent records during the last 2.5 Ma (Supplementary Figure 89).                   |
| Page 9: Deleted Anna Joy Drury                 | 29/05/2021 14:32:00                                                               |
| 8                                              |                                                                                   |
| Page 9: Inserted Anna Joy Drury                | 29/05/2021 14:32:00                                                               |
| 9                                              |                                                                                   |
| Page 9: Deleted Anna Joy Drury                 | 29/05/2021 17:37:00                                                               |
| •                                              |                                                                                   |
|                                                |                                                                                   |
| Page 9: Inserted Anna Joy Drury                | 21/03/2021 14:23:00                                                               |
| (                                              |                                                                                   |

| Page 9: Deleted An    | na Joy Drury 21/03/    | /2021 14:23:00                               |
|-----------------------|------------------------|----------------------------------------------|
| between               |                        |                                              |
| Dama (), Dalahad An   | 1 D 21/02/             | (2021 14:22:00                               |
| Page 9: Deleted An    | na Joy Drury 21/03/    | 2021 14:23:00                                |
| and                   |                        |                                              |
| Page 9: Incorted An   | no lov Drumy 21/02/    | /2021 14:22:00                               |
| to                    | 11a Joy Diary 21/03/   | 2021 14:23:00                                |
| 10                    |                        |                                              |
| Page 9: Inserted An   | na Joy Drury 21/03/    | /2021 14:23:00                               |
| )                     |                        |                                              |
|                       |                        |                                              |
| Page 9: Deleted An    | na Joy Drury 21/03/    | /2021 14:22:00                               |
| %                     |                        |                                              |
|                       |                        |                                              |
| Page 9: Inserted An   | na Joy Drury 21/03/    | /2021 14:22:00                               |
| content               |                        |                                              |
|                       |                        |                                              |
| Page 9: Deleted An    | na Joy Drury 21/03/    | /2021 14:23:00                               |
| spanning              |                        |                                              |
|                       |                        |                                              |
| Page 9: Inserted An   | na Joy Drury 21/03/.   | 2021 14:23:00                                |
| (                     |                        |                                              |
| Page 9: Inserted An   | na Joy Drury 21/03/    | /2021 14:23:00                               |
| )                     |                        |                                              |
| ,                     |                        |                                              |
| Page 9: Inserted An   | na Joy Drury 21/03/    | /2021 14:23:00                               |
| a                     |                        |                                              |
|                       |                        |                                              |
| Page 9: Inserted An   | na Joy Drury 24/05/    | /2021 18:26:00                               |
| on                    |                        |                                              |
|                       |                        |                                              |
| Page 9: Deleted An    | na Joy Drury 24/05/    | /2021 18:26:00                               |
| ng                    |                        |                                              |
| Dama Or Taxanda Am    |                        | (2021 14-22-00                               |
| batwaan               | na Joy Drury 21/03/.   | 2021 14:25:00                                |
| between               |                        |                                              |
| Page 10: Inserted     | Anna Joy Drury         | 21/03/2021 14:24:00                          |
| Because of the splice | e revisions between 27 | and 149 rmcd at Site 1264, we re-evaluated t |
|                       |                        |                                              |
| Page 10: Deleted      | Anna Joy Drury         | 21/03/2021 14:24:00                          |
| Т                     |                        |                                              |
| -                     |                        |                                              |
| Page 10: Deleted      | Anna Joy Drury         | 21/03/2021 14:26:00                          |
| has to be re-evaluate | d                      |                                              |
| Dana 10. Dalati       | Anna Jau P             | 21/02/2021 14:26:00                          |
| Page 10: Deleted      | Anna Joy Drury         | 21/03/2021 14:26:00                          |

| Page 10: Inserted         | Anna Joy Drury         | 21/03/2021 14:26:00                 |
|---------------------------|------------------------|-------------------------------------|
| /                         |                        |                                     |
| Page 10: Deleted          | Anna Joy Drury         | 21/03/2021 14:26:00                 |
| (                         |                        |                                     |
| Page 10: Deleted          | Anna Joy Drury         | 21/03/2021 14:26:00                 |
| ), resulting from the spl | lice revisions betwee  | en 27 and 149 rmcd at Site 1264     |
| Page 10: Inserted         | Anna Joy Drury         | 21/03/2021 14:28:00                 |
| we updated                |                        |                                     |
| Page 10: Deleted          | Anna Joy Drury         | 21/03/2021 14:28:00                 |
| were updated              |                        |                                     |
| Page 10: Inserted         | Anna Joy Drury         | 21/03/2021 14:28:00                 |
| cumulative                |                        |                                     |
| Page 10: Inserted         | Anna Joy Drury         | 21/03/2021 14:28:00                 |
| shift in the revised      |                        |                                     |
| Page 10: Deleted          | Anna Joy Drury         | 21/03/2021 14:28:00                 |
| cumulative                |                        |                                     |
| Page 10: Inserted         | Anna Joy Drury         | 21/03/2021 14:28:00                 |
| composite                 |                        |                                     |
| Page 10: Deleted          | Anna Joy Drury         | 21/03/2021 14:29:00                 |
| shift due to depth mode   | el/splice revisions in |                                     |
| Page 10: Inserted         | Anna Joy Drury         | 21/03/2021 14:29:00                 |
| of                        |                        |                                     |
| Page 10: Inserted         | Anna Joy Drury         | 21/03/2021 14:30:00                 |
| in the depth-domain       |                        |                                     |
| Page 10: Deleted          | Anna Joy Drury         | 21/03/2021 14:29:00                 |
| %                         |                        |                                     |
| Page 10: Deleted          | Anna Joy Drury         | 21/03/2021 14:29:00                 |
| record                    |                        |                                     |
| Page 10: Inserted         | Anna Joy Drury         | 21/03/2021 14:29:00                 |
| content record            |                        |                                     |
| Page 10: Inserted         | Anna Joy Drury         | 11/05/2021 14:32:00                 |
| using the flexible best-  | practice quidelines of | outlined in Sinnesael et al. (2019) |

| Page 10: Deleted         | Anna Joy Drury        | 21/03/2021 14:30:00 |
|--------------------------|-----------------------|---------------------|
| However,                 |                       |                     |
| Page 10: Inserted        | Anna Joy Drury        | 21/03/2021 14:30:00 |
| В                        |                       |                     |
| Page 10: Deleted         | Anna Joy Drury        | 21/03/2021 14:30:00 |
| b                        |                       |                     |
| Page 10: Deleted         | Anna Joy Drury        | 21/03/2021 14:34:00 |
| orbital forcing          |                       |                     |
| Page 10: Inserted        | Anna Joy Drury        | 21/03/2021 14:34:00 |
| of eccentricity (E), obl | iquity (T) and pr